# ANaGRAM: A Natural Gradient Relative to Adapted Model for efficient PINNs learning

**Nilo Schwencke**
Tau team, INRIA-Saclay
A&O team, LISN, Université Paris-Saclay
91190, Gif-sur-Yvette, France
`nilo.schwencke@protonmail.com`

**Cyril Furtlehner**
Tau team, INRIA-Saclay
A&O team, LISN, Université Paris-Saclay
91190, Gif-sur-Yvette, France
`cyril.furtlehner@inria.fr`

## Abstract

In the recent years, Physics Informed Neural Networks (PINNs) have received strong interest as a method to solve PDE driven systems, in particular for data assimilation purpose. This method is still in its infancy, with many shortcomings and failures that remain not properly understood. In this paper we propose a natural gradient approach to PINNs which contributes to speed-up and improve the accuracy of the training. Based on an in depth analysis of the differential geometric structures of the problem, we come up with two distinct contributions: (i) a new natural gradient algorithm that scales as $\min(P^2S, S^2P)$, where $P$ is the number of parameters, and $S$ the batch size; (ii) a mathematically principled reformulation of the PINNs problem that allows the extension of natural gradient to it, with proved connections to Green's function theory.

## 1 Introduction

Following the spectacular success of neural networks for over a decade (LeCun et al., 2015), intensive work has been carried out to apply these methods to numerical analysis (Cuomo et al., 2022). In particular, following the pioneering work of Dissanayake and Phan-Thien (1994) and Lagaris et al. (1998), Raissi et al. (2019a) have introduced Physics Informed Neural Networks (PINNs), a method designed to approximate solutions of partial differential equations (PDEs), using deep neural networks. Theoretically based on the universal approximation theorem of neural networks (Leshno et al., 1993), and put into practice by automatic differentiation (Baydin et al., 2018) for the computation of differential operators, this method has enjoyed a number of successes in fields as diverse as fluid mechanics (Raissi et al., 2019b;c; Sun et al., 2020; Raissi et al., 2020; Jin et al., 2021; de Wolff et al., 2021), bio-engineering (Sahli Costabal et al., 2020; Kissas et al., 2020) or free boundary problems (Wang and Perdikaris, 2021). Nevertheless, many limitations have been pointed out, notably the inability of these methods in their current formulation to obtain high-precision approximations when no additional data is provided (Krishnapriyan et al., 2021; Wang et al., 2021; Karnakov et al., 2022; Zeng et al., 2022). Recent work by Müller and Zeinhofer (2023), however, has substantially altered this state of affairs, proposing an algorithm similar to natural gradient methods in case of linear operator (*cf.* Section E), that achieves accuracies several orders of magnitude above previous methods.

**Contributions:** Müller and Zeinhofer (2024) argue for the need to take function-space geometry into account in order to further understand and improve scientific machine-learning methods. With this paper, we intend to support and extend their approach by making several contributions:

    (i) We highlight a principled mathematical framework that restates natural gradient in an equivalent, yet simpler way, leading us to propose ANaGRAM, a general-purpose natural gradient algorithm of reduced complexity $\mathcal{O}(\min(P^2 S, S^2 P))$ compared to $\mathcal{O}(P^3)$, where $P = \#parameters$ and $S = \#batch\ samples$.

    (ii) We reinterpret the PINNs framework from a functional analysis perspective in order to extend ANaGRAM to the PINN's context in a straightforward manner.

    (iii) We establish a direct correspondence between ANaGRAM for PINNs and the Green's function of the operator on the tangent space.

The rest of this article is organized as follows: in Section 2, after introducing neural networks and parametric models in Section 2.1 from a functional analysis perspective, we review two concepts crucial to our work: PINNs framework in Section 2.2, and natural gradient in Section 2.3. In Section 3, we introduce the notions of empirical tangent space and an expression for the corresponding notion of empirical natural gradient leading to ANaGRAM, described in algorithm 1. In Section 4, after reinterpreting PINNs as a regression problem from the right functional perspective in Section 4.1, yielding ANaGRAM algorithm 2 for PINNs, we state in Section 4.2 that natural gradient matches the Green's function of the operator on the tangent space and analyse the consequence of this on the interpretation of PINNs training process under ANaGRAM. Finally, in Section 5, we show empirical evidences of the performance of ANaGRAM on a selected benchmark of PDEs.

## 2 POSITION OF THE PROBLEM

### 2.1 NEURAL NETWORKS AND PARAMETRIC MODEL

Our starting point is the following functional definition of parametric models, of which neural networks are a non-linear special case:

**Definition 1** (Parametric model). Given a domain $\Omega$ of $\mathbb{R}^n$, $\mathbb{K} \in \{\mathbb{R}, \mathbb{C}\}$ and a Hilbert space $\mathcal{H}$ consisting of functions $\Omega \to \mathbb{K}^m$, a parametric model is a differentiable functional:

$$u : \begin{cases} \mathbb{R}^P & \to & \mathcal{H} \\ \boldsymbol{\theta} & \mapsto & \big(\boldsymbol{x} \in \Omega \mapsto u(\boldsymbol{x}; \boldsymbol{\theta})\big) \end{cases} . \tag{1}$$

To prevent confusion, we will write $u_{|\boldsymbol{\theta}}(\boldsymbol{x})$ instead of $u\big(\boldsymbol{x}; \boldsymbol{\theta}\big)$, for all $\boldsymbol{x} \in \Omega$.

Since a parametric model is differentiable by definition, we can define its differential:

**Definition 2** (Differential of a parametric model). Let $u : \mathbb{R}^P \to \mathcal{H}$ be a parametric model and $\boldsymbol{\theta} \in \mathbb{R}^P$. Then the differential of the parametric model $u$ in the parameter $\boldsymbol{\theta}$ is:

$$\mathrm{d}u_{|\boldsymbol{\theta}} : \begin{cases} \mathbb{R}^P & \to & \mathcal{H} \\ \boldsymbol{h} & \mapsto & \sum_{p=1}^P \boldsymbol{h}_p \frac{\partial u}{\partial \boldsymbol{\theta}_p} \end{cases} . \tag{2}$$

To simplify notations, we will write for all $1 \leqslant p \leqslant P$ and for all $\boldsymbol{\theta} \in \mathbb{R}^P$, $\partial_p u_{|\boldsymbol{\theta}}$, instead of $\frac{\partial u}{\partial \boldsymbol{\theta}_p}$.

Given a parametric model $u$, we can define the following two objects of interest:

**The image set of** $u$ : this is the set of functions reached by $u$, *i.e.* :

$$\mathcal{M} := \mathrm{Im}\, u := \big\{u_{|\boldsymbol{\theta}} \ : \ \boldsymbol{\theta} \in \mathbb{R}^P\big\} \subset \mathcal{H}. \tag{3}$$

    Although not strictly rigorous[1], $\mathcal{M}$ is often considered in deep-learning as a differential submanifold of $\mathcal{H}$, so we will keep this analogy in mind for pedagogical purposes.

**The tangent space of** $u$ **at** $\boldsymbol{\theta}$ : this is the image set of the differential of $u$ at $\boldsymbol{\theta}$, *i.e.* the linear subspace of $\mathcal{H}$ consisting of the functions reached by $\mathrm{d}u_{|\boldsymbol{\theta}}$, *i.e.* :

$$T_{\boldsymbol{\theta}}\mathcal{M} := \mathrm{Im}\, \mathrm{d}u_{|\boldsymbol{\theta}} = \mathrm{Span}\,\big(\partial_p u_{|\boldsymbol{\theta}} \ : \ 1 \leqslant p \leqslant P\big) \subset \mathcal{H}. \tag{4}$$

    Once again, this definition is made with reference to differential geometry.

We give several examples of Parametric models in Section B. We now introduce PINNs.

---

[1]In particular, because $u$ may not be injective.

## 2.2 Physics Informed Neural Networks (PINNs)

As in Definition 1, let us consider a domain $\Omega$ of $\mathbb{R}^n$ endowed with a probability measure $\mu$, $\mathbb{K} \in \{\mathbb{R}, \mathbb{C}\}$, $\partial\Omega$ its boundary endowed with a probability measure $\sigma$, and $\mathcal{H}$ a Hilbert space composed of functions $\Omega \to \mathbb{K}^m$. Then let us consider two functional (*inter alia* differential) operators:

$$D : \begin{cases} \mathcal{H} & \to & \mathrm{L}^2(\Omega \to \mathbb{R}, \mu) \\ u & \mapsto & D[u] \end{cases}, \qquad B : \begin{cases} \mathcal{H} & \to & \mathrm{L}^2(\partial\Omega \to \mathbb{R}, \sigma) \\ u & \mapsto & B[u] \end{cases}, \qquad (5)$$

that we will assume to be differentiable[2]. We can then consider the functional equation (*inter alia* PDE):

$$\begin{cases} D(u) = f \in \mathrm{L}^2(\Omega \to \mathbb{R}, \mu) & \text{in } \Omega \\ B(u) = g \in \mathrm{L}^2(\partial\Omega \to \mathbb{R}, \sigma) & \text{on } \partial\Omega \end{cases}. \qquad (6)$$

The PINNs framework, as introduced by Raissi et al. (2019a) consists then in approximating a solution to the PDE by making the ansatz $u = u_{|\boldsymbol{\theta}}$, with $u_{|\boldsymbol{\theta}}$ a neural network, sampling points $(x_i^D)_{1 \leqslant i \leqslant S_D}$ in $\Omega$ according to $\mu$, $(x_i^B)_{1 \leqslant i \leqslant S_B}$ in $\partial\Omega$ according to $\sigma$ and then to optimize the loss:

$$\ell(\boldsymbol{\theta}) := \frac{1}{2S_D} \sum_{i=1}^{S_D} \left( D[u_{|\boldsymbol{\theta}}](x_i^D) - f(x_i^D) \right)^2 + \frac{1}{2S_B} \sum_{i=1}^{S_B} \left( B[u_{|\boldsymbol{\theta}}](x_i^B) - g(x_i^B) \right)^2, \qquad (7)$$

by classical gradient descent techniques, used in the context of deep learning, such as Adam (Kingma and Ba, 2014), or L-BFGS (Liu and Nocedal, 1989). One of the cornerstones of Raissi et al. (2019a) is also to use automatic differentiation (Baydin et al., 2018) to compute the operators $D$ and $B$, thus obtaining quasi-exact calculations, whereas most classic techniques require either approximating operators with Finite Differences, or carrying out the calculations manually with Finite Elements.

Although appealing due to its simplicity and relative ease of implementation, this approach suffers from several well-documented empirical pathologies (Krishnapriyan et al., 2021; Wang et al., 2021; Grossmann et al., 2024), which can be understood as an ill conditioned problem (De Ryck et al., 2024; Liu et al., 2024) and for which several *ad hoc* procedures has been proposed (Karnakov et al., 2022; Zeng et al., 2022; McClenny and Braga-Neto, 2022). Following Müller and Zeinhofer (2024), we argue in this work that the key point is rather to theoretically understand the geometry of the problem and adapt PINNs training accordingly.

## 2.3 Natural Gradient

Natural gradient has been introduced, in the context of Information Geometry by Amari and Douglas (1998). Given a loss: $\ell : \boldsymbol{\theta} \to \mathbb{R}^+$, the gradient descent:

$$\boldsymbol{\theta}_{t+1} \leftarrow \boldsymbol{\theta}_t - \eta \, \nabla\ell, \qquad (8)$$

is replaced by the update:

$$\boldsymbol{\theta}_{t+1} \leftarrow \boldsymbol{\theta}_t - \eta \, F_{\boldsymbol{\theta}_t}^\dagger \nabla\ell, \qquad (9)$$

with $F_{\boldsymbol{\theta}_t}$ being the Gram-Matrix associated to a Fisher-Rao information metric (Amari, 2016) or equivalently, the Hessian of some Kullback-Leibler divergence (Kullback and Leibler, 1951), and $\dagger$ the Moore-Penrose pseudo-inverse. This notion has been later further extended to the more abstract setting of Riemannian metrics in the context of neural-networks by Ollivier (2015). In this case, given a Riemannian-(pseudo) metric $\mathcal{G}_{\boldsymbol{\theta}}$, the gradient-descent update is replaced by:

$$\boldsymbol{\theta}_{t+1} \leftarrow \boldsymbol{\theta}_t - \eta G_{\boldsymbol{\theta}_t}^\dagger \nabla\ell, \qquad (10)$$

where $G_{\boldsymbol{\theta}_t \, p,q} := \mathcal{G}_{\boldsymbol{\theta}_t}(\partial_p u_{|\boldsymbol{\theta}_t}, \partial_q u_{|\boldsymbol{\theta}_t})$ is the Gram matrix of partial derivatives relative to $\mathcal{G}_{\boldsymbol{\theta}_t}$. Despite its mathematically principled advantage, natural gradient suffers from its computational cost, which makes it prohibitive, if not untractable for real world applications. Indeed:

- Computation of the Gram matrix $G_{\boldsymbol{\theta}_t}$ is at least quadratic in the number of parameters.

- Inversion of $G_{\boldsymbol{\theta}_t}$ is cubic in the number of parameters.

---

[2]It can be shown that, if $D$ and $B$ are defined and differentiable on $\mathcal{C}^\infty(\Omega \to \mathbb{K}^m)$ then such a $\mathcal{H}$ always exists; *cf.* chapter 12 of Berezansky et al. (1996).

Different approaches have been proposed to circumvent this limitations. The most prominent one is K-FAC introduced by Heskes (2000) and further extended by Martens and Grosse (2015); Grosse and Martens (2016), which approximates the Gram matrix by block-diagonal matrices. This approximation can be understood as making the ansatz that the partial derivatives of weights belonging to different layers are orthogonal. A refinement of this method has been proposed by George et al. (2018), in which the eigen-structure of the block-diagonal matrices are carefully taken into account in order to provide a better approximation of the diagonal rescaling induced by the inversion of the Gram matrix. This method has been extented to PINNs by Dangel et al. (2025). In a completely different vein, Ollivier (2017) has proposed a statistical approach that has been proved to converge to the natural gradient update in the $0$ learning rate limit.

To conclude this section, let us give a more geometric interpretation of natural gradient. To this end, let us consider the standard quadratic regression problem :

$$\ell(\boldsymbol{\theta}) := \frac{1}{2S} \sum_{i=1}^{S} \left( u_{|\boldsymbol{\theta}}(x_i) - f(x_i) \right)^2 , \tag{11}$$

with $u_{|\boldsymbol{\theta}}$ a parametric model, for instance a neural-network, $(x_i)$ sampled from some probability measure $\mu$ on some domain $\Omega$ of $\mathbb{R}^N$. In the limit $S \to \infty$ (population limit), this loss can be reinterpreted as the evaluation at $u_{|\boldsymbol{\theta}}$ of the functional loss:

$$\mathcal{L} : v \in \mathrm{L}^2(\Omega, \mu) \mapsto \frac{1}{2} \|v - f\|_{\mathrm{L}^2(\Omega,\mu)}^2 . \tag{12}$$

Taking the Fréchet derivative, one gets: for all $v, h \in \mathrm{L}^2(\Omega, \mu)$

$$\mathrm{d}\mathcal{L}|_v(h) = \langle v - f , h \rangle_{\mathrm{L}^2(\Omega,\mu)} , \tag{13}$$

*i.e.* the functional gradient of $\mathcal{L}$ is $\nabla \mathcal{L}_{|v} := v - f$. As noted for instance in Verbockhaven et al. (2024), Natural gradient has then to be interpreted from the functional point of view as the projection of $\nabla \mathcal{L}_{|u_{|\boldsymbol{\theta}}}$ onto the tangent space $T_{\boldsymbol{\theta}}\mathcal{M}$ with respect to the $\mathrm{L}^2(\Omega, \mu)$ metric. However, this functional update must be converted into a parameter space update. Since the parameter space $\mathbb{R}^P$ is somehow identified with $T_{\boldsymbol{\theta}}\mathcal{M}$ *via* the differential application $\mathrm{d}u_{|\boldsymbol{\theta}}$, it would be sufficient to take the inverse of this application to obtain the parametric update. In general $\mathrm{d}u_{|\boldsymbol{\theta}}$ is not invertible but at least it admits a pseudo-inverse $\mathrm{d}u_{|\boldsymbol{\theta}}^\dagger$. Moreover, since $T_{\boldsymbol{\theta}}\mathcal{M} = \mathrm{Im}\, \mathrm{d}u_{|\boldsymbol{\theta}}$ by definition, $\mathrm{d}u_{|\boldsymbol{\theta}}^\dagger$ is defined on all $T_{\boldsymbol{\theta}}\mathcal{M}$. Thus, we have that the natural gradient in the population limits corresponds to the update:

$$\boldsymbol{\theta}_{t+1} \leftarrow \boldsymbol{\theta}_t - \eta \, \mathrm{d}u_{|\boldsymbol{\theta}_t}^\dagger \left( \Pi_{T_{\boldsymbol{\theta}_t}\mathcal{M}}^\perp \left( \nabla \mathcal{L}_{|u_{|\boldsymbol{\theta}_t}} \right) \right) . \tag{14}$$

Note that the use of the pseudo-inverse implies that the update in the parameter space happens in the subspace $(\mathrm{Ker}\, \mathrm{d}u_{|\boldsymbol{\theta}})^\perp \subset \mathbb{R}^P$.

## 3 EMPIRICAL NATURAL GRADIENT AND ANAGRAM

In practice, one cannot reach the population limit and thus Equation (14) is only an asymptotic update. Nevertheless, we can derive a more accurate update, when we can rely only on a finite set of points $(x_i)_{i=1}^S$ that is usually called a batch. Following Jacot et al. (2018), we know that standard quadratic gradient descent update with respect to a batch in the vanishing learning rate limit $\eta \to 0$, rewrites in the functional space as:

$$\frac{\mathrm{d}u_{|\boldsymbol{\theta}_t}}{\mathrm{d}t}(x) = - \sum_{i=1}^{S} NTK_{\boldsymbol{\theta}_t}(x, x_i) \left( u_{|\boldsymbol{\theta}_t}(x_i) - y_i \right), \quad NTK_{\boldsymbol{\theta}}(x, y) := \sum_{p=1}^{P} \left( \partial_p u_{|\boldsymbol{\theta}}(x) \right) \left( \partial_p u_{|\boldsymbol{\theta}}(y) \right)^t . \tag{15}$$

Furthermore, Rudner et al. (2019); Bai et al. (2022) show that under natural gradient descent, the **Neural Tangent Kernel (NTK)** should be replaced in Equation (15) by the **Natural NTK (NNTK)**:

$$NNTK_{\boldsymbol{\theta}}(x, y) := \sum_{1 \leqslant p,q \leqslant P} \left( \partial_p u_{|\boldsymbol{\theta}}(x) \right) G_{\boldsymbol{\theta}\, pq}^\dagger \left( \partial_q u_{|\boldsymbol{\theta}}(y) \right)^t , \quad G_{\boldsymbol{\theta}\, p,q} := \langle \partial_p u_{|\boldsymbol{\theta}} , \partial_q u_{|\boldsymbol{\theta}} \rangle_{\mathcal{H}} . \tag{16}$$

As a consequence, one may see that the update under natural gradient descent with respect to a batch $(x_i)_{i=1}^S$ happens in a subspace of the tangent space, namely the **empirical Tangent Space**:

$$\widehat{T}_{\boldsymbol{\theta},(x_i)}^{NNTK}\mathcal{M} := \mathrm{Span}(NNTK_{\boldsymbol{\theta}}(\cdot,x_i) \,:\, (x_i)_{1\leqslant i\leqslant S}) \subset T_{\boldsymbol{\theta}}\mathcal{M}. \tag{17}$$

Subsequently, Equation (14) can then be adapted to define the **empirical Natural Gradient update**:

$$\boldsymbol{\theta}_{t+1} \leftarrow \boldsymbol{\theta}_t - \eta\, du_{|\boldsymbol{\theta}_t}^{\dagger}\left(\Pi_{\widehat{T}_{\boldsymbol{\theta},(x_i)}^{NNTK}\mathcal{M}}^{\perp}\left(\nabla\mathcal{L}_{|u_{|\boldsymbol{\theta}_t}}\right)\right). \tag{18}$$

Note that this update can be understood from the functional perspective as the standard Nyström method (Sun et al., 2015), bridging the gap between our work and the many methods developed in this field. Nevertheless, the $NNTK_{\boldsymbol{\theta}}$ kernel cannot be computed explicitly in our case, since it requires *a priori* inverting the Gram matrix, which adds further challenge. With this in mind, we present a first result, encapsulated in the following theorem, which is one of our main contributions:

**Theorem 1** (ANaGRAM). *Let us define for all $1 \leqslant i \leqslant S$ and for all $1 \leqslant p \leqslant P$:*

$$\widehat{\phi}_{\boldsymbol{\theta}\,i,p} := \partial_p u_{|\boldsymbol{\theta}}(x_i)\,; \qquad \widehat{\nabla\mathcal{L}}_{|u_{|\boldsymbol{\theta}}\,i} := \nabla\mathcal{L}_{|u_{|\boldsymbol{\theta}}}(x_i) = u_{|\boldsymbol{\theta}}(x_i) - f(x_i).$$

*Then:* $\qquad du_{|\boldsymbol{\theta}}^{\dagger}\left(\Pi_{\widehat{T}_{\boldsymbol{\theta},(x_i)}^{NNTK}\mathcal{M}}^{\perp}\nabla\mathcal{L}_{|u_{|\boldsymbol{\theta}}}\right) = \left(\widehat{\phi}_{\boldsymbol{\theta}}^{\dagger} + E_{\boldsymbol{\theta}}^{metric}\right)\left(\widehat{\nabla\mathcal{L}}_{|u_{|\boldsymbol{\theta}}} + E_{\boldsymbol{\theta}}^{\perp}\right),$ (19)

*where $E_{\boldsymbol{\theta}}^{metric}$ and $E_{\boldsymbol{\theta}}^{\perp}$ are correction terms specified in Equations (57) and (58) in Section C.3, respectively accounting for the metric's impact on empirical tangent space definition, and the substraction of the evaluation of the orthogonal part[3] of the functional gradient.*

A proof of this theorem, as well as a more comprehensive introduction to empirical natural gradient, encompassing a *détour* through RKHS theory, can be found in Section C.
*Remark* 1. In some important cases the correction terms $E_{\boldsymbol{\theta}}^{metric}$ and $E_{\boldsymbol{\theta}}^{\perp}$ vanishes. This happens for instance for $E_{\boldsymbol{\theta}}^{\perp}$ when solving $D[u] = 0$ with $D$ linear and $u$ an MLP (see Section B.2). We refer to Proposition 2 and Remark 6 at the end of Section C.3. $E_{\boldsymbol{\theta}}^{metric}$ cancels out in the following case:

**Proposition 1.** *There exist $P$ points $(\hat{x}_i)$ such that $\widehat{T}_{\boldsymbol{\theta},(x_i)}^{NNTK}\mathcal{M} = T_{\boldsymbol{\theta}}\mathcal{M}$. Then notably $E_{\boldsymbol{\theta}}^{metric} = 0$.*

As a first approximation, we can neglect those two terms, yielding the following vanilla algorithm:

---
**Algorithm 1:** vanilla ANaGRAM
***

**Input:** • $u : \mathbb{R}^P \to \mathrm{L}^2(\Omega,\mu)$ // neural network architecture

  • $\boldsymbol{\theta}_0 \in \mathbb{R}^P$ // initialization of the neural network

  • $f \in \mathrm{L}^2(\Omega,\mu)$ // target function of the quadratic regression

  • $(x_i) \in \Omega^S$ // a batch in $\Omega$

  • $\epsilon > 0$ // cutoff level to compute the pseudo inverse

1 **repeat**

2 $\quad \widehat{\phi}_{\boldsymbol{\theta}_t} \leftarrow \left(\partial_p u_{|\boldsymbol{\theta}_t}(x_i)\right)_{1\leqslant i\leqslant S,\, 1\leqslant p\leqslant P}$ // Computed *via* auto-differentiation

3 $\quad \widehat{U}_{\boldsymbol{\theta}_t}, \widehat{\Delta}_{\boldsymbol{\theta}_t}, \widehat{V}_{\boldsymbol{\theta}_t}^t \leftarrow SVD\left(\widehat{\phi}_{\boldsymbol{\theta}_t}\right)$

4 $\quad \widehat{\Delta}_{\boldsymbol{\theta}_t} \leftarrow \left(\widehat{\Delta}_{\boldsymbol{\theta}_t\,p} \text{ if } \widehat{\Delta}_{\boldsymbol{\theta}_t\,p} > \epsilon \text{ else } 0\right)_{1\leqslant p\leqslant P}$

5 $\quad \widehat{\nabla\mathcal{L}} \leftarrow \left(u_{|\boldsymbol{\theta}_t}(x_i) - f(x_i)\right)_{1\leqslant i\leqslant S}$

6 $\quad d_{\boldsymbol{\theta}_t} \leftarrow \widehat{V}_{\boldsymbol{\theta}_t}\widehat{\Delta}_{\boldsymbol{\theta}_t}^{\dagger}\widehat{U}_{\boldsymbol{\theta}_t}^t\widehat{\nabla\mathcal{L}}$

7 $\quad \eta_t \leftarrow \underset{\eta\in\mathbb{R}^+}{\arg\min}\sum_{1\leqslant i\leqslant S}\left(f(x_i) - u_{|\boldsymbol{\theta}_t - \eta d_{\boldsymbol{\theta}_t}}(x_i)\right)^2$ // Using *e.g.* line search

8 $\quad \boldsymbol{\theta}_{t+1} \leftarrow \boldsymbol{\theta}_t - \eta_t\, d_{\boldsymbol{\theta}_t}$

9 **until** *stop criterion met*

---

Note that algorithm 1 is equivalent to Gauss-Newton algorithm (Cai et al., 2019) applied to the empirical loss in Equation (11) also considered recently in Jnini et al. (2024) with a different setting.

---
[3]orthogonal to the whole tangent space $T_{\boldsymbol{\theta}}\mathcal{M}$.

Nevertheless, our work aims at a more general approach, giving rise to different algorithms depending on the approximations of $E_{\boldsymbol{\theta}}^{\mathrm{metric}}$ and $E_{\boldsymbol{\theta}}^{\perp}$. One of the pleasant byproducts of the ANaGRAM framework is also that it leads to a straightforward criterion to choose points in the batch, namely:

$$(x_i^*) := \underset{(x_i)\in\Omega^S}{\arg\min} \|\Pi_{\mathrm{Span}(NNTK_{\boldsymbol{\theta}}(x_i,\cdot):1\leqslant i\leqslant S)}^{\perp}(\nabla\mathcal{L}) - \nabla\mathcal{L}\|_{\mathcal{H}}, \tag{20}$$

which is amenable to various approximations, subject to further investigations. Taking the best advantage of this criterion should eventually allow us to use natural gradient in a mini-batch setting while staying close to the convergence rate of the full batch natural gradient as characterized in Xu et al. (2024). We will now show how ANaGRAM can be applied to the PINNs framework.

## 4 ANaGRAM FOR PINNs

Generalizing ANaGRAM to PINNs only requires to change the problem perspective.

### 4.1 PINNs AS A LEAST-SQUARES REGRESSION PROBLEM

The only difference between the losses of Equation (7) and Equation (11) is the use of the differential operator $D$ and the boundary operator $B$ in Equation (7). More precisely, PINNs and standard quadratic regression problems are essentially similar, except that in the case of PINNs we use the compound model $(D, B) \circ u$ instead of $u$ directly, where, using the definitions of Equation (5):

$$(D, B) \circ u : \begin{cases} \mathbb{R}^P & \to & \mathcal{H} & \to & \mathbf{L^2}(\boldsymbol{\Omega}, \partial\boldsymbol{\Omega}) := L^2(\Omega \to \mathbb{R}, \mu) \times L^2(\partial\Omega \to \mathbb{R}, \sigma) \\ \boldsymbol{\theta} & \mapsto & u_{|\boldsymbol{\theta}} & \mapsto & (D[u_{|\boldsymbol{\theta}}], B[u_{|\boldsymbol{\theta}}]) \end{cases}. \tag{21}$$

The derivation of vanilla ANaGRAM in PINNs context is then straightforward:

---
**Algorithm 2:** vanilla ANaGRAM for PINNs
---

**Input:**• $u : \mathbb{R}^P \to \mathcal{H}$ // neural network architecture

- $\boldsymbol{\theta}_0 \in \mathbb{R}^P$ // initialization of the neural network
- $D : \mathcal{H} \to L^2(\Omega \to \mathbb{R}, \mu)$ // differential operator
- $B : \mathcal{H} \to L^2(\partial\Omega \to \mathbb{R}, \sigma)$ // boundary operator
- $f \in L^2(\Omega \to \mathbb{R}, \mu)$ // source term
- $g \in L^2(\partial\Omega \to \mathbb{R}, \sigma)$ // boundary value
- $(x_i^D) \in \Omega^{S_D}$ // a batch in $\Omega$
- $(x_i^B) \in \Omega^{S_B}$ // a batch in $\partial\Omega$
- $\epsilon > 0$ // cutoff level to compute the pseudo inverse

1 **repeat**

2 $\quad \hat{\phi}_{\boldsymbol{\theta}_t} \leftarrow \left( \left( \partial_p D[u_{|\boldsymbol{\theta}_t}](x_i^D) \right)_{i=1}^{S_D}, \quad \left( \partial_p B[u_{|\boldsymbol{\theta}_t}](x_i^B) \right)_{i=1}^{S_B} \right)_{p=1}^P$ // via autodiff

3 $\quad \hat{V}_{\boldsymbol{\theta}_t}, \hat{\Delta}_{\boldsymbol{\theta}_t}, \hat{U}_{\boldsymbol{\theta}_t}^t \leftarrow SVD(\hat{\phi}_{\boldsymbol{\theta}_t})$

4 $\quad \hat{\Delta}_{\boldsymbol{\theta}_t} \leftarrow \left( \hat{\Delta}_{\boldsymbol{\theta}_t\,r} \text{ if } \hat{\Delta}_{\boldsymbol{\theta}_t\,r} > \epsilon \text{ else } 0 \right)_{1\leqslant r\leqslant P}$

5 $\quad \widehat{\nabla\mathcal{L}} \leftarrow \begin{pmatrix} \left( D[u_{|\boldsymbol{\theta}_t}](x_i^D) - f(x_i^D) \right)_{1\leqslant i\leqslant S_D} \\ \left( B[u_{|\boldsymbol{\theta}_t}](x_i^B) - g(x_i^B) \right)_{1\leqslant i\leqslant S_B} \end{pmatrix}$

6 $\quad d_{\boldsymbol{\theta}_t} \leftarrow \hat{V}_{\boldsymbol{\theta}_t} \hat{\Delta}_{\boldsymbol{\theta}_t}^{\dagger} \hat{U}_{\boldsymbol{\theta}_t}^t \widehat{\nabla\mathcal{L}}$

7 $\quad \eta_t \leftarrow \underset{\eta\in\mathbb{R}^+}{\arg\min} \frac{1}{2S_D} \underset{1\leqslant i\leqslant S_D}{\sum} \left( f(x_i^D) - D[u_{|\boldsymbol{\theta}_t-\eta d_{\boldsymbol{\theta}_t}}](x_i^D) \right)^2 +$

$\quad\quad \frac{1}{2S_B} \underset{1\leqslant i\leqslant S_B}{\sum} \left( g(x_i^B) - B[u_{|\boldsymbol{\theta}_t-\eta d_{\boldsymbol{\theta}_t}}](x_i^B) \right)^2$ // Using e.g. line search

8 $\quad \boldsymbol{\theta}_{t+1} \leftarrow \boldsymbol{\theta}_t - \eta_t d_{\boldsymbol{\theta}_t}$

9 **until** *stop criterion met*

---

Adaptation of the definitions in Sections 2.3 and 3 to the case of PINNs are detailed in Sections C.4 and C.5 respectively. We now present the link between PINN's natural gradient and Green's function.

## 4.2 PINNs Natural Gradient is a Green's Function

Knowing the Green's function of a linear operator is one of the most optimal ways of solving the associated PDE, since it then suffices to estimate an integral to approximate a solution (Duffy, 2015). However, this requires prior knowledge of the Green's function, which is not always possible. Here, we show that using the natural gradient for PINNs implicitly uses the operator's Green's function. In Section D, we briefly recall the main definitions required to state and prove the following theorem:

**Theorem 2.** *Let $D : \mathcal{H} \to L^2(\Omega \to \mathbb{R}, \mu)$ be a linear differential operator and $u : \mathbb{R}^P \to \mathcal{H}$ a parametric model. Then for all $\boldsymbol{\theta} \in \mathbb{R}^P$, the generalized Green's function of $D$ on $T_{\boldsymbol{\theta}}\mathcal{M} = \operatorname{Im} du_{|\boldsymbol{\theta}}$ is given by: for all $x, y \in \Omega$*

$$g_{T_{\boldsymbol{\theta}}\mathcal{M}}(x, y) := \sum_{1 \leqslant p, q \leqslant P} \partial_p u_{|\boldsymbol{\theta}}(x) \, G_{p,q}^{\dagger} \partial_q D[u_{|\boldsymbol{\theta}}](y), \tag{22}$$

*with: for all $1 \leqslant p, q \leqslant P$*

$$G_{pq} := \left\langle \partial_p D[u_{|\boldsymbol{\theta}}] , \, \partial_q D[u_{|\boldsymbol{\theta}}] \right\rangle_{L^2(\Omega \to \mathbb{R}, \mu)}. \tag{23}$$

*In particular, the natural gradient of PINNs[4] can be rewritten:*

$$\boldsymbol{\theta}_{t+1} \leftarrow \boldsymbol{\theta}_t - \eta \, du_{|\boldsymbol{\theta}_t}^{\dagger} \left( x \in \Omega \mapsto \int_{\Omega} g_{T_{\boldsymbol{\theta}_t}\mathcal{M}}(x, y) \nabla \mathcal{L}_{|\boldsymbol{\theta}_t}(y) \mu(dy) \right), \tag{24}$$

A few comments should be made about Equation (24). First, if $\eta = 1$, then the natural gradient can be understood as the least-square's solution of $D[u] = f$ at order 1, *i.e.* in the affine space $u_{|\boldsymbol{\theta}_t} + T_{\boldsymbol{\theta}_t}\mathcal{M}$. However, it does not hold *a priori* that:

- $D[u_{|\boldsymbol{\theta}_t} + T_{\boldsymbol{\theta}_t}\mathcal{M}]$ correctly approximates $f \in L^2(\Omega \to \mathbb{R}, \mu)$.

- $u_{|\boldsymbol{\theta}_t} + T_{\boldsymbol{\theta}_t}\mathcal{M}$ correctly approximates the image space $\mathcal{M} = \{u_{|\boldsymbol{\theta}} : \boldsymbol{\theta} \in \mathbb{R}^P\}$.

Multiplying by a learning rate $\eta \ll 1$ is then essential. In this way, natural gradient can be understood as moving in the direction of the solution of $D[u] = f$ in the affine space $u_{|\boldsymbol{\theta}_t} + T_{\boldsymbol{\theta}_t}\mathcal{M}$, and thus getting closer to the solution, while expecting that the change induced by this update will improve the approximation space $u_{|\boldsymbol{\theta}_{t+1}} + T_{\boldsymbol{\theta}_{t+1}}\mathcal{M}$[5]. On the other hand, when we approach the end of the optimization, *i.e.* when the space $D[u_{|\boldsymbol{\theta}_t} + T_{\boldsymbol{\theta}_t}\mathcal{M}]$ approximates $f$ "well enough", while $du_{|\boldsymbol{\theta}_t}$ approximates "well enough" $\mathcal{M}$, then it is in our best interest to solve the equation completely, *i.e.* to take learning rates $\eta$ close to 1. This is why the use of line search in ANaGRAM (*cf.* line 6 in Algorithm 2) is essential. We should then conclude that the quality of the solution found by the parametric model $u$ **depends only** on:

- How well $\Gamma = \{D[u_{|\boldsymbol{\theta}}] : \boldsymbol{\theta} \in \mathbb{R}^P\}$ can approximate the source $f \in L^2(\Omega \to \mathbb{R}, \mu)$.

- The curvature of $\Gamma$. More precisely, if its non-linear structure prevents convergence to a $D[u_{|\boldsymbol{\theta}}]$ such that $f - D[u_{|\boldsymbol{\theta}}]$ is non-negligible, while being orthogonal to the tangent space $D[T_{\boldsymbol{\theta}_t}\mathcal{M}]$.

If we assume now that $D$ is also nonlinear, then all the above analysis also holds for the linear operators $dD_{|u_{|\boldsymbol{\theta}_t}}$, the difference being that the operator changes at each step. This means that in the case of non-linear operators, we have to deal with both the non-linearity of $D$ and $u$, but that does not change the overall dynamic.

Finally, assuming that both $D$ and $u$ are linear (this is for instance the case when we assume $u$ to be a linear combination of basis functions, like in Finite Elements, or Fourier Series). Then "learning" $u_{|\boldsymbol{\theta}}$ with natural gradient (and learning rate 1) corresponds to solve the equation in the least-squares sense with a generalized Green's function.

---

[4]*cf.* last line of Table 17 in Section C.4.

[5]To our best knowledge, rigorous proof of this phenomenon has not yet been provided. We can therefore only rely on empirical evidence, which is illustrated in Section 5.

## 5 EXPERIMENTS

We test ANaGRAM on four problems: 2 D Laplace equation ; 1+1 D heat equation ; 5 D Laplace equation ; and 1+1 D Allen-Cahn equation. The first three problems comes from Müller and Zeinhofer (2023), while the last one is proposed in Lu et al. (2021).

For training, we use multilayer perceptrons with varying layer sizes and $\tanh$ activations, along with fixed batches of points: a batch of size $S_D$ to discretize $\Omega$ and a batch of size $S_B$ to discretize $\partial\Omega$. The layer size specifications, cutoff factor $\epsilon$, values of $S_D$ and $S_B$, and discretization procedures are specified separately for each problem. Currently, the cutoff factor is chosen manually and warrants further investigation.

For these various problems, we display as a function of gradient descent steps, the medians over 10 different initializations, of $L^2$ error $E_{L^2}$ and test loss $E_{\text{test}}$, with shaded area indicating the range between the first and third quartiles. $E_{L^2}$ is defined as: given test points $(x_i)_{i=1}^S$, $E_{L^2}(\boldsymbol{\theta}) := \sqrt{\frac{1}{S_{L^2}} \sum_{i=1}^{S_{L^2}} \left| u_{|\boldsymbol{\theta}}(x_i) - u^*(x_i) \right|^2}$, where $u^*$ is a known solution to the PDE and $S$ is taken 10 times bigger than the $\Omega$ batch size $S_D$, while $E_{\text{test}}$ is the empirical PINNs loss $\ell$ of Equation (7), computed with a distinct set of points, of size 5 times bigger than the $\Omega$ batch size $S_D$. We compare ANaGRAM to Energy Natural Gradient descent (E-NGD) (Müller and Zeinhofer, 2023), vanilla gradient descent (GD) with line-search, Adam (Kingma and Ba, 2014) with exponentially decaying learning-rate after $10^{15}$ steps as in Müller and Zeinhofer (2023) as well as L-BFGS (Liu and Nocedal, 1989). The corresponding CPU times are also provided in tables for reference. The code is made avaible at `https://github.com/IloneM/ANaGRAM/` and further implementation and computation details are provided in Section A.1.

**2 D Laplace equation** : We consider the two dimensional Laplace equation and its solution:

$$\begin{cases} \Delta u = -2\pi^2 \sin(\pi x_1)\sin(\pi x_2) & \text{in } \Omega = [0,1]^2 \\ u = 0 & \text{on } \partial\Omega \end{cases} ; \quad u^*(x_1, x_2) = \sin(\pi x_1)\sin(\pi x_2). \quad (25)$$

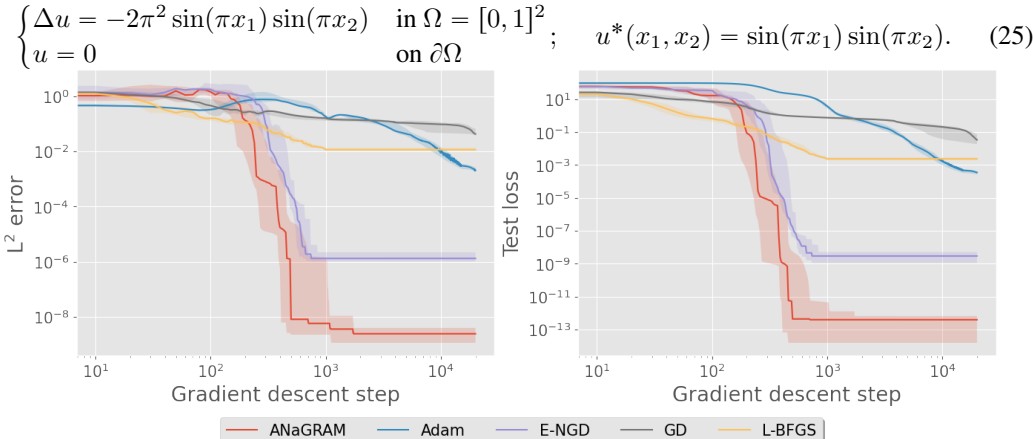

Figure 1: Median absolute $L^2$ errors and Test losses for the 2 D Laplace equation.

We choose $S_D = 900$ equi-distantly spaced points in the interior of $\Omega$ and $S_B = 120$ equally spaced points on the boundary $\partial\Omega$ (30 on each side). ANaGRAM, E-NGD and L-BFGS are applied for 2000 iterations each, while GD and Adam are trained for $20 \times 10^3$ iterations. The network consists of a single hidden layer with a width of 32, resulting in a total of $P = 129$ parameters. The cutoff factor is set to $\epsilon = 1 \times 10^{-6}$.

| CPU time (s) | Per step | Full |
|---|---|---|
| ANaGRAM | 7.16e-02 | **1.25e+02** |
| Adam | **1.23e-02** | 2.44e+02 |
| E-NGD | 1.94e-01 | 1.88e+02 |
| GD | 2.07e-02 | 4.13e+02 |
| L-BFGS | 1.95e-01 | 1.95e+02 |

**1+1 D Heat equation** : We consider the $(1+1)$ dimensional Heat equation and its solution:

$$\begin{cases} \partial_t u - \frac{1}{4}\partial_{xx} u = 0 & \text{in } \Omega = [0,1]^2 \\ u = 0 & \text{on } \partial\Omega_{\text{border}} = [0,1] \times \{0,1\} \\ u(0,x) = \sin(\pi x) & \text{on } \partial\Omega_0 = \{0\} \times [0,1] \end{cases} ; \quad u^*(t,x) = \exp\left(-\frac{\pi^2 t}{4}\right)\sin(\pi x). \quad (26)$$

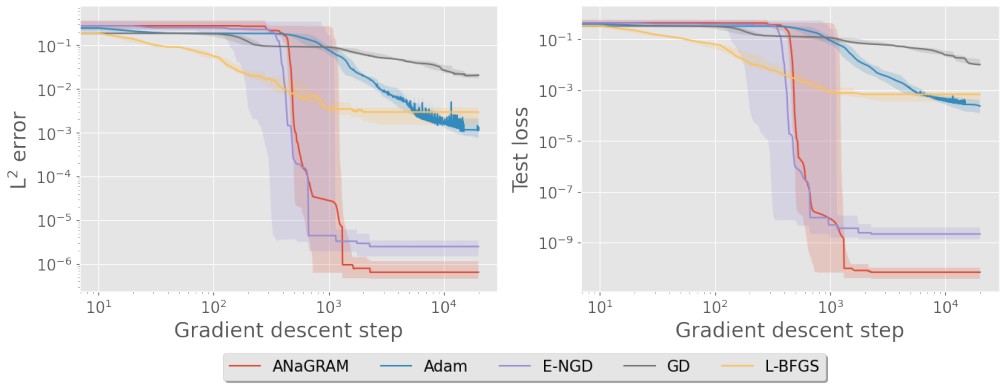

Figure 2: Median absolute $L^2$ errors and Test losses for the Heat equation.

We choose $S_D = 900$ equi-distantly spaced points in the interior of $\Omega$ and $S_B = 90$ equally spaced points on the boundary $\partial\Omega$ (30 on $\partial\Omega_0$ and 30 on each side of $\partial\Omega_{\text{border}}$). ANaGRAM, E-NGD and L-BFGS are applied for 2000 iterations each, while GD and Adam are trained for $20 \times 10^3$ iterations. The network consists of a single hidden layer with a width of 64, resulting in a total of $P = 257$ parameters. The cutoff factor is set to $\epsilon = 1 \times 10^{-5}$.

| CPU time (s) | Per step | Full |
|---|---|---|
| ANaGRAM | 1.29e-01 | **3.78e+02** |
| Adam | **2.12e-02** | 4.15e+02 |
| E-NGD | 1.78e-01 | 4.04e+02 |
| GD | 3.87e-02 | 7.68e+02 |
| L-BFGS | 1.30e-01 | 3.91e+02 |

**5 D Laplace equation** : We consider the five dimensional Laplace equation and its solution:

$$\begin{cases} \Delta u = \pi^2 \sum_{k=1}^5 \sin(\pi x_k) & \text{in } \Omega = [0,1]^5 \\ u = \sum_{k=1}^5 \sin(\pi x_k) & \text{on } \partial\Omega \end{cases} ; \qquad u^*(x) = \sum_{k=1}^5 \sin(\pi x_k), \qquad (27)$$

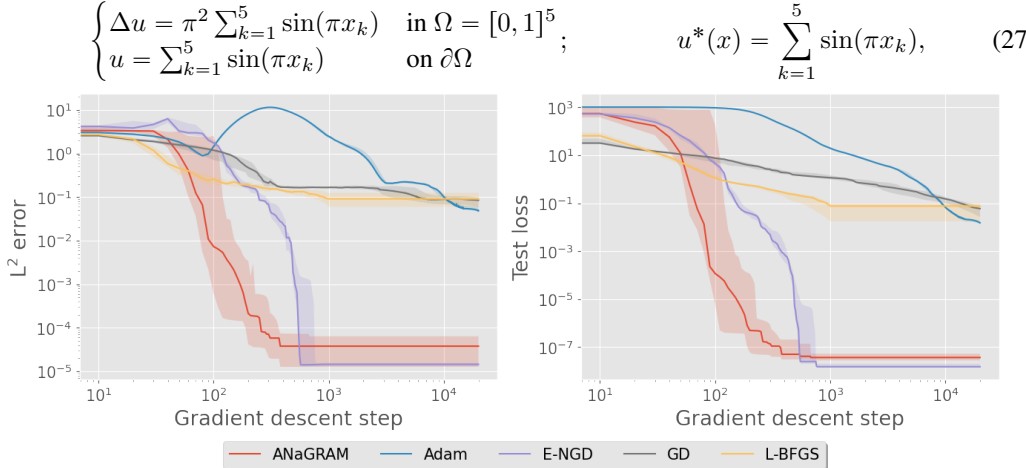

Figure 3: Median absolute $L^2$ errors and Test losses for the 5 D Laplace equation.

We choose $S_D = 4000$ uniformly drawn points in the interior of $\Omega$ and $S_B = 500$ uniformly drawn points on the boundary $\partial\Omega$. ANaGRAM, E-NGD and L-BFGS are applied for 1000 iterations each, while GD and Adam are trained for $20 \times 10^3$ iterations. The network consists of a single hidden layer with a width of 64, resulting in a total of $P = 449$ parameters. The cutoff factor is set to $\epsilon = 5.10^{-7} \times \Delta_{\boldsymbol{\theta}\max}$, where $\Delta_{\boldsymbol{\theta}\max}$ is the maximal eigenvalue of $\widehat{\phi}_{\boldsymbol{\theta}}$ (cf. line 1 of algorithm 2).

| CPU time (s) | Per step | Full |
|---|---|---|
| ANaGRAM | 7.18e-01 | 4.88e+02 |
| Adam | **6.65e-02** | 1.29e+03 |
| E-NGD | 6.52e+00 | 4.96e+03 |
| GD | 2.69e-01 | 5.38e+03 |
| L-BFGS | 2.96e-01 | **2.96e+02** |

**1+1 D Allen-Cahn equation** We consider the $(1+1)$ dimensional Allen-Cahn equation:

$$\begin{cases} \partial_t u - 10^{-3}\,\partial_{xx}u - 5(u - u^3) = 0 & \text{in } \Omega = [0,1] \times [-1,1] \\ u = -1 & \text{on } \partial\Omega_{\text{border}} = [0,1] \times \{-1,1\} \\ u(0,x) = x^2\cos(\pi x) & \text{on } \partial\Omega_0 = \{0\} \times [-1,1] \end{cases} \qquad (28)$$

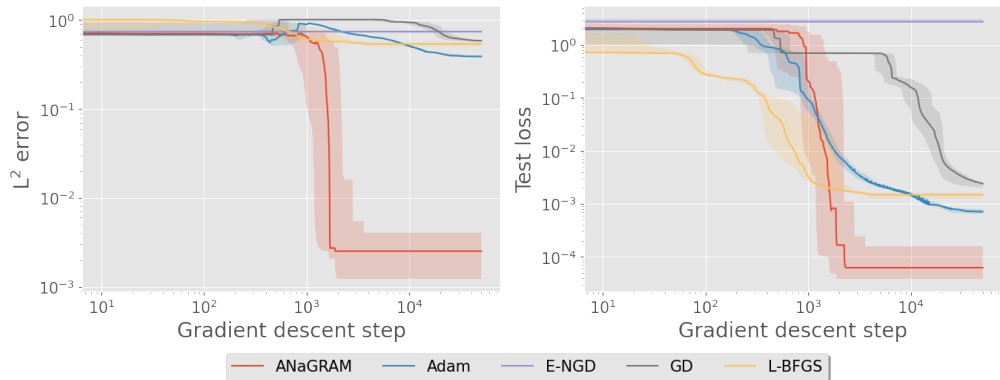

Figure 4: Median absolute $L^2$ errors and Test losses for the Allen-Cahn equation.

We choose $S_D = 900$ equi-distantly spaced points in the interior of $\Omega$ and $S_B = 90$ equally spaced points on the boundary $\partial\Omega$ (30 on $\partial\Omega_0$ and 30 on each side of $\partial\Omega_{\text{border}}$). ANaGRAM and L-BFGS are applied for 4000 iterations each, E-NGD for 1000 iterations, while classical gradient descent (GD) and Adam are trained for $50 \times 10^3$ iterations. The network consists of three hidden layers with a width of 20, resulting in a total of $P = 921$ parameters. The cutoff factor is set to $\epsilon = 5.10^{-7} \times \Delta_{\boldsymbol{\theta}\max}$, where $\Delta_{\boldsymbol{\theta}\max}$ is the maximal eigenvalue of $\widehat{\phi}_{\boldsymbol{\theta}}$ (cf. line 1 of algorithm 2).

| CPU time (s) | Per step | Full |
|---|---|---|
| ANaGRAM | 6.01e-01 | 2.16e+03 |
| Adam | **2.82e-02** | **1.18e+03** |
| E-NGD | 1.30e+00 | 6.52e+03 |
| GD | 8.59e-02 | 4.28e+03 |
| L-BFGS | 4.07e-01 | 1.60e+03 |

**Results summary** : We demonstrated that our approach can achieve comparable accuracy to Müller and Zeinhofer (2023) on linear problems, consistent with the equivalence established in Section E, while maintaining a per-step computational cost at most reasonably higher than that of Adam. Excluding Adam and GD, which consistently get stuck at high error levels, the bottom line is that ANaGRAM consistently outperforms both E-NGD and L-BFGS—often by a significant margin—on at least one or even both criteria: precision and computation time. The cases where the computation times of E-NGD and ANaGRAM are similar occur when small-sized architectures are sufficient for the problem.

## 6 Conclusion and Perspectives

We introduce empirical Natural Gradient, a new kind of natural gradient that scales linearly with respect to the number of parameters and extend it to PINNs framework through a mathematically principled reformulation. We show that this update implicitly corresponds to the use of the Green's function of the operator. We give empirical evidences that this optimization in its simplest form (vanilla ANaGRAM) already achieves highly accurate solutions, comparable to Müller and Zeinhofer (2023) for linear PDEs at a fraction of the computational cost, and with significant improvements for non-linear equations, for which equivalence of the two algorithms does not hold anymore.

Still, the present formulation of the algorithm has two limitations: one concerns the chosing procedure of the batch points, which is so far limited to simple heuristics; the second is the hyperparameter tunning, more specifically the cutoff factor, which is so far chosen by hand, while it may probably be automatically chosen based on the spectrum of $\widehat{\phi}_{\boldsymbol{\theta}}$.

Important perspectives include exploring approximations schemes for terms $E_{\boldsymbol{\theta}}^{\text{metric}}$ (e.g. using Nyström's methods, cf. Sun et al. (2015)) and $E_{\boldsymbol{\theta}}^{\perp}$ (e.g. using Cohen and Migliorati (2017)), introduced in Theorem 1, the design of an optimal collocation points procedure, coupled with SVD cut-off factor adaptation strategy for ANaGRAM, as well as incorporation of common optimization techniques, such as momentum. From a theoretical point of view, it seems particularly important to us to include data assimilation in this theoretical setting, and understand its regularizing effect, while establishing connections to classical solvers such as FEMs.

ACKNOWLEDGMENTS

The work has been supported by the french ANR grant Scalp (ANR-24-CE23-1320).

We wish to thank Matthieu Dolbeault for insightful discussions on sampling methods for $L^2$-approximations, which eventually led to the concept of empirical tangent space.

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

## A COMPLEMENTARY MATERIAL FOR THE EXPERIMENTS

### A.1 DESCRIPTION OF THE EXPERIMENTAL SETTING

**Description of the Method** We base our code on Müller and Zeinhofer (2023). For our 4 numerical experiments, we apply ANaGRAM gradient steps as defined in the Algorithm 2. As in Müller and Zeinhofer (2023), we choose the interval $[0, 1]$ for the line search determining the learning rate, 1 corresponding to solving the (linearized PDE) with the Green's function (*cf.* Section 4.2). The neural network weights are initialized using the Glorot normal initialization (Glorot and Bengio, 2010).

**Computation Details** As in Müller and Zeinhofer (2023), our implementation relies on JAX (Bradbury et al., 2018), where all derivatives are computed using JAX automatic differentiation, and the singular value decomposition computation is carried out by the scipy (Virtanen et al., 2020) implementation of JAX. Stochastic gradient descent, Adam, as well as L-BFGS relies on the implementation of DeepMind et al. (2020). All experiments were run on a 11th Gen Intel® Core™ i7-1185G7 @ 3.00GHz Laptop CPU in double precision (float64).

### A.2 FIGURES RELATIVE TO COMPUTATION TIME

#### A.2.1 2 D LAPLACE

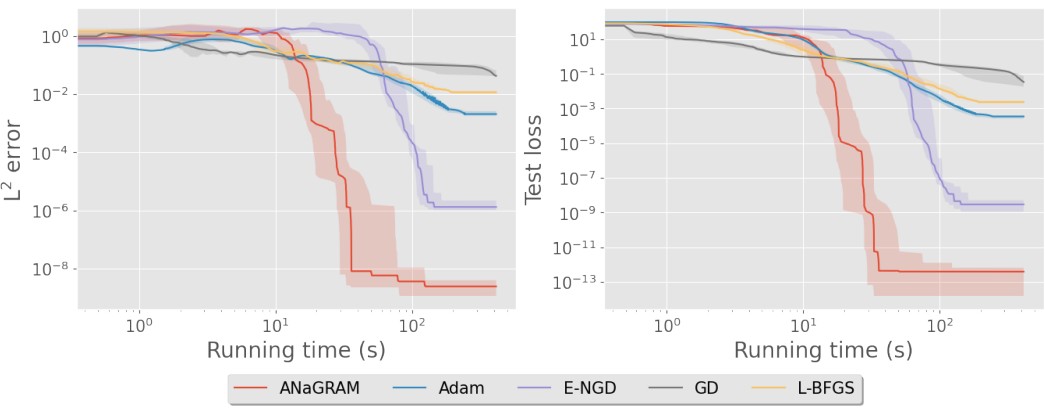

Figure 5: Median absolute $L^2$ errors and Test losses for the 2 D Laplace equation across 10 different initializations for the five optimizers, relative to computation time. The shaded area indicates the range between the first and third quartiles.

#### A.2.2 HEAT

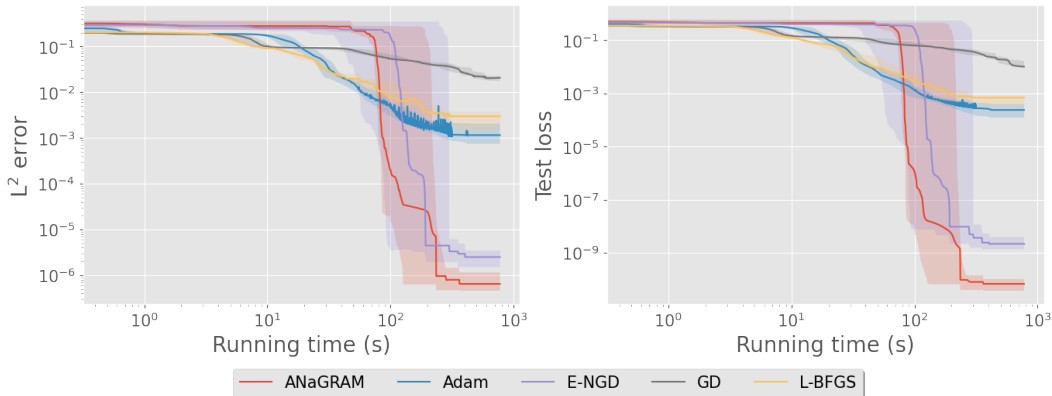

Figure 6: Median absolute $L^2$ errors and Test losses for the Heat equation across 10 different initializations for the five optimizers, relative to computation time. The shaded area indicates the range between the first and third quartiles.

### A.2.3  5 D LAPLACE

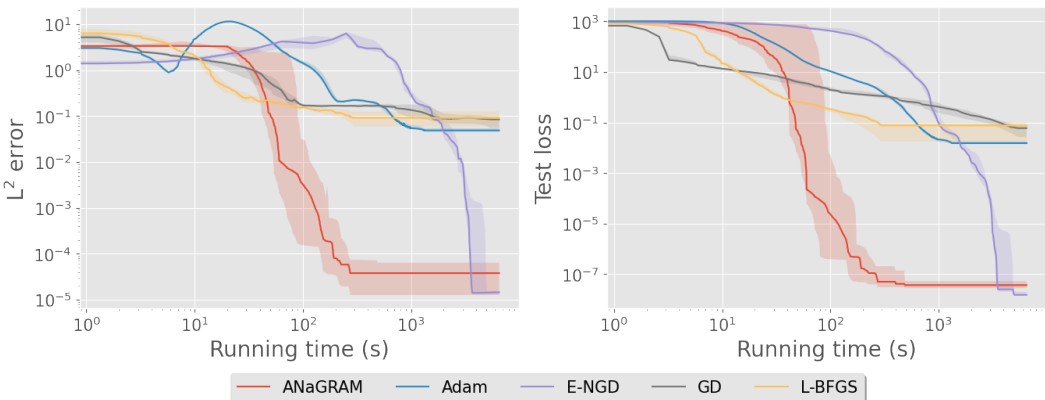

Figure 7: Median absolute $L^2$ errors and Test losses for the 5 D Laplace equation across 10 different initializations for the five optimizers, relative to computation time (except for ENGD for which we only took 3 initializations). The shaded area indicates the range between the first and third quartiles.

### A.2.4  ALLEN-CAHN

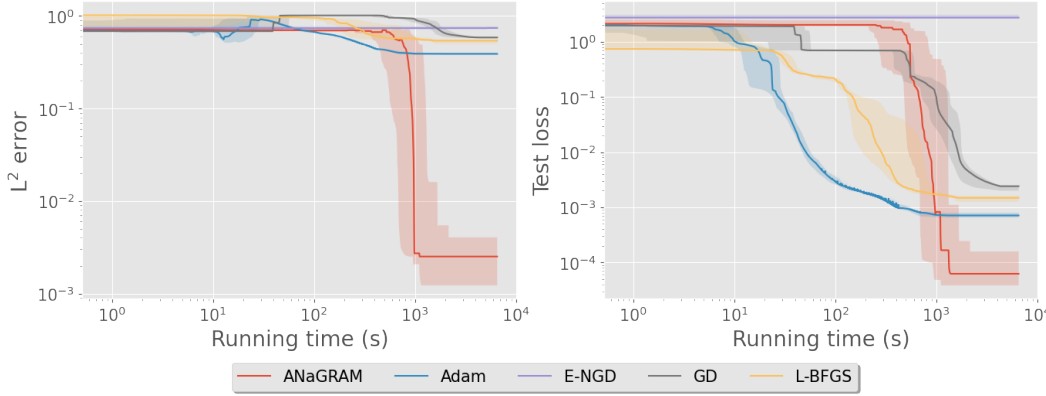

Figure 8: Median absolute $L^2$ errors and Test losses for the Allen-Cahn equation across 10 different initializations for the five optimizers, relative to computation time (except for ENGD for which we only took 3 initializations). The shaded area indicates the range between the first and third quartiles.

### A.3  STATISTICAL TABLES OF RESULTS

### A.3.1  2 D LAPLACE

Table 1: Median, Maximum and Minimum $L^2$-errors of the optimizers for the 2 D Laplace equation.

|  | Median | Minimum | Maximum |
|---|---|---|---|
| ANaGRAM | **2.42e-09** | **1.70e-10** | **1.19e-08** |
| Adam | 2.05e-03 | 1.67e-03 | 2.86e-03 |
| E-NGD | 1.31e-06 | 8.43e-07 | 3.87e-05 |
| GD | 4.25e-02 | 1.01e-02 | 1.25e-01 |
| L-BFGS | 1.15e-02 | 3.08e-03 | 1.55e-02 |

Table 2: Median, Maximum and Minimum of the test loss of the optimizers for the 2 D Laplace equation.

|  | Median | Minimum | Maximum |
|---|---|---|---|
| ANaGRAM | **3.85e-13** | **8.49e-15** | **1.43e-12** |
| Adam | 3.51e-04 | 2.29e-04 | 4.31e-04 |
| E-NGD | 2.91e-09 | 1.01e-10 | 3.57e-08 |
| GD | 3.42e-02 | 4.32e-03 | 1.76e-01 |
| L-BFGS | 2.37e-03 | 8.91e-04 | 9.09e-03 |

Table 3: Mean and Standard deviation of $L^2$-errors of the optimizers for the 2 D Laplace equation.

|  | mean | std |
|---|---|---|
| ANaGRAM | **3.49e-09** | **3.58e-09** |
| Adam | 2.19e-03 | 4.18e-04 |
| E-NGD | 5.37e-06 | 1.18e-05 |
| GD | 5.41e-02 | 1.57e-02 |
| L-BFGS | 1.13e-02 | 2.94e-03 |

Table 4: Mean and Standard deviation of of the test loss of the optimizers for the 2 D Laplace equation.

|  | mean | std |
|---|---|---|
| ANaGRAM | **4.27e-13** | **4.66e-13** |
| Adam | 3.37e-04 | 7.66e-05 |
| E-NGD | 7.00e-09 | 1.11e-08 |
| GD | 5.39e-02 | 5.39e-02 |
| L-BFGS | 3.04e-03 | 2.23e-03 |

## A.3.2 HEAT

Table 5: Median, Maximum and Minimum $L^2$-errors of the optimizers for the Heat equation.

|  | Median | Minimum | Maximum |
|---|---|---|---|
| ANaGRAM | **6.48e-07** | **3.67e-07** | **6.15e-06** |
| Adam | 1.07e-03 | 5.96e-04 | 3.94e-03 |
| E-NGD | 2.50e-06 | 1.02e-06 | 6.38e-06 |
| GD | 2.02e-02 | 1.04e-02 | 2.39e-02 |
| L-BFGS | 2.97e-03 | 5.14e-04 | 6.73e-03 |

Table 6: Median, Maximum and Minimum test loss of the optimizers for the Heat equation.

|  | Median | Minimum | Maximum |
|---|---|---|---|
| ANaGRAM | **6.82e-11** | **1.90e-11** | **2.50e-10** |
| Adam | 2.37e-04 | 1.11e-04 | 1.41e-03 |
| E-NGD | 2.18e-09 | 5.62e-10 | 1.35e-08 |
| GD | 1.01e-02 | 4.70e-03 | 1.93e-02 |
| L-BFGS | 6.84e-04 | 5.85e-05 | 3.34e-03 |

Table 7: Mean and Standard deviation of $L^2$-errors of the optimizers for the Heat equation.

|  | mean | std |
|---|---|---|
| ANaGRAM | **1.28e-06** | **1.75e-06** |
| Adam | 1.55e-03 | 5.19e-04 |
| E-NGD | 2.89e-06 | 1.77e-06 |
| GD | 1.92e-02 | 9.60e-04 |
| L-BFGS | 3.09e-03 | 1.74e-03 |

Table 8: Mean and Standard deviation of test loss of the optimizers for the Heat equation.

|  | mean | std |
|---|---|---|
| ANaGRAM | **8.56e-11** | **7.05e-11** |
| Adam | 3.63e-04 | 3.93e-04 |
| E-NGD | 3.53e-09 | 3.83e-09 |
| GD | 1.20e-02 | 1.05e-03 |
| L-BFGS | 8.54e-04 | 9.16e-04 |

### A.3.3   5 D LAPLACE

Table 9: Median, Maximum and Minimum $L^2$-errors of the optimizers for the 5 D Laplace equation.

|  | Median | Minimum | Maximum |
|---|---|---|---|
| ANaGRAM | 3.76e-05 | **6.99e-06** | 8.23e-05 |
| Adam | 4.86e-02 | 3.41e-02 | 6.08e-02 |
| E-NGD | **1.40e-05** | 1.18e-05 | **1.64e-05** |
| GD | 8.44e-02 | 1.50e-02 | 1.28e-01 |
| L-BFGS | 9.08e-02 | 1.55e-02 | 1.71e-01 |

Table 10: Median, Maximum and Minimum test loss of the optimizers for the 5 D Laplace equation.

|  | Median | Minimum | Maximum |
|---|---|---|---|
| ANaGRAM | 3.68e-08 | **1.03e-08** | 2.20e-07 |
| Adam | 1.53e-02 | 1.02e-02 | 2.54e-02 |
| E-NGD | **1.51e-08** | 1.50e-08 | **2.51e-08** |
| GD | 6.00e-02 | 6.37e-03 | 1.11e-01 |
| L-BFGS | 7.65e-02 | 5.34e-03 | 2.25e-01 |

Table 11: Mean and Standard deviation of $L^2$-errors of the optimizers for the 5 D Laplace equation.

|  | mean | std |
|---|---|---|
| ANaGRAM | 4.00e-05 | 2.93e-05 |
| Adam | 4.83e-02 | 8.06e-03 |
| E-NGD | **1.41e-05** | **2.29e-06** |
| GD | 7.64e-02 | 1.75e-02 |
| L-BFGS | 1.00e-01 | 2.19e-02 |

Table 12: Mean and Standard deviation of test loss of the optimizers for the 5 D Laplace equation.

|  | mean | std |
|---|---|---|
| ANaGRAM | 6.37e-08 | 7.01e-08 |
| Adam | 1.63e-02 | 4.19e-03 |
| E-NGD | **1.84e-08** | **5.55e-09** |
| GD | 5.64e-02 | 3.60e-02 |
| L-BFGS | 8.20e-02 | 6.80e-02 |

### A.3.4 ALLEN-CAHN

Table 13: Median, Maximum and Minimum $L^2$-errors of the optimizers for the Allen-Cahn equation.

|  | Median | Minimum | Maximum |
|---|---|---|---|
| ANaGRAM | **2.51e-03** | **6.14e-04** | **2.04e-02** |
| Adam | 3.90e-01 | 3.78e-01 | 4.80e-01 |
| E-NGD | 7.39e-01 | 7.32e-01 | 8.10e-01 |
| GD | 5.86e-01 | 5.43e-01 | 8.37e-01 |
| L-BFGS | 5.40e-01 | 4.33e-01 | 7.45e-01 |

Table 14: Median, Maximum and Minimum test loss of the optimizers for the Allen-Cahn equation.

|  | Median | Minimum | Maximum |
|---|---|---|---|
| ANaGRAM | **6.22e-05** | **1.45e-05** | 1.38e-03 |
| Adam | 7.10e-04 | 6.29e-04 | **1.18e-03** |
| E-NGD | 2.74e+00 | 2.58e+00 | 3.43e+00 |
| GD | 2.41e-03 | 1.70e-03 | 1.93e-02 |
| L-BFGS | 1.48e-03 | 9.08e-04 | 5.09e-03 |

Table 15: Mean and Standard deviation of $L^2$-errors of the optimizers for the Allen-Cahn equation.

|  | mean | std |
|---|---|---|
| ANaGRAM | **4.32e-03** | 5.93e-03 |
| Adam | 4.02e-01 | **5.19e-04** |
| E-NGD | 7.60e-01 | 4.24e-02 |
| GD | 6.06e-01 | 5.99e-02 |
| L-BFGS | 5.49e-01 | 6.85e-02 |

Table 16: Mean and Standard deviation of test loss of the optimizers for the Allen-Cahn equation.

|  | mean | std |
|---|---|---|
| ANaGRAM | **2.19e-04** | 4.16e-04 |
| Adam | 7.81e-04 | **1.01e-04** |
| E-NGD | 2.92e+00 | 4.51e-01 |
| GD | 3.95e-03 | 5.41e-03 |
| L-BFGS | 1.77e-03 | 1.19e-03 |

## A.4 ADDITIONAL EXPERIMENT : BURGERS EQUATION

We consider the $(1 + 1)$ dimensional Burger equation:

$$\begin{cases} \partial_t u + u\,\partial_x u - \nu\partial_{xx}u = 0 & \text{in } \Omega = [0,1] \times [-1,1] \\ u = 0 & \text{on } \partial\Omega_{\text{border}} = [0,1] \times \{-1,1\} \\ u(0,x) = -\sin(\pi x) & \text{on } \partial\Omega_0 = \{0\} \times [-1,1] \end{cases} \tag{29}$$

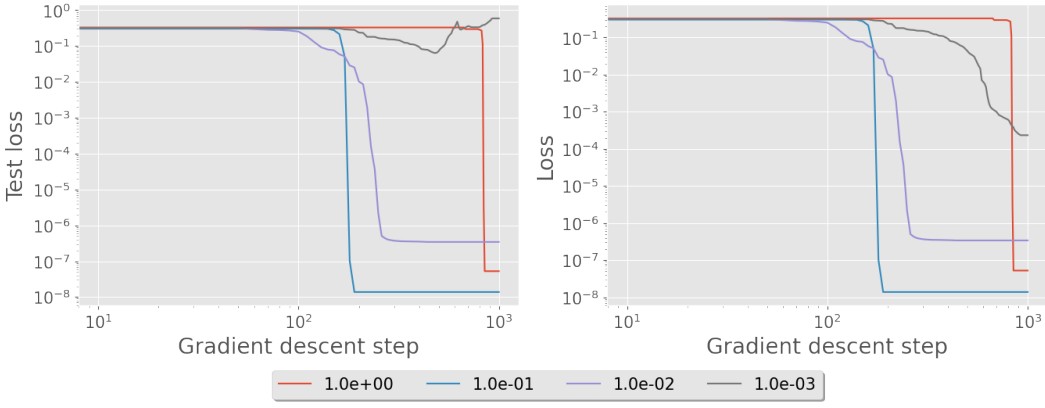

Figure 9: Test and train losses for the Burgers equation for various viscosities $\nu$.

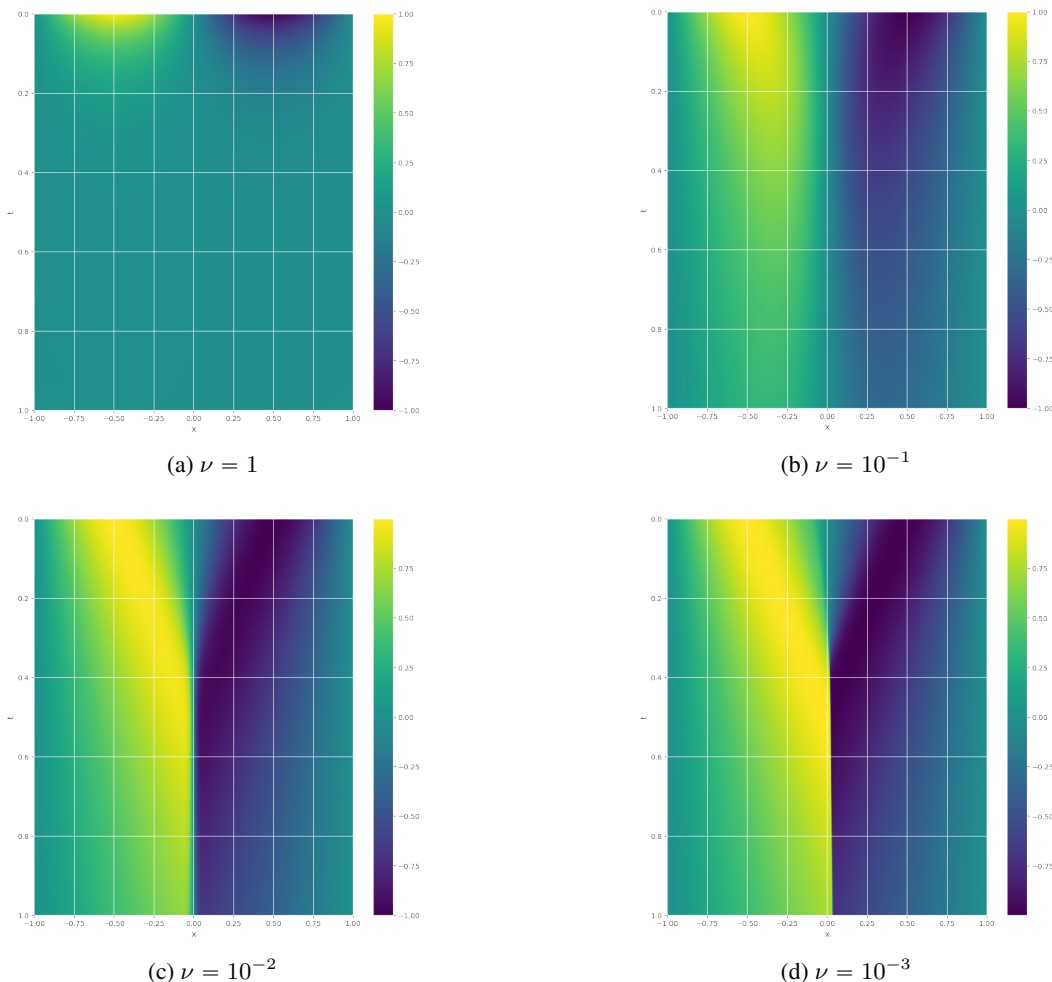

Figure 10: PINN solution profiles for the Burgers equation for various viscosities $\nu$

We considered the burger with 4 viscosities $\nu \in \{1, 10^{-1}, 10^{-2}, 10^{-3}\}$. Since a shock develops in finite time as $\nu \to 0$, using a regular grid did not yield sufficiently accurate results. Consequently, we manually adjusted the grid. Specifically, we defined two discretizations $D_t$, $D_x$, corresponding to $[0, 1]$ and $[-1, 1]$, respectively, as follows:

- $D_t$: A regular grid discretization with 240 points.
- $D_x$: A non-regular grid discretization constructed as:
  - A regular-grid of $60 = \frac{240}{4}$ points on $[-1, -0.25)$,
  - A grid defined as $-2^{-D_{\log}}$ on $[-0.25, 0)$, where $D_{\log}$ is a regular grid of $60 = \frac{240}{4}$ point on $[2, 32]$,
  - The point 0,
  - A grid defined as $2^{-D_{\log}}$ on $(0, 0.25]$, where $D_{\log}$ is a regular grid of $60 = \frac{240}{4}$ point on $[2, 32]$,
  - A regular grid of $60 = \frac{240}{4}$ points on $(0.25, 1]$.

With these definitions, the following grids were constructed:

- $G_\Omega = D_t \times D_x$: The grid for the domain $\Omega = [0, 1] \times [-1, 1]$,
- $G_{\partial\Omega_{\text{border}}} = D_t \times \{-1, 1\}$: The grid for the boundary $\partial\Omega_{\text{border}} = [0, 1] \times \{-1, 1\}$,
- $G_{\partial\Omega_0} = \{0\} \times D_x$: The grid for the boundary $\partial\Omega_0 = \{0\} \times [-1, 1]$.

We applied ANaGRAM for 1000 iterations, using a neural network with three hidden layers, each containing 32 neurons, resulting in a total of $P = 2241$ parameters. The cutoff factor is set to $\epsilon = 5.10^{-7} \times \Delta_{\boldsymbol{\theta}\max}$, where $\Delta_{\boldsymbol{\theta}\max}$ represents the largest eigenvalue of $\widehat{\phi}_{\boldsymbol{\theta}}$ (see line 1 of Algorithm 2). In Figure 9, we present the training and test losses for the different viscosities. The test loss was calculated using a grid constructed similarly to the training grid but with five times as many points.

For a viscosity of $\nu = 10^{-3}$, the test loss does not appear to converge, whereas the train loss does. This behavior can be attributed to the grid computation method, which disproportionately emphasizes points near the shock. This explanation is supported by the solution profile shown in Figure 10, where, for $\nu = 10^{-3}$, the shock in the learned solution is slightly shifted from $x = 0$ toward $x > 0$.

## B EXAMPLES OF PARAMETRIC MODELS

### B.1 PARTIAL FOURIER'S SERIES

Let us fix a dimension $d \in \mathbb{N}$. We then define the $N$-partial Fourier's Serie in $[0,1]^d$ as:

$$S_N : \begin{cases} \mathbb{R}^{[\![-N,N]\!]^d} & \to & \mathrm{L}^2([0,1]^d \to \mathbb{C}) \\ (\alpha_{k_1,\ldots,k_d}) & \mapsto & \left( x \in [0,1]^d \mapsto \sum_{k_1=-N}^{N} \cdots \sum_{k_d=-N}^{N} \alpha_{k_1,\ldots,k_d} e^{2i\pi\left(\sum_{l=1}^d k_l x_l\right)} \right) \end{cases} . \tag{30}$$

We see that for all $k \in [\![-N,N]\!]^d$ and $\boldsymbol{\theta} \in \mathbb{R}^{[\![-N,N]\!]^d}$, $\partial_k S_{N|\boldsymbol{\theta}} = \left( x \in \Omega \mapsto e^{2i\pi\left(\sum_{l=1}^d k_l x_l\right)} \right)$. As a consequence: for all $\boldsymbol{\theta} \in \mathbb{R}^{[\![-N,N]\!]^d}$

$$\mathrm{d}S_{N|\boldsymbol{\theta}} = S_N,$$

an thus: for all $\boldsymbol{\theta} \in \mathbb{R}^{[\![-N,N]\!]^d}$

$$\mathcal{M} = T_{\boldsymbol{\theta}}\mathcal{M} = \mathrm{Span}\left( x \in [0,1]^d \mapsto e^{2i\pi\left(\sum_{l=1}^d k_l x_l\right)} : k \in [\![-N,N]\!]^d \right) \tag{31}$$

This precisely means that $S_N$ is a linear parametric model.

### B.2 MULTILAYER PERCEPTRON

Historically, Multilayer perceptrons (MLPs) were the first neural network models to be proposed (Rosenblatt, 1958). Without going into an unnecessarily formal description, we will define MLPs of depth $L \in \mathbb{N}$ as a function $\mathbb{R}^n \to \mathbb{R}^m$ defined by induction:

**Initialization (Input Layer)** : $n_0 = n$, and

$$\mathbf{a}^{(0)} := x \in \mathbb{R}^{n_0}$$

**Inductive Step (Hidden Layers)** : for all $1 \leqslant l \leqslant L-1$, $n_l \in \mathbb{N}$, $\sigma^{(l)} : \mathbb{R} \to \mathbb{R}$, and

$$\mathbf{z}^{(l)} := \underbrace{\mathbf{W}^{(l)}}_{\in \mathbb{R}^{n_l, n_{l-1}}} \mathbf{a}^{(l-1)} + \underbrace{\mathbf{b}^{(l)}}_{\in \mathbb{R}^{n_l}}; \qquad \mathbf{a}^{(l)} := \underbrace{\sigma^{(l)}}_{\text{componentwise}} (\mathbf{z}^{(l)}),$$

**Final Step (Output Layer)** :

$$f_{\left(\mathbf{W}^{(l)}, \mathbf{b}^{(l)}\right)_{l=1}^L}(x) := \underbrace{\mathbf{W}^{(L)}}_{\in \mathbb{R}^{m, n_{L-1}}} \mathbf{a}^{(L-1)} + \underbrace{\mathbf{b}^{(L)}}_{\in \mathbb{R}^m}, \tag{32}$$

Equipped with this definition, we define a parametric model $u$ associated to $f_{\left(\mathbf{W}^{(l)}, \mathbf{b}^{(l)}\right)_{l=1}^L}$ by considering any differentiable parametrization $\varphi : \mathbb{R}^P \to \Pi_{l=1}^L \mathbb{R}^{w_{l-1} \times w_l} \times \mathbb{R}^{w_l}$, and then defining:

$$u : \begin{cases} \mathbb{R}^P & \to & \mathcal{H} \\ \boldsymbol{\theta} & \mapsto & f_{\varphi(\boldsymbol{\theta})} \end{cases}, \tag{33}$$

*i.e.* using $\varphi$ to encode the coefficients of the weights $\mathbf{W}^{(l)}$ and biases $\mathbf{b}^{(l)}$ in the coordinates of a vector in $\mathbb{R}^P$. Note that if $\varphi$ is bijective, then $P = \sum_{l=1}^L (w_{l-1}+1)w_l$.

*Remark* 2. A parametric model $u$ associated to an MLP, as defined in Equation (33), is not linear, if activations $(\sigma^{(l)})_{1 \leqslant l \leqslant L}$ are not, and $L \geqslant 2$ (yielding the qualifier "deep" in "deep learning"). Nevertheless, if $\varphi$ is linear, we may note that $u$ is still linear with respect to the parameters associated to weight $\mathbf{W}^{(L)}$ and bias $\mathbf{b}^{(L)}$. In particular, for all $\boldsymbol{\theta} \in \mathbb{R}^P$, $u_{|\boldsymbol{\theta}} \in \mathrm{Im}\, \mathrm{d}u_{|\boldsymbol{\theta}}$.

## C    Comprehensive introduction to Empirical Natural Gradient and ANaGRAM framework

In this section, we propose a more comprehensive introduction to the concepts introduced in Section 3, as well as proofs for Proposition 1, Theorem 1 and Proposition 2, which are stated therein. To this end, we need to review the notions of Neural Tangent Kernel (NTK) in Section C.1 and Reproducing Kernel Hilbert Space (RKHS) in Section C.2, before introducing empirical Natural Gradient in Section C.3, which is the key theoretical concept behind ANaGRAM.

### C.1    Neural Tangent Kernel (NTK)

**Neural Tangent Kernel (NTK)** has been introduced by Jacot et al. (2018) as a fundamental tool connecting neural networks to kernel methods, another very popular tool in Machine-learning (Schölkopf et al., 2002). More precisely, it shows that for an empirical quadratic loss:

$$\ell(\boldsymbol{\theta}) := \frac{1}{2S} \sum_{i=1}^{S} \left( u_{|\boldsymbol{\theta}}(x_i) - f(x_i) \right)^2, \tag{11}$$

the gradient descent:

$$\boldsymbol{\theta}_{t+1} \leftarrow \boldsymbol{\theta}_t - \eta \, \nabla \ell, \tag{8}$$

can be reinterpreted in the functional space, in the limit $\eta \to 0$, as the the differential equation in $u_{|\boldsymbol{\theta}} : \mathbb{R}^+ \to \mathrm{L}^2 \left( \Omega \to \mathbb{R} \right)$, with the initial data $u_{|\boldsymbol{\theta}_0} = u_0 \in \mathrm{L}^2 \left( \Omega \to \mathbb{R} \right)$:

$$\frac{\mathrm{d}u_{|\boldsymbol{\theta}_t}}{\mathrm{d}t}(x) = - \sum_{i=1}^{S} NTK_{\boldsymbol{\theta}_t}(x, x_i) \left( u_{|\boldsymbol{\theta}_t}(x_i) - y_i \right), \quad NTK_{\boldsymbol{\theta}}(x, y) := \sum_{p=1}^{P} \left( \partial_p u_{|\boldsymbol{\theta}}(x) \right) \left( \partial_p u_{|\boldsymbol{\theta}}(y) \right)^t. \tag{15}$$

By the same observations as for Equation (14), we observe that Equation (15) induces the following differential equation in $\boldsymbol{\theta} : \mathbb{R}^+ \to \mathbb{R}^P$, with the initial data $\boldsymbol{\theta}(0) = \boldsymbol{\theta}_0 \in \mathbb{R}^P$:

$$\frac{\mathrm{d}\boldsymbol{\theta}}{\mathrm{d}t} = \mathrm{d}u_{|\boldsymbol{\theta}_t}^\dagger \left( - \sum_{i=1}^{N} NTK_{\boldsymbol{\theta}_t}(x, x_i)(u_{|\boldsymbol{\theta}_t}(x_i) - y_i) \right) = - \sum_{i=1}^{N} \mathrm{d}u_{|\boldsymbol{\theta}_t}^\dagger \left( NTK_{\boldsymbol{\theta}_t}(x, x_i) \right) (u_{|\boldsymbol{\theta}_t}(x_i) - y_i), \tag{34}$$

Using Euler's approximation method, this can of course be rewritten as the discrete upate:

$$\boldsymbol{\theta}_{t+1} = \boldsymbol{\theta}_t - \eta \sum_{i=1}^{N} \mathrm{d}u_{|\boldsymbol{\theta}_t}^\dagger \left( NTK_{\boldsymbol{\theta}_t}(x, x_i) \right) (u_{|\boldsymbol{\theta}_t}(x_i) - y_i), \tag{35}$$

Note that if we assume $\mathrm{d}u_{|\boldsymbol{\theta}_t}$ to be inversible, we can *implicitly* compute the inverse, yielding:

$$\mathrm{d}u_{|\boldsymbol{\theta}_t}^\dagger \left( NTK_{\boldsymbol{\theta}_t}(x, x_i) \right) = \sum_{p=1}^{P} \partial_p u_{|\boldsymbol{\theta}_t}(x_i) \, e^{(p)}, \tag{36}$$

making Equation (35) effectively correspond to the usual gradient descent in Equation (8). Rudner et al. (2019) further extended this framework to the case of natural gradient descent in the context of information geometry. They demonstrate that the learning dynamic in this scenario is governed by a new kernel, named the **Natural Neural Tangent Kernel (NNTK)**, which is defined as: for all $\boldsymbol{\theta} \in \mathbb{R}^P$

$$NNTK_{\boldsymbol{\theta}}(x, y) := \sum_{1 \leqslant p, q \leqslant P} \left( \partial_p u_{|\boldsymbol{\theta}}(x) \right) F_{\boldsymbol{\theta} \, pq}^\dagger (\partial_p u_{|\boldsymbol{\theta}}(y))^t, \tag{37}$$

where $F_{\boldsymbol{\theta}}$ is the Fisher information matrix. In the more general context of Riemannian Geometry, Bai et al. (2022) show that the NNTK is given by: for all $\boldsymbol{\theta} \in \mathbb{R}^P$

$$NNTK_{\boldsymbol{\theta}}(x, y) := \sum_{1 \leqslant p, q \leqslant P} \left( \partial_p u_{|\boldsymbol{\theta}}(x) \right) G_{\boldsymbol{\theta} \, pq}^\dagger (\partial_p u_{|\boldsymbol{\theta}}(y))^t, \tag{38}$$

with $G_{\boldsymbol{\theta}}$ being the Gram matrix relative to a Riemannian metric $\mathcal{G}_{\boldsymbol{\theta}}$ as introduced in Section 2.3: for all $\boldsymbol{\theta} \in \mathbb{R}^P$, for all $1 \leqslant p, q \leqslant P$

$$G_{\boldsymbol{\theta} \, p, q} := \mathcal{G}_{\boldsymbol{\theta}}(\partial_p u_{|\boldsymbol{\theta}}, \partial_q u_{|\boldsymbol{\theta}}). \tag{39}$$

In particular, when $\mathcal{G}_{\boldsymbol{\theta}}$ is given by the metric of an ambient Hilbert space $\mathcal{H}$, this yields: for all $\boldsymbol{\theta} \in \mathbb{R}^P$, for all $1 \leqslant p, q \leqslant P$, for all $x, y \in \Omega$

$$NNTK_{\boldsymbol{\theta}}(x,y) := \sum_{1 \leqslant p,q \leqslant P} \left(\partial_p u_{|\boldsymbol{\theta}}(x)\right) G_{\boldsymbol{\theta}\,pq}^{\dagger} \left(\partial_q u_{|\boldsymbol{\theta}}(y)\right)^t, \quad G_{\boldsymbol{\theta},q} := \left\langle \partial_p u_{|\boldsymbol{\theta}}, \partial_q u_{|\boldsymbol{\theta}} \right\rangle_{\mathcal{H}}. \quad (16)$$

For the quadratic problem of Equation (11), natural gradient then yields the functional dynamics:

$$\frac{\mathrm{d} u_{|\boldsymbol{\theta}_t}}{\mathrm{d}t}(x) = -\sum_{i=1}^{N} NNTK_{\boldsymbol{\theta}_t}(x, x_i)(u_{|\boldsymbol{\theta}_t}(x_i) - y_i). \quad (40)$$

In the following, we will further explore the (N)NTK and its connection to the natural gradient in the context of Reproducing Kernel Hilbert Space (RKHS) theory.

## C.2 A PERSPECTIVE ON REPRODUCING KERNEL HILBERT SPACES (RKHS)

In this subsection we will carefully review the intimate link between neural tangent kernels, projections and reproducing kernels. To begin, let us define what a kernel function is, following Paulsen and Raghupathi (2016, Definition 2.12):

**Definition 3.** A function $k : \Omega \times \Omega \to \mathbb{K}$, $\mathbb{K} \in \{\mathbb{R}, \mathbb{C}\}$ is called a **kernel function** provided that: for all $N \in \mathbb{N}$, for all $(x_i) \in \Omega^N$, for all $\alpha \in \mathbb{K}^N$

$$\sum_{1 \leqslant i,j \leqslant N} \alpha_i k(x_i, x_j) \overline{\alpha}_j \geqslant 0 \quad (41)$$

Equipped with Definition 3, we can state the following theorem, which binds different perspectives on RKHS:

**Theorem 3.** *An Hilbert space $\mathcal{H}$ consisting of functions $\Omega \to \mathbb{K}$, $\mathbb{K} \in \{\mathbb{R}, \mathbb{C}\}$, is a Reproducing Kernel Hilbert Space if and only if one of the following equivalent conditions are met:*

*1. There is a kernel $k : \Omega \times \Omega \to \mathbb{R}$ such that $\mathcal{H} = \overline{\mathrm{Span}\left(k(x, \cdot) : x \in \Omega\right)}$ and: for all $x, y \in \Omega$*

$$\left\langle k(x, \cdot), k(y, \cdot) \right\rangle_{\mathcal{H}} = k(x, y). \quad (42)$$

*In particular, we have the reproducing property: for all $v \in \mathcal{H}$, for all $x \in \Omega$*

$$v(x) = \left\langle k(x, \cdot), v \right\rangle_{\mathcal{H}}. \quad (43)$$

*2. For all $x \in \Omega$, the evaluation form $e_x : f \in \mathcal{H} \mapsto f(x)$ is continuous.*

A proof of this theorem can be found in Paulsen and Raghupathi (2016, Definition 1.1, Proposition 2.13 and Moore's Theorem 2.14). We now draw some easy but essential consequences from Theorem 3:

**Corollary 1.** *Any finite dimensional Hilbert space $\mathcal{H}$ is a RKHS*

*Proof.* Since $\mathcal{H}$ is finite dimensional, all norms are equivalent. In particular $\|\cdot\|_{\infty} : f \in \mathcal{H} \mapsto \sup_{x \in \Omega} |f(x)|$ is equivalent to $\|\cdot\|_{\mathcal{H}}$. Then by point 2 in Theorem 3, $\mathcal{H}$ is a RKHS, since for all $x \in \Omega$, $e_x$ is continuous for $\|\cdot\|_{\infty}$. $\qquad \square$

In order to set out an important theorem that highlights the link between RKHS and projections, we need the following definition:

**Definition 4.** A linear operator $A : \mathcal{H} \to \mathcal{H}$ is an integral operator given that there is $k : \Omega \times \Omega \to \mathbb{K}$, $\mathbb{K} \in \{\mathbb{R}, \mathbb{C}\}$, such that: for all $f \in \mathcal{H}$, for all $x \in \Omega$

$$A(f)(x) = \left\langle k(x, \cdot), f \right\rangle_{\mathcal{H}}. \quad (44)$$

We can now state:

**Theorem 4.** *If $\mathcal{H}_0 \subset \mathcal{H}$ is a RKHS, then the orthogonal projection onto $\mathcal{H}_0$, $\Pi_{\mathcal{H}_0} : \mathcal{H} \to \mathcal{H}_0$ is an integral operator whose kernel is the reproducing kernel $k$ of $\mathcal{H}_0$, i.e. : for all $f \in \mathcal{H}$, for all $x \in \Omega$*

$$\Pi_{\mathcal{H}_0}(f)(x) = \langle k(x, \cdot), f \rangle_{\mathcal{H}} \tag{45}$$

*In addition, $k$ is given by: for all $x, y \in \Omega$*

$$k(x, y) = \sum_{i \in \mathbb{N}} L_i(x) L_i(y) \tag{46}$$

*where $(L_i)_{i \in \mathbb{N}}$ is any orthonormal basis of $\mathcal{H}_0$.*

*Proof.* The second claim is essentially equivalent to Paulsen and Raghupathi (2016, Theorem 2.4). To prove the first claim, let us notice that the reproducing property: for all $v \in \mathcal{H}_0$, for all $x \in \Omega$

$$v(x) = \langle k(x, \cdot), v \rangle_{\mathcal{H}}, \tag{47}$$

exactly means that the identity on $\mathcal{H}_0$ is an integral operator whose kernel is the reproducing kernel $k$ of $\mathcal{H}_0$. Since the restriction of $\Pi_{\mathcal{H}_0}$ to $\mathcal{H}_0$ is the identity on $\mathcal{H}_0$, it is sufficient to conclude the proof to show that: for all $f \in \mathcal{H}_0^\perp$, for all $x \in \Omega$

$$\langle k(x, \cdot), f \rangle_{\mathcal{H}} = 0. \tag{48}$$

But this is an immediate consequence of the second claim. $\qquad\square$

*Remark* 3. Theorem 4 encapsulates finite dimensional case, since one may take $L_i = 0$ for $i$ greater than $D \in \mathbb{N}$, yielding $\dim(\mathcal{H}_0) \leqslant D$, which implies in particular that $\mathcal{H}_0$ is an RKHS by Corollary 1.
*Remark* 4. The assumption $\mathcal{H}_0$ is an RKHS is essential, since there is no guaranty that such a space (finite dimension aside) is indeed a RKHS. One may think for instance to the case where the $(L_i)$ are the Fourier's polynomials defined in Section B.1. In this case the associated kernel is the Dirichlet kernel, which is well known to be non convergent, neither pointwise, nor in $\mathrm{L}^2([0, 2\pi])$.

Theorem 4 prompts the question of how to construct such an orthogonal basis $(L_i)$. Assume that we already have a basis for $\mathcal{H}_0$, i.e., $\mathcal{H}_0 = \overline{\mathrm{Span}(u_p : p \in \mathbb{N})} \subset \mathcal{H}$. While a Gram-Schmidt procedure could be used, there is another approach that, in a certain sense, is far more optimal. For the sake of simplicity, let us use suppose that $\mathcal{H}_0$ is finite dimensional[6]. Then:

**Lemma 1.** *Let us be $\mathcal{H}_0 := \mathrm{Span}(u_p : 1 \leqslant p \leqslant P) \subset \mathcal{H}$ and consider the Gram matrix $G_{pq} := \langle u_p, u_q \rangle_{\mathcal{H}}$ of $(u_p)$ and its eigen-decomposition $G = U\Delta^2 U^t$. Then:*

$$L_p := \sum_{1 \leqslant q \leqslant P} u_q U_{q,p} \Delta_p^\dagger, \tag{49}$$

*is an orthonormal basis of $\mathcal{H}_0$. In particular, $\Pi_{\mathcal{H}_0}$ is an integral operator whose kernel is:*

$$k(x, y) = \sum_{1 \leqslant p,q \leqslant P} u_p(x) G_{p,q}^\dagger u_q(y). \tag{50}$$

*Furthermore $L_p$ are the left-singular vector of the so-called **synthesis** operator[7]:*

$$\mathcal{T} : \begin{cases} \mathbb{R}^P & \to & \mathcal{H}_0 \\ \alpha & \mapsto & \sum_{1 \leqslant p \leqslant P} \alpha_p u_p \end{cases}. \tag{51}$$

*Proof.* Since $\mathcal{H}_0$ is generated by the finite basis $(u_p)_{1 \leqslant p \leqslant P}$, it is an RKHS by Corollary 1 and there exist (for instance by the Gram-Schmidt procedure), an orthonormal basis $(V_p)_{1 \leqslant p \leqslant P}$ of $\mathcal{H}_0$. Then by Theorem 4, the operator $\Pi$ defined by: for all $f \in \mathcal{H}$

$$\Pi(f) := \sum_{1 \leqslant p \leqslant P} V_p \langle f, V_p \rangle \tag{52}$$

---

[6]The infinite-dimensional case is more technical, as we have to be careful with the continuity of linear applications. As we are only considering a finitely-parameterized model, this is beyond the scope of our present work.
[7]Name and notation are taken from Adcock and Huybrechs (2019).

is the orthogonal projection onto $\mathcal{H}_0$. But the basis $(L_p)_{\leqslant p \leqslant P}$ of Equation (49) is precisely orthonormal. Indeed:

$$
\begin{aligned}
\langle L_p \,,\, L_q \rangle &= \left\langle \sum_{1 \leqslant k \leqslant P} u_k U_{k,p} \Delta_p^\dagger \,,\, \sum_{1 \leqslant l \leqslant P} u_l U_{l,q} \Delta_q^\dagger \right\rangle \\
&= \sum_{1 \leqslant k \leqslant P} \sum_{1 \leqslant l \leqslant P} \Delta_p^\dagger U_{p,k} \langle u_k \,,\, u_l \rangle U_{l,q} \Delta_q^\dagger = {\boldsymbol{e}^{(p)}}^t \Delta^\dagger U^t G U \Delta^\dagger \boldsymbol{e}^{(q)} \\
&= {\boldsymbol{e}^{(p)}}^t \Delta^\dagger U^t U \Delta^2 U^t U \Delta^\dagger \boldsymbol{e}^{(q)} = \delta_{pq},
\end{aligned}
$$

where $\delta_{pq}$ is the Kronecker symbol such that $\delta_{pq} = 1$ if and only if $p = q$. Now building $\Pi$ upon this basis yields: for all $f \in \mathcal{H}$

$$
\begin{aligned}
\Pi(f) &= \sum_{1 \leqslant p \leqslant P} L_p \langle L_p \,,\, f \rangle = \sum_{1 \leqslant p \leqslant P} \sum_{1 \leqslant k \leqslant P} \sum_{1 \leqslant l \leqslant P} u_k U_{k,p} \Delta_p^\dagger \langle u_l U_{l,p} \Delta_p^\dagger \,,\, f \rangle \\
&= \sum_{1 \leqslant k \leqslant P} \sum_{1 \leqslant l \leqslant P} u_k \left( \sum_{1 \leqslant p \leqslant P} U_{k,p} {\Delta_p^2}^\dagger U_{p,l} \right) \langle u_l \,,\, f \rangle = \sum_{1 \leqslant k,l \leqslant P} u_k G_{k,l}^\dagger \langle u_l \,,\, f \rangle
\end{aligned}
$$

Thus, the kernel of the projection $\Pi_{\mathcal{H}_0}$ onto $\mathcal{H}_0$ is precisely:

$$
k(x,y) = \sum_{1 \leqslant k,l \leqslant P} u_k(x) G_{k,l}^\dagger u_l(y).
$$

Finally, let us write the SVD of the *synthesis* operator: $\forall \alpha \in \mathbb{R}^P$

$$
\mathcal{T}(\alpha) = \sum_{1 \leqslant p \leqslant P} v_p \Lambda_p W_p^t \alpha \in \mathcal{H}_0. \tag{53}
$$

Then in particular, we have: for all $1 \leqslant p \leqslant P$

$$
v_p = \mathcal{T}(W_p \Lambda_p^\dagger),
$$

and: for all $1 \leqslant p \leqslant P$

$$
u_p = \mathcal{T}(\boldsymbol{e}^{(p)}).
$$

This implies that: for all $1 \leqslant p, q \leqslant P$

$$
\begin{aligned}
G_{p,q} &= \langle u_p \,,\, u_q \rangle = \left\langle \mathcal{T}(\boldsymbol{e}^{(p)}) \,,\, \mathcal{T}(\boldsymbol{e}^{(q)}) \right\rangle \\
&\overset{(53)}{=} \sum_{1 \leqslant k,l \leqslant P} {\boldsymbol{e}^{(p)}}^t W_k \Lambda_k \underbrace{\langle v_k \,,\, v_l \rangle}_{=\delta_{kl}} \Lambda_l W_l^t \boldsymbol{e}^{(q)} = {\boldsymbol{e}^{(p)}}^t W \Lambda^2 W^t \boldsymbol{e}^{(q)}.
\end{aligned}
$$

This means that $(W_p)$ and $(\Lambda_p^2)$ are respectively the eigenvectors and eigenvalues of $G$. The result follows by unicity of eigen-decomposition and respective identification of $(W_p)$ to $(U_p)$ and $(\Lambda_p)$ to $(\Delta_p)$ in Equation (49). □

This observation will enable us to establish the main result of this section, linking RKHS theory, NTK and natural gradient, in the following corollary.

**Corollary 2.** *The $NNTK_{\boldsymbol{\theta}}$ defined in Equation (16) is the kernel of the projection $\Pi_{T_{\boldsymbol{\theta}}\mathcal{M}} : \mathcal{H} \to \mathcal{H}$ onto $T_{\boldsymbol{\theta}}\mathcal{M}$.*

*Proof.* This is a direct consequence of Lemma 1, since $T_{\boldsymbol{\theta}}\mathcal{M} = \mathrm{Span}(\partial_p u_{|\boldsymbol{\theta}} : 1 \leqslant p \leqslant P)$. □

In the following, we will derive some consequences from NNTK theory, leading to the concept of the empirical Natural Gradient (eNG).

### C.3 EMPIRICAL NATURAL GRADIENT (ENG)

To begin, we need to make a key observation:

- Equation (15), namely:

$$\frac{\mathrm{d}u_{|\boldsymbol{\theta}_t}}{\mathrm{d}t}(x) = -\sum_{i=1}^{S} NTK_{\boldsymbol{\theta}_t}(x, x_i)\left(u_{|\boldsymbol{\theta}_t}(x_i) - y_i\right),$$

  shows that the empirical dynamics under gradient descent happens in the space:

$$\widehat{T}^{NTK}_{\boldsymbol{\theta},(x_i)}\mathcal{M} := \mathrm{Span}(NTK_{\boldsymbol{\theta}}(\cdot, x_i) : (x_i)_{1 \leqslant i \leqslant N}) \subset T_{\boldsymbol{\theta}}\mathcal{M}. \tag{54}$$

- Likewise

$$\frac{\mathrm{d}u_{|\boldsymbol{\theta}_t}}{\mathrm{d}t}(x) = -\sum_{i=1}^{N} NNTK_{\boldsymbol{\theta}_t}(x, x_i)(u_{|\boldsymbol{\theta}_t}(x_i) - y_i), \tag{40}$$

  shows that the empirical dynamics under natural gradient descent happens in the space:

$$\widehat{T}^{NNTK}_{\boldsymbol{\theta},(x_i)}\mathcal{M} := \mathrm{Span}(NNTK_{\boldsymbol{\theta}}(\cdot, x_i) : (x_i)_{1 \leqslant i \leqslant S}) \subset T_{\boldsymbol{\theta}}\mathcal{M}. \tag{17}$$

Both spaces $\widehat{T}^{NTK}_{\boldsymbol{\theta},(x_i)}\mathcal{M}$ and $\widehat{T}^{NNTK}_{\boldsymbol{\theta},(x_i)}\mathcal{M}$ are subspaces of the tangent space $T_{\boldsymbol{\theta}}\mathcal{M} := \mathrm{Im}\,\mathrm{d}u_{|\boldsymbol{\theta}}$. Therefore, it remains true that the empirical functional dynamics occurs within $T_{\boldsymbol{\theta}}\mathcal{M}$. However, $\widehat{T}^{NTK}_{\boldsymbol{\theta},(x_i)}\mathcal{M}$ and $\widehat{T}^{NNTK}_{\boldsymbol{\theta},(x_i)}\mathcal{M}$ are the smallest subspaces in which the empirical functional dynamics take place, respectively for classical and natural gradient descent. We encapsulate it in a defintion:

**Definition 5** (empirical tangent space). Given a parametric model $u : \mathbb{R}^P \to \mathcal{H}$, and a batch of points $(x_i)_{1 \leqslant i \leqslant S}$, the **empirical tangent space relative to the points** $(x_i)_{1 \leqslant i \leqslant S}$, is the space:

$$\widehat{T}^{NNTK}_{\boldsymbol{\theta},(x_i)}\mathcal{M} := \mathrm{Span}(NNTK_{\boldsymbol{\theta}}(\cdot, x_i) : (x_i)_{1 \leqslant i \leqslant S}) \subset T_{\boldsymbol{\theta}}\mathcal{M}. \tag{17}$$

When the context is clear, its name will be abbreviated **empirical tangent space** and it will be denoted:

$$\widehat{T}_{\boldsymbol{\theta}}\mathcal{M} := \mathrm{Span}\left(NNTK_{\boldsymbol{\theta}}(\cdot, x_i) : (x_i)_{1 \leqslant i \leqslant S}\right) \tag{55}$$

The second key observation is the following : since natural gradient descent, in the limit $S \to \infty$ (population limit), is given by the update (*cf.* end of Section 2.3):

$$\boldsymbol{\theta}_{t+1} \leftarrow \boldsymbol{\theta}_t - \eta\,\mathrm{d}u^{\dagger}_{|\boldsymbol{\theta}_t}\left(\Pi^{\perp}_{T_{\boldsymbol{\theta}_t}\mathcal{M}}\left(\nabla\mathcal{L}_{|u_{|\boldsymbol{\theta}_t}}\right)\right), \tag{14}$$

one may also define a similar update in the empirical tangent space $\widehat{T}_{\boldsymbol{\theta}}\mathcal{M}$. This observation motivates the following definition, already introduced in Equation (18):

**Definition 6** (empirical Natural Gradient (eNG)). The **empirical Natural Gradient (eNG)** is the update given by the projection of the functional gradient on the empirical tangent space $\widehat{T}_{\boldsymbol{\theta}}\mathcal{M}$, *i.e.* :

$$\boldsymbol{\theta}_{t+1} \leftarrow \boldsymbol{\theta}_t - \eta\,\mathrm{d}u^{\dagger}_{|\boldsymbol{\theta}_t}\left(\Pi^{\perp}_{\widehat{T}_{\boldsymbol{\theta}_t}\mathcal{M}}\nabla\mathcal{L}_{|u_{|\boldsymbol{\theta}_t}}\right). \tag{56}$$

The problem now is to find a tractable procedure to compute the update of Equation (56). This is the aim of Theorem 1 stated in Section 3, that we now recall and prove:

**Theorem 1** (ANaGRAM). *Let us define for all $1 \leqslant i \leqslant S$ and for all $1 \leqslant p \leqslant P$:*

$$\widehat{\phi}_{\boldsymbol{\theta}\,i,p} := \partial_p u_{|\boldsymbol{\theta}}(x_i)\,; \qquad\qquad \widehat{\nabla\mathcal{L}}_{|u_{|\boldsymbol{\theta}}\,i} := \nabla\mathcal{L}_{|u_{|\boldsymbol{\theta}}}(x_i) = u_{|\boldsymbol{\theta}}(x_i) - f(x_i).$$

*Then:*

$$\mathrm{d}u^{\dagger}_{|\boldsymbol{\theta}}\left(\Pi^{\perp}_{\widehat{T}^{NNTK}_{\boldsymbol{\theta},(x_i)}\mathcal{M}}\nabla\mathcal{L}_{|u_{|\boldsymbol{\theta}}}\right) = \left(\widehat{\phi}^{\dagger}_{\boldsymbol{\theta}} + E^{metric}_{\boldsymbol{\theta}}\right)\left(\widehat{\nabla\mathcal{L}}_{|u_{|\boldsymbol{\theta}}} + E^{\perp}_{\boldsymbol{\theta}}\right), \tag{19}$$

*where $E^{metric}_{\boldsymbol{\theta}}$ and $E^{\perp}_{\boldsymbol{\theta}}$ are correction terms specified in Equations (57) and (58) in Section C.3, respectively accounting for the metric's impact on empirical tangent space defintion, and the substraction of the evaluation of the orthogonal part[8] of the functional gradient.*

---

[8] orthogonal to the whole tangent space $T_{\boldsymbol{\theta}}\mathcal{M}$.

**Theorem 1 specifications** The corrections terms $E_{\boldsymbol{\theta}}^{\text{metric}}$ and $E_{\boldsymbol{\theta}}^{\perp}$ are given by:

$$E_{\boldsymbol{\theta}}^{\text{metric}} = \widehat{V}_{\boldsymbol{\theta}} \left( \boldsymbol{I}_P - \Pi_r \right) \widehat{V}_{\boldsymbol{\theta}}^t G_{\boldsymbol{\theta}}^\dagger \widehat{V}_{\boldsymbol{\theta}} \Pi_r \left( \Pi_r \widehat{V}_{\boldsymbol{\theta}}^t G_{\boldsymbol{\theta}}^\dagger \widehat{V}_{\boldsymbol{\theta}} \Pi_r \right)^\dagger \widehat{\Delta}_{\boldsymbol{\theta}}^\dagger \widehat{U}_{\boldsymbol{\theta}}^t, \tag{57}$$

with:

- $\Pi_r = \sum_{p=1}^r \boldsymbol{e}^{(p)} \boldsymbol{e}^{(p)t}$, the projection onto the $r$ first coordinates of $\mathbb{R}^P$.
- $\boldsymbol{I}_P$, the identity of $\mathbb{R}^P$
- $\widehat{U}_{\boldsymbol{\theta}} \widehat{\Delta}_{\boldsymbol{\theta}} \widehat{V}_{\boldsymbol{\theta}}^t = SVD(\widehat{\phi}_{\boldsymbol{\theta}})$
- for all $1 \leqslant p, q \leqslant P$, $G_{\boldsymbol{\theta}\, p,q} = \left\langle \partial_p u_{|\boldsymbol{\theta}} , \partial_q u_{|\boldsymbol{\theta}} \right\rangle_{\mathcal{H}}$

$$E_{\boldsymbol{\theta}}^{\perp} = \left( \left\langle NNTK_{\boldsymbol{\theta}}(x_i, \cdot) , \nabla\mathcal{L} \right\rangle_{\mathcal{H}} - \nabla\mathcal{L}(x_i) \right)_{1 \leqslant i \leqslant S} = \left( - \left( \Pi_{T_{\boldsymbol{\theta}}^\perp \mathcal{M}}^\perp \nabla\mathcal{L} \right)(x_i) \right)_{1 \leqslant i \leqslant S} \tag{58}$$

*Proof.* First of all, following Lemma 1, we see that the projection kernel into $\widehat{T}_{\boldsymbol{\theta}}\mathcal{M}$ is given by: for all $x, y \in \Omega$

$$\hat{k}(x, y) = \sum_{1 \leqslant i,j \leqslant S} NNTK_{\boldsymbol{\theta}}(x_i, x) \widehat{G}_{\boldsymbol{\theta}\, i,j}^\dagger NNTK_{\boldsymbol{\theta}}(x_j, y), \tag{59}$$

with: for all $1 \leqslant i, j \leqslant S$, for all $\boldsymbol{\theta} \in \mathbb{R}^P$

$$\widehat{G}_{\boldsymbol{\theta}\, i,j} = \left\langle NNTK_{\boldsymbol{\theta}}(x_i, \cdot) , NNTK_{\boldsymbol{\theta}}(x_j, \cdot) \right\rangle_{\mathcal{H}} = NNTK_{\boldsymbol{\theta}}(x_i, x_j), \tag{60}$$

where last equality comes from the fact that $NNTK_{\boldsymbol{\theta}}$ is the reproducing kernel of $T_{\boldsymbol{\theta}}\mathcal{M}$. We can then simplify Equation (56):

$$\mathrm{d}u_{|\boldsymbol{\theta}_t}^\dagger \left( \Pi_{\widehat{T}_{\boldsymbol{\theta}_t}\mathcal{M}}^\perp \nabla\mathcal{L}_{|u_{|\boldsymbol{\theta}_t}} \right) = \mathrm{d}u_{|\boldsymbol{\theta}_t}^\dagger \left( x \in \Omega \mapsto \left\langle \hat{k}(x, \cdot) , \nabla\mathcal{L}_{|u_{|\boldsymbol{\theta}_t}} \right\rangle_{\mathcal{H}} \right) \tag{61}$$

$$= \sum_{1 \leqslant i,j \leqslant S} \mathrm{d}u_{|\boldsymbol{\theta}_t}^\dagger (NNTK_{\boldsymbol{\theta}_t}(x_i, \cdot)) \widehat{G}_{\boldsymbol{\theta}_t\, i,j}^\dagger \left\langle NNTK_{\boldsymbol{\theta}_t}(x_j, \cdot) , \nabla\mathcal{L}_{|u_{|\boldsymbol{\theta}_t}} \right\rangle_{\mathcal{H}} \tag{62}$$

$$= \sum_{\substack{1 \leqslant p,q \leqslant P \\ 1 \leqslant i,j \leqslant S}} \mathrm{d}u_{|\boldsymbol{\theta}_t}^\dagger \left( \partial_p u_{|\boldsymbol{\theta}_t} G_{\boldsymbol{\theta}_t\, p,q}^\dagger \right) \partial_q u_{|\boldsymbol{\theta}_t}(x_i) \widehat{G}_{\boldsymbol{\theta}_t\, i,j}^\dagger \left\langle NNTK_{\boldsymbol{\theta}_t}(x_j, \cdot) , \nabla\mathcal{L}_{|u_{|\boldsymbol{\theta}_t}} \right\rangle_{\mathcal{H}}$$

$$\tag{63}$$

$$= \sum_{\substack{1 \leqslant p,q \leqslant P \\ 1 \leqslant i,j \leqslant S}} G_{\boldsymbol{\theta}_t\, p,q}^\dagger \partial_q u_{|\boldsymbol{\theta}_t}(x_i) \widehat{G}_{\boldsymbol{\theta}_t\, i,j}^\dagger \left\langle NNTK_{\boldsymbol{\theta}_t}(x_j, \cdot) , \nabla\mathcal{L}_{|u_{|\boldsymbol{\theta}_t}} \right\rangle_{\mathcal{H}}, \tag{64}$$

Using the empirical jacobian matrix introduced in the statement of Theorem 1, namely: for all $\boldsymbol{\theta} \in \mathbb{R}^P$ and for all $1 \leqslant i \leqslant S$, for all $1 \leqslant p \leqslant P$

$$\widehat{\phi}_{\boldsymbol{\theta}\, i,p} := \partial_p u_{|\boldsymbol{\theta}}(x_i). \tag{65}$$

Equation (60) rewrites: for all $\boldsymbol{\theta} \in \mathbb{R}^P$

$$\widehat{G}_{\boldsymbol{\theta}} = \widehat{\phi}_{\boldsymbol{\theta}} G_{\boldsymbol{\theta}}^\dagger \widehat{\phi}_{\boldsymbol{\theta}}^t. \tag{66}$$

Introducing also: for all $\boldsymbol{\theta} \in \mathbb{R}^P$ and for all $1 \leqslant i \leqslant S$

$$\widehat{\nabla\mathcal{L}}_{\boldsymbol{\theta}\, i}^{\|} := \left\langle NNTK_{\boldsymbol{\theta}}(x_i, \cdot) , \nabla\mathcal{L}_{|u_{|\boldsymbol{\theta}}} \right\rangle_{\mathcal{H}}. \tag{67}$$

Equation (64) then rewrites:

$$\mathrm{d}u_{|\boldsymbol{\theta}_t}^\dagger \left( \Pi_{\widehat{T}_{\boldsymbol{\theta}_t}\mathcal{M}}^\perp \nabla\mathcal{L}_{|u_{|\boldsymbol{\theta}_t}} \right) = G_{\boldsymbol{\theta}_t}^\dagger \widehat{\phi}_{\boldsymbol{\theta}_t}^t \left( \widehat{\phi}_{\boldsymbol{\theta}_t} G_{\boldsymbol{\theta}_t}^\dagger \widehat{\phi}_{\boldsymbol{\theta}_t}^t \right)^\dagger \widehat{\nabla\mathcal{L}}_{\boldsymbol{\theta}_t}^{\|}. \tag{68}$$

Using now the SVD of $\widehat{\phi}_{\boldsymbol{\theta}}$, also introduced in the statement of Theorem 1, namely: for all $\boldsymbol{\theta} \in \mathbb{R}^P$

$$\widehat{\phi}_{\boldsymbol{\theta}} = \widehat{U}_{\boldsymbol{\theta}} \widehat{\Delta}_{\boldsymbol{\theta}} \widehat{V}_{\boldsymbol{\theta}}^t, \tag{69}$$

we may express the pseudo-inverse of $\widehat{\phi}_{\boldsymbol{\theta}}$ as:

$$\widehat{\phi}_{\boldsymbol{\theta}}^{\dagger} = \widehat{U}_{\boldsymbol{\theta}}\widehat{\Delta}_{\boldsymbol{\theta}}^{\dagger}\widehat{V}_{\boldsymbol{\theta}}^{t}, \tag{70}$$

and rewrite Equation (66) as:

$$\widehat{G}_{\boldsymbol{\theta}} = \widehat{U}_{\boldsymbol{\theta}}\widehat{\Delta}_{\boldsymbol{\theta}}\widehat{V}_{\boldsymbol{\theta}}^{t}G_{\boldsymbol{\theta}}^{\dagger}\widehat{V}_{\boldsymbol{\theta}}\widehat{\Delta}_{\boldsymbol{\theta}}\widehat{U}_{\boldsymbol{\theta}}^{t}. \tag{71}$$

Let use denote $r \leqslant S$ the rank of $\widehat{\phi}_{\boldsymbol{\theta}}$, and introduce $\Pi_r := \sum_{p=1}^{r}\boldsymbol{e}^{(p)}\boldsymbol{e}^{(p)\,t}$ the projection onto the first $r$ coordinates of the canonical basis $\left(\boldsymbol{e}^{(p)}\right)_{p=1}^{P}$ of $\mathbb{R}^P$. Observe that, in particular, $\widehat{\Delta}_{\boldsymbol{\theta}}\widehat{\Delta}_{\boldsymbol{\theta}}^{\dagger} = \Pi_r$. Then, noting that $\widehat{U}_{\boldsymbol{\theta}}$ is orthogonal, and $\widehat{\Delta}_{\boldsymbol{\theta}}$ diagonal:

$$\widehat{G}_{\boldsymbol{\theta}}^{\dagger} = \widehat{U}_{\boldsymbol{\theta}}\widehat{\Delta}_{\boldsymbol{\theta}}^{\dagger}\left(\Pi_r\widehat{V}_{\boldsymbol{\theta}}^{t}G_{\boldsymbol{\theta}}^{\dagger}\widehat{V}_{\boldsymbol{\theta}}\Pi_r\right)^{\dagger}\widehat{\Delta}_{\boldsymbol{\theta}}^{\dagger}\widehat{U}_{\boldsymbol{\theta}}^{t} = \widehat{U}_{\boldsymbol{\theta}}\widehat{\Delta}_{\boldsymbol{\theta}}^{\dagger}\Sigma_{\boldsymbol{\theta}}^{\dagger}\widehat{\Delta}_{\boldsymbol{\theta}}^{\dagger}\widehat{U}_{\boldsymbol{\theta}}^{t}, \tag{72}$$

where

$$\Sigma_{\boldsymbol{\theta}} := \Pi_r\widehat{V}_{\boldsymbol{\theta}}^{t}G_{\boldsymbol{\theta}}^{\dagger}\widehat{V}_{\boldsymbol{\theta}}\Pi_r. \tag{73}$$

Inserting Equation (72) and Equation (69) in Equation (68), we get:

$$\mathrm{d}u_{|\boldsymbol{\theta}_t}^{\dagger}\left(\Pi_{\widehat{T}_{\boldsymbol{\theta}_t}\mathcal{M}}^{\perp}\nabla\mathcal{L}_{|u_{|\boldsymbol{\theta}_t}}\right) = G_{\boldsymbol{\theta}}^{\dagger}\widehat{V}_{\boldsymbol{\theta}}\underbrace{\widehat{\Delta}_{\boldsymbol{\theta}}\widehat{U}_{\boldsymbol{\theta}}^{t}\widehat{U}_{\boldsymbol{\theta}}\widehat{\Delta}_{\boldsymbol{\theta}}^{\dagger}}_{=\Pi_r}\Sigma_{\boldsymbol{\theta}}^{\dagger}\widehat{\Delta}_{\boldsymbol{\theta}}^{\dagger}\widehat{U}_{\boldsymbol{\theta}}^{t}\widehat{\nabla\mathcal{L}}_{\boldsymbol{\theta}}^{\parallel} \tag{74}$$

$$= \widehat{V}_{\boldsymbol{\theta}}\widehat{V}_{\boldsymbol{\theta}}^{t}G_{\boldsymbol{\theta}}^{\dagger}\widehat{V}_{\boldsymbol{\theta}}\,\Pi_r\,\Sigma_{\boldsymbol{\theta}}^{\dagger}\widehat{\Delta}_{\boldsymbol{\theta}}^{\dagger}\widehat{U}_{\boldsymbol{\theta}}^{t}\widehat{\nabla\mathcal{L}}_{\boldsymbol{\theta}}^{\parallel} \tag{75}$$

$$= \widehat{V}_{\boldsymbol{\theta}}\left(\left(\boldsymbol{I}_P - \Pi_r\right) + \Pi_r\right)\widehat{V}_{\boldsymbol{\theta}}^{t}G_{\boldsymbol{\theta}}^{\dagger}\widehat{V}_{\boldsymbol{\theta}}\,\Pi_r\,\Sigma_{\boldsymbol{\theta}}^{\dagger}\widehat{\Delta}_{\boldsymbol{\theta}}^{\dagger}\widehat{U}_{\boldsymbol{\theta}}^{t}\widehat{\nabla\mathcal{L}}_{\boldsymbol{\theta}}^{\parallel} \tag{76}$$

$$= \Big(\widehat{V}_{\boldsymbol{\theta}}\Sigma_{\boldsymbol{\theta}}\Sigma_{\boldsymbol{\theta}}^{\dagger}\widehat{\Delta}_{\boldsymbol{\theta}}^{\dagger}\widehat{U}_{\boldsymbol{\theta}}^{t} \tag{77}$$

$$+ \underbrace{\widehat{V}_{\boldsymbol{\theta}}\left(\boldsymbol{I}_P - \Pi_r\right)\widehat{V}_{\boldsymbol{\theta}}^{t}G_{\boldsymbol{\theta}}^{\dagger}\widehat{V}_{\boldsymbol{\theta}}\,\Pi_r\,\Sigma_{\boldsymbol{\theta}}^{\dagger}\widehat{\Delta}_{\boldsymbol{\theta}}^{\dagger}\widehat{U}_{\boldsymbol{\theta}}^{t}}_{E_{\boldsymbol{\theta}}^{\text{metric}}}\Big)\widehat{\nabla\mathcal{L}}_{\boldsymbol{\theta}}^{\parallel} \tag{78}$$

$$= \left(\widehat{V}_{\boldsymbol{\theta}}\widehat{\Delta}_{\boldsymbol{\theta}}^{\dagger}\widehat{U}_{\boldsymbol{\theta}}^{t} + E_{\boldsymbol{\theta}}^{\text{metric}}\right)\widehat{\nabla\mathcal{L}}_{\boldsymbol{\theta}}^{\parallel} = \left(\widehat{\phi}_{\boldsymbol{\theta}}^{\dagger} + E_{\boldsymbol{\theta}}^{\text{metric}}\right)\widehat{\nabla\mathcal{L}}_{\boldsymbol{\theta}}^{\parallel}. \tag{79}$$

Finally, by decomposing $\nabla\mathcal{L}_{|u_{|\boldsymbol{\theta}}}$ into its collinear and orthogonal components to $T_{\boldsymbol{\theta}}\mathcal{M}$, *i.e.*: for all $\boldsymbol{\theta} \in \mathbb{R}^P$

$$\nabla\mathcal{L}_{|u_{|\boldsymbol{\theta}}} = \Pi_{T_{\boldsymbol{\theta}}\mathcal{M}}^{\perp}\left(\nabla\mathcal{L}_{|u_{|\boldsymbol{\theta}}}\right) + \Pi_{T_{\boldsymbol{\theta}}^{\perp}\mathcal{M}}^{\perp}\left(\nabla\mathcal{L}_{|u_{|\boldsymbol{\theta}}}\right), \tag{80}$$

and using the notation $\widehat{\nabla\mathcal{L}}_{|u_{|\boldsymbol{\theta}}}$ introduced in Theorem 1 statement, we have: for all $\boldsymbol{\theta} \in \mathbb{R}^P$, for all $1 \leqslant i \leqslant S$

$$\widehat{\nabla\mathcal{L}}_{|u_{|\boldsymbol{\theta}}\,i} = \nabla\mathcal{L}_{|u_{|\boldsymbol{\theta}}}(x_i) = \Pi_{T_{\boldsymbol{\theta}}\mathcal{M}}^{\perp}\left(\nabla\mathcal{L}_{|u_{|\boldsymbol{\theta}}}\right)(x_i) + \Pi_{T_{\boldsymbol{\theta}}^{\perp}\mathcal{M}}^{\perp}\left(\nabla\mathcal{L}_{|u_{|\boldsymbol{\theta}}}\right)(x_i) \tag{81}$$

$$= \left\langle NNTK_{\boldsymbol{\theta}}(x_i, \cdot), \nabla\mathcal{L}_{|u_{|\boldsymbol{\theta}}}\right\rangle_{\mathcal{H}} + \Pi_{T_{\boldsymbol{\theta}}^{\perp}\mathcal{M}}^{\perp}\left(\nabla\mathcal{L}_{|u_{|\boldsymbol{\theta}}}\right)(x_i) \tag{82}$$

$$= \widehat{\nabla\mathcal{L}}_{\boldsymbol{\theta}\,i}^{\parallel} - E_{\boldsymbol{\theta}\,i}^{\perp}, \tag{83}$$

where Equation (82) comes from the fact that $NNTK_{\boldsymbol{\theta}}$ is the kernel defining $\Pi_{T_{\boldsymbol{\theta}}\mathcal{M}}^{\perp}$, and Equation (83) uses the defintion given by Equation (67) and the notation $E_{\boldsymbol{\theta}}^{\perp}$ introduced in Theorem 1 statement, specified in Equation (58). Thus $\widehat{\nabla\mathcal{L}}_{\boldsymbol{\theta}}^{\parallel} = \widehat{\nabla\mathcal{L}}_{|u_{|\boldsymbol{\theta}}} + E_{\boldsymbol{\theta}}^{\perp}$, which concludes.

$\square$

*Remark* 5. Note that the implicit inversion made in order to obtain Equation (64) is now exact, in the sense that we do not need to assume that $\mathrm{d}u_{|\boldsymbol{\theta}_t}$ is invertible anymore as in Equation (36), since the eigenvectors and eigenvalues of $G_{\boldsymbol{\theta}_t}$ respectively match singular vectors and singular values of $\mathrm{d}u_{|\boldsymbol{\theta}_t}$, as stated in Lemma 1. We call this the **exact implicit inversion trick**.

For the sake of understanding, let us suppose, that $\widehat{\phi}_{\boldsymbol{\theta}}$ is of rank $P$. Then in particular $S \geqslant P$ and Equation (72) rewrites:

$$\left(\widehat{\phi}_{\boldsymbol{\theta}}G_{\boldsymbol{\theta}}^{\dagger}\widehat{\phi}_{\boldsymbol{\theta}}^{t}\right)^{\dagger} = \left(\widehat{\phi}_{\boldsymbol{\theta}}^{t}\right)^{\dagger}G_{\boldsymbol{\theta}}\widehat{\phi}_{\boldsymbol{\theta}}^{\dagger}. \tag{84}$$

Since $S \geqslant P$, we also have $\widehat{\phi}_{\boldsymbol{\theta}}^t \left(\widehat{\phi}_{\boldsymbol{\theta}}^t\right)^\dagger = I_P$ and thus, Equation (68) simplifies to:

$$\mathrm{d}u_{|\boldsymbol{\theta}_t}^\dagger \left(\Pi_{\widehat{T}_{\boldsymbol{\theta}_t}\mathcal{M}}^\perp \nabla\mathcal{L}_{|u_{|\boldsymbol{\theta}_t}}\right) = G_{\boldsymbol{\theta}}^\dagger \widehat{\phi}_{\boldsymbol{\theta}}^t \left(\widehat{\phi}_{\boldsymbol{\theta}}^t\right)^\dagger G_{\boldsymbol{\theta}} \widehat{\phi}_{\boldsymbol{\theta}}^\dagger \widehat{\nabla\mathcal{L}}_{\boldsymbol{\theta}}^\| = G_{\boldsymbol{\theta}}^\dagger G_{\boldsymbol{\theta}} \widehat{\phi}_{\boldsymbol{\theta}}^\dagger \widehat{\nabla\mathcal{L}}_{\boldsymbol{\theta}} = \widehat{\phi}_{\boldsymbol{\theta}}^\dagger \widehat{\nabla\mathcal{L}}_{\boldsymbol{\theta}}^\|, \quad (85)$$

where last equality comes from the fact that $\left(\widehat{\phi}_{\boldsymbol{\theta}}^\dagger \widehat{\nabla\mathcal{L}}_{\boldsymbol{\theta}}^\|\right) \in \operatorname{Im} G_{\boldsymbol{\theta}}$ by its own definition and the one of $NNTK_{\boldsymbol{\theta}}$. This means that under those conditions, the term $E_{\boldsymbol{\theta}}^{\mathrm{metric}}$ of Theorem 1 vanishes. Unexpectedly, the assumption $\widehat{\phi}_{\boldsymbol{\theta}}$ has rank $P$ can be satisfied for a specific subset of points : those guaranteed by Proposition 1, restated below, which we will now prove:

**Proposition 1.** *There exist $P$ points $(\hat{x}_i)$ such that $\widehat{T}_{\boldsymbol{\theta},(x_i)}^{NNTK}\mathcal{M} = T_{\boldsymbol{\theta}}\mathcal{M}$. Then notably $E_{\boldsymbol{\theta}}^{metric} = 0$.*

*Proof.* Let $d := \dim(T_{\boldsymbol{\theta}}\mathcal{M}) \leqslant P$. By definition of $NNTK_{\boldsymbol{\theta}}$ (*cf.* Equation (16)), we have for all $x \in \Omega$:

$$NNTK_{\boldsymbol{\theta}}(\cdot, x) = \sum_{p=1}^{P} \alpha_p \partial_p u_{|\boldsymbol{\theta}} \in T_{\boldsymbol{\theta}}\mathcal{M}, \quad (86)$$

with for all $1 \leqslant p \leqslant P$, $\alpha_p = \sum_{q=1}^{P} \left(G_{\boldsymbol{\theta}}^\dagger\right)_{p,q} \partial_q u_{|\boldsymbol{\theta}}(x) \in \mathbb{R}$. Therefore $\widehat{T}_{\boldsymbol{\theta}}\mathcal{M} \subset T_{\boldsymbol{\theta}}\mathcal{M}$. We will start by showing that $\overline{\operatorname{Span}(NNTK_{\boldsymbol{\theta}}(\cdot, x) \ : \ x \in \Omega)} = T_{\boldsymbol{\theta}}\mathcal{M}$. $\overline{\operatorname{Span}(NNTK_{\boldsymbol{\theta}}(\cdot, x) \ : \ x \in \Omega)} \subset T_{\boldsymbol{\theta}}\mathcal{M}$ is clear from Equation (86). Let us now be $u \in T_{\boldsymbol{\theta}}\mathcal{M} \bigcap \overline{\operatorname{Span}(NNTK_{\boldsymbol{\theta}}(\cdot, x) \ : \ x \in \Omega)}^\perp$. Since $u \in \overline{\operatorname{Span}(NNTK_{\boldsymbol{\theta}}(\cdot, x) \ : \ x \in \Omega)}^\perp$, we have: for all $x \in \Omega$

$$0 = \langle NNTK_{\boldsymbol{\theta}}(\cdot, x), u \rangle = u(x),$$

where last equality comes from the fact that $NNTK_{\boldsymbol{\theta}}$ is the reproducing kernel of $T_{\boldsymbol{\theta}}\mathcal{M}$ (*cf.* Theorem 4). Therefore $u = 0$ and thus:

$$T_{\boldsymbol{\theta}}\mathcal{M} \bigcap \overline{\operatorname{Span}(NNTK_{\boldsymbol{\theta}}(\cdot, x) \ : \ x \in \Omega)}^\perp = \{0\},$$

*i.e.* $T_{\boldsymbol{\theta}}\mathcal{M} \subset \overline{\operatorname{Span}(NNTK_{\boldsymbol{\theta}}(\cdot, x) \ : \ x \in \Omega)}^{\perp\perp} = \overline{\operatorname{Span}(NNTK_{\boldsymbol{\theta}}(\cdot, x) \ : \ x \in \Omega)}$, which concludes. Now, since $T_{\boldsymbol{\theta}}\mathcal{M}$ is of finite dimension $d \leqslant P$, so is $\overline{\operatorname{Span}(NNTK_{\boldsymbol{\theta}}(\cdot, x) \ : \ x \in \Omega)}$, and since $(NNTK_{\boldsymbol{\theta}}(\cdot, x))_{x \in \Omega}$ is a generating family, one may extract a free subfamily out of it, which will be of cardinal $d \leqslant P$, *i.e.* there exist $d \leqslant P$ points $(\hat{x}_i)_{1 \leqslant i \leqslant d}$ such that $\widehat{T}_{\boldsymbol{\theta}}\mathcal{M} = \operatorname{Span}\left(NNTK_{\boldsymbol{\theta}}(\cdot, \hat{x}_i) \ : \ 1 \leqslant i \leqslant d\right) = T_{\boldsymbol{\theta}}\mathcal{M}$ and thus:

$$\Pi_{\widehat{T}_{\boldsymbol{\theta}}\mathcal{M}}^\perp \nabla\mathcal{L} = \Pi_{T_{\boldsymbol{\theta}}\mathcal{M}}^\perp \nabla\mathcal{L}.$$

If $d < P$, the sequence $(\hat{x}_i)_{1 \leqslant i \leqslant d}$ can be extended with additional $P - d$ arbitrary points. $\qquad\square$

Finally, in some cases, we have $\Pi_{T_{\boldsymbol{\theta}}^\perp\mathcal{M}}^\perp \left(\nabla\mathcal{L}_{|u_{|\boldsymbol{\theta}}}\right) = 0$ and thus $\widehat{\nabla\mathcal{L}}_{\boldsymbol{\theta}}^\perp = 0$, *i.e.* $\widehat{\nabla\mathcal{L}}_{\boldsymbol{\theta}}^\| = \widehat{\nabla\mathcal{L}}_{\boldsymbol{\theta}}$. This is the focus of Proposition 2, stated and proved hereafter:

**Proposition 2.** *If $u$ can be factorized as $u_{|\boldsymbol{\theta}} = L_{|\boldsymbol{\theta}_1}\left[C_{|\boldsymbol{\theta}_2}\right]$, with $\boldsymbol{\theta} = (\boldsymbol{\theta}_1, \boldsymbol{\theta}_2) \in \mathbb{R}^{P_1 + P_2}$, $C : \mathbb{R}^{P_2} \to \mathcal{H}_1$, $L : \mathbb{R}^{P_1} \to \mathcal{F}(\mathcal{H}_1 \to \mathcal{H})$ linear in $\boldsymbol{\theta}_1$, and $f = 0$, then $E_{\boldsymbol{\theta}}^\perp = 0$.*

*Proof.* From the discussion of Section 2.3, more precisely the identification of the Fréchet derivative of the functional loss in Equation (12), we have that the functional gradient for quadratic regression is:

$$\nabla\mathcal{L}_{u_{\boldsymbol{\theta}}} = u_{|\boldsymbol{\theta}} - f. \quad (87)$$

Assuming that $f = 0$, this reduces to:

$$\nabla\mathcal{L}_{u_{\boldsymbol{\theta}}} = u_{|\boldsymbol{\theta}}. \quad (88)$$

Now using the assumption and notations of Proposition 2, we see that:

$$\partial_{\boldsymbol{\theta}_1} u_{|\boldsymbol{\theta}}(\boldsymbol{\theta}_1) = \mathrm{d}L_{|\boldsymbol{\theta}_1}(\boldsymbol{\theta}_1)\left[C_{|\boldsymbol{\theta}_2}\right] = L_{|\boldsymbol{\theta}_1}\left[C_{|\boldsymbol{\theta}_2}\right] = u_{|(\boldsymbol{\theta}_1, \boldsymbol{\theta}_2)} = u_{|\boldsymbol{\theta}}, \quad (89)$$

where the second equality comes from the linearity of $L$ with respect to $\boldsymbol{\theta}_1$. In particular this implies that $u_{|\boldsymbol{\theta}} \in T_{\boldsymbol{\theta}}\mathcal{M}$, which concludes. $\qquad\square$

*Remark* 6. The question arises as to whether Proposition 2 has any concrete application, *i.e.* whether this situation occurs in real applications. As it happens, the hypothesis of Proposition 2 is verified in particular when solving the functional equation:

$$D[u] = 0, \tag{90}$$

using an MLP as parametric model $u$, with $D$ linear. Indeed, refering to the definition of an MLP in Section B.2, and specifically to the definition of the last layer in Equation (32), we see that MLP can be decomposed into $u_{|(\boldsymbol{\theta}_1, \boldsymbol{\theta}_2)} = L_{|\boldsymbol{\theta}_1}\left[C_{|\boldsymbol{\theta}_2}\right]$, with $\boldsymbol{\theta}_1$ being the parameters encoding the last layer. Now, forming the coumpound model defined in Equation (21) of Section 4.1 with the operator $D$ yields:

$$D \circ u = D \circ L_{|\boldsymbol{\theta}_1}\left[C_{|\boldsymbol{\theta}_2}\right] = \underbrace{D \circ L}_{=:L^D}{}_{|\boldsymbol{\theta}_1}\left[C_{|\boldsymbol{\theta}_2}\right] = L^D_{|\boldsymbol{\theta}_1}\left[C_{|\boldsymbol{\theta}_2}\right], \tag{91}$$

and thus the coumpound model is still verifying the assumption of Proposition 2. $f$ being null according to Equation (90), we have that all the hypotheses of the proposition are verified. In real-life applications, boundary conditions also need to be taken into account, as mentionned in Equation (21). However, these are a simple $L^2$ regression problem when the boundary conditions are Dirichlet, and therefore do not present the same conditioning difficulties as for regression with respect to the differential operator. This last fact, combined with Proposition 2, in our view partly explains the strong discrepancy between results for linear and non-linear problems.

In future work, we plan to carry out an in-depth analysis of the estimation of the $E_{\boldsymbol{\theta}}^{\text{metric}}$ and $E_{\boldsymbol{\theta}}^{\perp}$ terms, and their impact on both the overall theoretical framework and the training dynamics. We also aim to develop a more accurate method for approximating them.

## C.4 ADAPTING NATURAL GRADIENT DEFINITIONS TO PINNS

In Section 4.1, we highlighted the fact that PINNs can be understood as a classical quadratic regression problem, simply by substituting the model $u$ with the compound model $(D, B) \circ u$ introduced in:

$$(D, B) \circ u : \left\{ \begin{array}{ccccc} \mathbb{R}^P & \to & \mathcal{H} & \to & \mathbf{L^2}\left(\boldsymbol{\Omega}, \partial\boldsymbol{\Omega}\right) := L^2(\Omega \to \mathbb{R}, \mu) \times L^2(\partial\Omega \to \mathbb{R}, \sigma) \\ \boldsymbol{\theta} & \mapsto & u_{|\boldsymbol{\theta}} & \mapsto & (D[u_{|\boldsymbol{\theta}}], B[u_{|\boldsymbol{\theta}}]) \end{array} \right. . \tag{21}$$

We then pointed out that this simple remark made it possible to extend ANaGRAM, and more generally natural gradient, to the case of PINNs without further difficulty.

To make this adaptation more explicit, we compare in Table 17 below the objects introduced in Section 2 to define natural gradient and empirical natural gradient in classical quadradic regression context, with their equivalents for the PINNs case.

Natural neural tangent kernel and empirical tangent space pose certain technical difficulties and are therefore dealt with separately in Section C.5 below.

Table 17: Comparison of objects introduced in Section 2 for classical and PINNs regression.

| | Classical regression | PINNs regression |
|---|---|---|
| Image set of the model | $$\mathcal{M} := \operatorname{Im} u$$ $$= \{u_{|\boldsymbol{\theta}} : \boldsymbol{\theta} \in \mathbb{R}^P\}$$ $$\subset \mathrm{L}^2(\Omega)$$ | $$\Gamma := \operatorname{Im}\big((D,B) \circ u\big)$$ $$= \big\{\big(D[u_{|\boldsymbol{\theta}}], B[u_{|\boldsymbol{\theta}}]\big) : \boldsymbol{\theta} \in \mathbb{R}^P\big\}$$ $$\subset \mathrm{L}^2(\Omega, \partial\Omega) = \mathrm{L}^2(\Omega) \times \mathrm{L}^2(\partial\Omega)$$ |
| Model differential | $$\mathrm{d}u_{|\boldsymbol{\theta}} \in \mathcal{L}in\big(\mathbb{R}^P \to \mathrm{L}^2(\Omega)\big)$$ such that: $$u_{|\boldsymbol{\theta}+h} = u_{|\boldsymbol{\theta}}$$ $$+ \mathrm{d}u_{|\boldsymbol{\theta}}(h)$$ $$+ o(\|h\|_2)$$ | $$\mathrm{d}\big((D,B) \circ u\big)_{|\boldsymbol{\theta}} \in \mathcal{L}in\big(\mathbb{R}^P \to \mathrm{L}^2(\Omega, \partial\Omega)\big)$$ such that: $$\big((D,B)\circ u\big)_{|\boldsymbol{\theta}+h} = \big((D,B)\circ u\big)_{|\boldsymbol{\theta}}$$ $$+ \mathrm{d}\big((D,B)\circ u\big)_{|\boldsymbol{\theta}}(h)$$ $$+ o(\|h\|_2)$$ |
| Partial derivatives | $$\partial_p u_{|\boldsymbol{\theta}} = \mathrm{d}u_{|\boldsymbol{\theta}}(e^{(p)})$$ $$= \lim_{\varepsilon \to 0} \frac{u_{|\boldsymbol{\theta}+\varepsilon e^{(p)}} - u_{|\boldsymbol{\theta}}}{\varepsilon}$$ | $$\partial_p \big((D,B)\circ u\big)_{|\boldsymbol{\theta}} = \mathrm{d}\big((D,B)\circ u\big)_{|\boldsymbol{\theta}}(e^{(p)})$$ $$= \big(\partial_p D[u_{|\boldsymbol{\theta}}], \partial_p B[u_{|\boldsymbol{\theta}}]\big)$$ $$= \Big(\mathrm{d}D_{|u_{|\boldsymbol{\theta}}}\big(\partial_p u_{|\boldsymbol{\theta}}\big),$$ $$\mathrm{d}B_{|u_{|\boldsymbol{\theta}}}\big(\partial_p u_{|\boldsymbol{\theta}}\big)\Big)$$ |
| Tangent space | $$T_{\boldsymbol{\theta}}\mathcal{M} := \operatorname{Im} \mathrm{d}u_{|\boldsymbol{\theta}}$$ $$= \operatorname{Span}\big(\partial_p u_{|\boldsymbol{\theta}}\big)$$ $$= \Big\{\sum_{p=1}^{P} h_p \partial_p u_{|\boldsymbol{\theta}} :$$ $$(h_p) \in \mathbb{R}^P\Big\}$$ | $$T_{\boldsymbol{\theta}}\Gamma := \operatorname{Im} \mathrm{d}\big((D,B)\circ u\big)_{|\boldsymbol{\theta}}$$ $$= \operatorname{Span}\big(\partial_p D[u_{|\boldsymbol{\theta}}], \partial_p B[u_{|\boldsymbol{\theta}}]\big)$$ $$= \Big\{\sum_{p=1}^{P} h_p \big(\partial_p D[u_{|\boldsymbol{\theta}}], \partial_p B[u_{|\boldsymbol{\theta}}]\big) :$$ $$(h_p) \in \mathbb{R}^P\Big\}$$ |
| Functional loss | $$\mathcal{L} : \begin{cases} \mathrm{L}^2(\Omega) \to \mathbb{R}^+ \\ v \mapsto \frac{1}{2}\|v - f\|^2_{\mathrm{L}^2(\Omega)} \end{cases}$$ | $$\mathcal{L} : \begin{cases} \mathrm{L}^2(\Omega, \partial\Omega) \to \mathbb{R}^+ \\ v \mapsto \frac{1}{2}\|v - (f,g)\|^2_{\mathrm{L}^2(\Omega, \partial\Omega)} \end{cases}$$ |
| Functional gradient | $$\nabla \mathcal{L}_{\boldsymbol{\theta}} = \nabla \mathcal{L}_{|u_{|\boldsymbol{\theta}}}$$ $$= \underbrace{u_{|\boldsymbol{\theta}} - f}_{\in \mathrm{L}^2(\Omega)}$$ | $$\nabla \mathcal{L}_{\boldsymbol{\theta}} = \nabla \mathcal{L}_{|((D,B)\circ u)_{|\boldsymbol{\theta}}}$$ $$= \underbrace{\big((D,B)\circ u\big)_{|\boldsymbol{\theta}} - (f,g)}_{\in \mathrm{L}^2(\Omega, \partial\Omega)}$$ |
| Natural gradient | $$\mathrm{d}u_{|\boldsymbol{\theta}}^{\dagger}\big(\Pi^{\perp}_{T_{\boldsymbol{\theta}}\mathcal{M}}(\nabla\mathcal{L}_{\boldsymbol{\theta}})\big)$$ | $$\mathrm{d}\big((D,B)\circ u\big)_{|\boldsymbol{\theta}}^{\dagger}\big(\Pi^{\perp}_{T_{\boldsymbol{\theta}}\Gamma}(\nabla\mathcal{L}_{\boldsymbol{\theta}})\big)$$ |

## C.5 NATURAL NEURAL TANGENT KERNEL AND EMPIRICAL TANGENT SPACE OF PINNS

Seeing PINNs as a quadratic regression problem with respect to the coumpound model of Equation (21), as established in Section 4.1, we see that the "natural" defintion of NNTK that arises from Lemma 1 is: for all $\boldsymbol{\theta} \in \mathbb{R}^P$, for all $x, y \in (\Omega \times \partial\Omega)$

$$NNTK_{\boldsymbol{\theta}}(x,y) = \sum_{1 \leqslant p,q \leqslant P} \partial_p \left((D,B) \circ u\right)_{|\boldsymbol{\theta}} (x) G_{\boldsymbol{\theta}\,p,q}^{\dagger} \partial_q \left((D,B) \circ u\right)_{|\boldsymbol{\theta}} (y)^t.$$

with:

$$G_{\boldsymbol{\theta}} := \left\langle \partial_p \left((D,B) \circ u\right)_{|\boldsymbol{\theta}} , \partial_q \left((D,B) \circ u\right)_{|\boldsymbol{\theta}} \right\rangle_{\mathrm{L}^2(\Omega,\partial\Omega)} \tag{92}$$

The problem lies in the fact, that in order to define the empirical Tangent Space, we would like to be able to separate $\Omega$ and $\partial\Omega$ contributions. To do this, we have to remark that the coumpound model defined in Equation (21) outputs functions that have a two-dimensional output, *i.e.* the function is vector-valued and not scalar-valued anymore. More precisely, we have that for all $f \in \mathrm{Im}((D,B) \circ u) = \Gamma \subset \mathrm{L}^2(\Omega,\partial\Omega) = \mathrm{L}^2(\Omega \to \mathbb{R}) \times \mathrm{L}^2(\partial\Omega \to \mathbb{R})$, there exist $f_\Omega \in \mathrm{L}^2(\Omega \to \mathbb{R})$ and $f_{\partial\Omega} \in \mathrm{L}^2(\partial\Omega \to \mathbb{R})$ such that $f = (f_\Omega, f_{\partial\Omega})$. Thus for all $x = (x_\Omega, x_{\partial\Omega}) \in \Omega \times \partial\Omega$

$$f(x) = (f_\Omega(x_\Omega), f_{\partial\Omega}(x_{\partial\Omega})) \in \mathbb{R}^2. \tag{93}$$

Hence, the associated reproducing kernel should be a bit revisited. In particular the reproducing property rewrites (Alvarez et al., 2012, Section 3.2): for all $f = (f_\Omega, f_{\partial\Omega}) \in T_{\boldsymbol{\theta}}\Gamma \subset \mathrm{L}^2(\Omega,\partial\Omega)$, for all $x = (x_\Omega, x_{\partial\Omega}) \in \Omega \times \partial\Omega$ and for all $c \in \mathbb{R}^2$,

$$\langle f , NNTK_{\boldsymbol{\theta}}(\cdot,x)c \rangle = f(x)^T c = f_\Omega(x_\Omega)c_1 + f_{\partial\Omega}(x_{\partial\Omega})c_2. \tag{94}$$

In particular, we have: $f_\Omega(x_\Omega) = \left\langle f , NNTK_{|\boldsymbol{\theta}}(\cdot,x)e^{(1)} \right\rangle$ ; $f_{\partial\Omega}(x_{\partial\Omega}) = \left\langle f , NNTK_{|\boldsymbol{\theta}}(\cdot,x)e^{(2)} \right\rangle$. This means that the contributions coming from $\Omega$ and $\partial\Omega$ are linearly independent and can therefore be separated. More precisely, this can be done by defining the partial $NNTKs$:

- for all $y \in \Omega \times \partial\Omega$, for all $x_\Omega \in \Omega$, for all $x_{\partial\Omega} \in \partial\Omega$:

$$NNTK_{|\boldsymbol{\theta}}^{\Omega}(y, x_\Omega) := NNTK_{|\boldsymbol{\theta}}(y, (x_\Omega, x_{\partial\Omega}))e^{(1)} \tag{95}$$

$$= \sum_{1 \leqslant p,q \leqslant P} \partial_p((D,B) \circ u)_{|\boldsymbol{\theta}}(y) G_{\boldsymbol{\theta}\,p,q}^{\dagger} \partial_q(D \circ u)_{|\boldsymbol{\theta}}(x_\Omega), \tag{96}$$

- for all $y \in \Omega \times \partial\Omega$, for all $x_{\partial\Omega} \in \partial\Omega$, for all $x_\Omega \in \Omega$:

$$NNTK_{|\boldsymbol{\theta}}^{\partial\Omega}(y, x_{\partial\Omega}) := NNTK_{|\boldsymbol{\theta}}(y, (x_\Omega, x_{\partial\Omega}))e^{(2)} \tag{97}$$

$$= \sum_{1 \leqslant p,q \leqslant P} \partial_p((D,B) \circ u)_{|\boldsymbol{\theta}}(y) G_{\boldsymbol{\theta}\,p,q}^{\dagger} \partial_q(B \circ u)_{|\boldsymbol{\theta}}(x_{\partial\Omega}), \tag{98}$$

Then in particular: for all $x = (x_\Omega, x_{\partial\Omega}) \in \Omega \times \partial\Omega$, for all $f \in T_{\boldsymbol{\theta}}\Gamma$

- $f_\Omega(x_\Omega) = \left\langle f , NNTK_{|\boldsymbol{\theta}}^{\Omega}(\cdot, x_\Omega) \right\rangle = \left\langle f , NNTK_{|\boldsymbol{\theta}}(\cdot,x)e^{(1)} \right\rangle$.

- $f_{\partial\Omega}(x_{\partial\Omega}) = \left\langle f , NNTK_{|\boldsymbol{\theta}}^{\partial\Omega}(\cdot, x_{\partial\Omega}) \right\rangle = \left\langle f , NNTK_{|\boldsymbol{\theta}}(\cdot,x)e^{(2)} \right\rangle$.

This allows us to define an empirical tangent space for PINNs in the same way as in Equation (17), namely: Given two batches $(x_i^\Omega) \in \Omega^{S_\Omega}$ and $(x_i^{\partial\Omega}) \in \partial\Omega^{S_{\partial\Omega}}$, we define the associated empirical tangent space:

$$\widehat{T}_{\boldsymbol{\theta},(x_i^\Omega),(x_j^{\partial\Omega})}^{NNTK} := \mathrm{Span} \left( NNTK_{|\boldsymbol{\theta}}^{\Omega}(\cdot, x_i^\Omega), NNTK_{|\boldsymbol{\theta}}^{\partial\Omega}(\cdot, x_j^{\partial\Omega}) : 1 \leqslant i \leqslant S_\Omega, 1 \leqslant j \leqslant S_{\partial\Omega} \right) \tag{99}$$

Empirical Natural Gradient and associated ANaGRAM derivation poses then no particular difficulty and are done in a similar way to Section C.3.

To make this more concrete, we plot below some NTK and NNTK for the heat equation and the 2D Laplace equation. What we observe is that the NNTK yields a much more specialized kernel that

NTK, in the sense that it is much more localized and also perfectly centered on the reference points. This localization property is expected to be more pronounced as the complexity of the model is increased. The drastic discrepancy observed between NTK and NNTK explain why PINNs fail to train under classical descent while empirical natural gradient solves this issue. The excellent locality of the NNTK kernel leads indeed to a much better optimization schema because the residues are indeed arbitrarily shrunk in small regions around each sample batch point by independent modifications of the function which is obviously impossible with the NTK. When increasing the complexity of the model the spatial range of the NNTK is expected to decrease accordingly and then more points will be needed to "percolate" the optimization over the domain. In such case if the batch size increases too much various principled strategies can be considered to control the complexity of the empirical natural gradient, precisely because the NNTK defines a natural distance between points which can be leverage to define an approximate block Gram matrix. In our point of view this is one of the great advantages of the empirical tangent space over the "parameters" tangent space, because no such good metric is given for free in parameters space.

### C.5.1    NTK AND NNTK PLOTS OF LAPLACE 2 D EQUATION

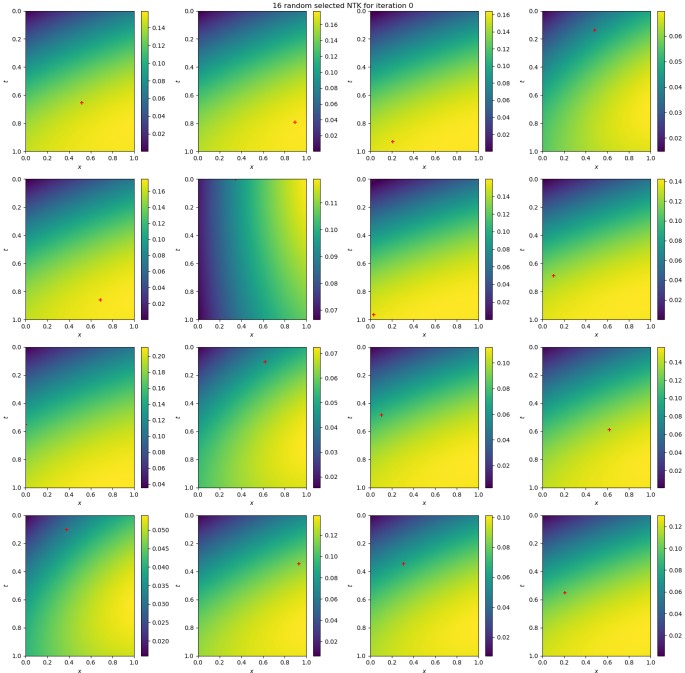

Figure 11: NTK at initialization for Laplace equation in 2 D. Reading: the red cross on each subfigure representing a point $x_i$, the plot represents the function $NTK_{\boldsymbol{\theta}_0}(\cdot, x_i)$

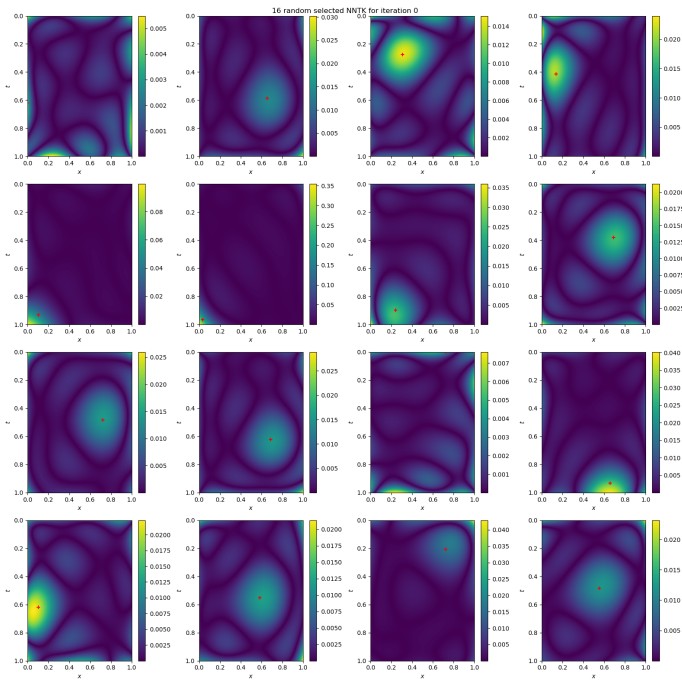

Figure 12: NNTK at initialization for Laplace equation in 2 D. Reading: the red cross on each subfigure representing a point $x_i$, the plot represents the function $NNTK_{\boldsymbol{\theta}_0}(\cdot, x_i)$

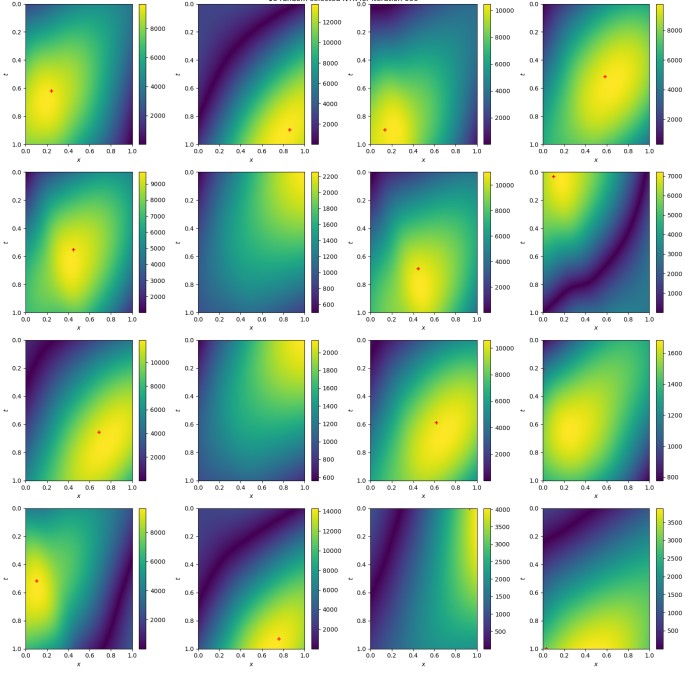

Figure 13: NTK at the end of optimization for Laplace equation in 2 D. Reading: the red cross on each subfigure representing a point $x_i$, the plot represents the function $NTK_{\boldsymbol{\theta}_{\mathrm{end}}}(\cdot, x_i)$

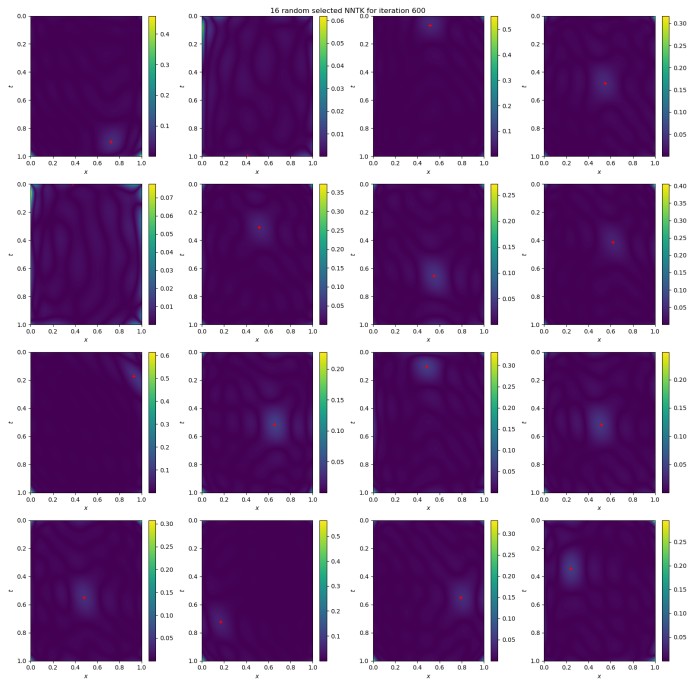

Figure 14: NNTK at the end of optimization for Laplace equation in 2 D. Reading: the red cross on each subfigure representing a point $x_i$, the plot represents the function $NNTK_{\boldsymbol{\theta}_{\text{end}}}(\cdot, x_i)$

### C.5.2 NTK AND NNTK PLOTS OF 1+1 D HEAT EQUATION

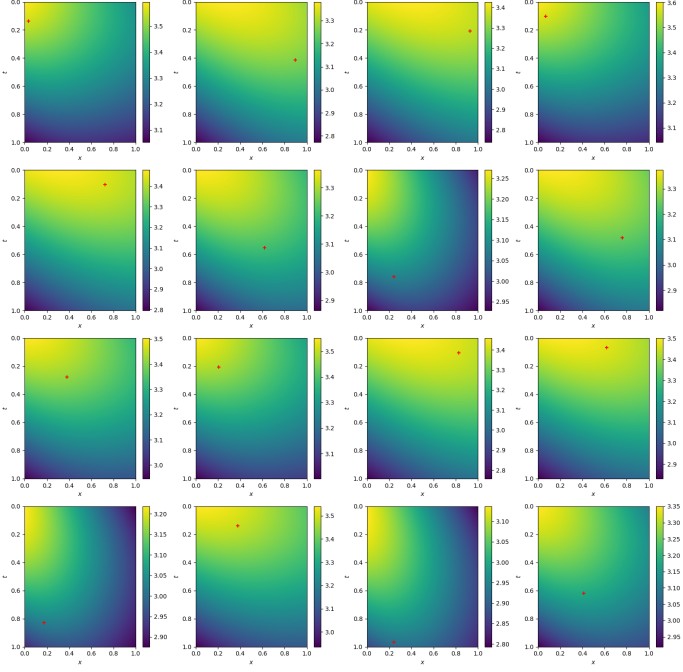

Figure 15: NTK at initialization for Heat equation in 1+1 D. Reading: the red cross on each subfigure representing a point $x_i$, the plot represents the function $NTK_{\boldsymbol{\theta}_0}(\cdot, x_i)$

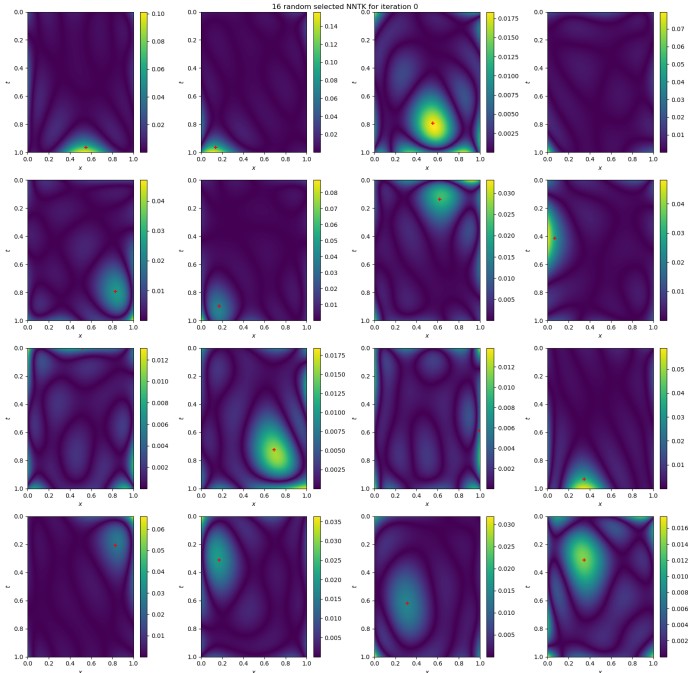

Figure 16: NNTK at initialization for Heat equation in 1+1 D. Reading: the red cross on each subfigure representing a point $x_i$, the plot represents the function $NNTK_{\boldsymbol{\theta}_0}(\cdot, x_i)$

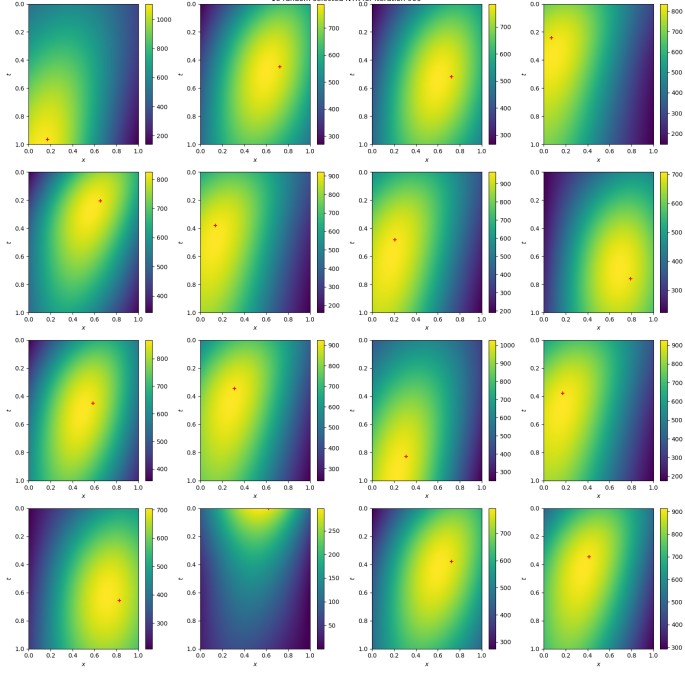

Figure 17: NTK at the end of optimization for Heat equation in 1+1 D. Reading: the red cross on each subfigure representing a point $x_i$, the plot represents the function $NTK_{\boldsymbol{\theta}_{\mathrm{end}}}(\cdot, x_i)$

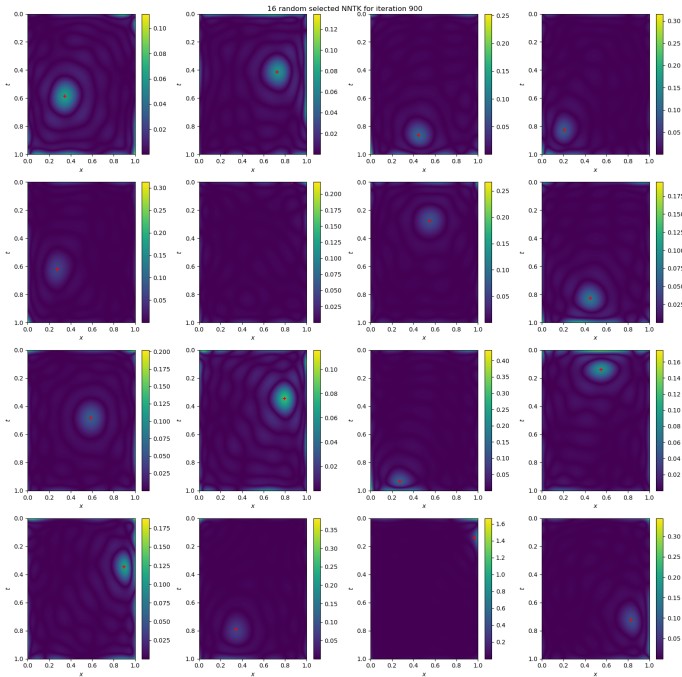

Figure 18: NNTK at the end of optimization for Heat equation in 1+1 D. Reading: the red cross on each subfigure representing a point $x_i$, the plot represents the function $NNTK_{\boldsymbol{\theta}_{\mathrm{end}}}(\cdot, x_i)$

## D    CONNECTION OF NATURAL GRADIENT OF PINNS TO GREEN'S FUNCTION

In this section, we will establish the relationship between Natural Gradient and Green's function. To set the stage, let us first introduce some definitions.

### D.1    PRELIMINARY DEFINITIONS

**Definition 7** (Green's function). Let $D : \mathcal{H} \to \mathrm{L}^2(\Omega \to \mathbb{R}, \mu)$ be a linear differential operator. A Green's function of $D$ is then any kernel function $g : \Omega \times \Omega \to \mathbb{R}$ such that the operator:

$$R : \begin{cases} D[\mathcal{H}] & \to & \mathcal{H} \\ f & \mapsto & \left( x \in \Omega \mapsto \int_\Omega g(x,s) f(s) \mu(\mathrm{d}s) \right) \end{cases} , \tag{100}$$

is a right-inverse to $D$, *i.e.* such that $D \circ R = I_{\mathcal{H}}$.

*Remark* 7.  We can rephrase Definition 7, by saying that $g$ is a Green's function if: for all $x, s \in \Omega$

$$D[g(\cdot, s)](x) = \delta_x(s),$$

where $\delta_x$ is the Dirac's distribution centered in $x$.

*Remark* 8.  Definition 7 implies that, $D[u] = f \in D[\mathcal{H}]$ is solved by $u(x) := \int_\Omega g(x,s) f(s) \mu(\mathrm{d}s)$.

In order to obtain more meaningful results, we will need the following generalizations:

**Definition 8** (Solution in the least-squares sense). Let $D : \mathcal{H} \to \mathrm{L}^2(\Omega \to \mathbb{R}, \mu)$ be a linear differential operator, $\mathcal{H}_0 \subset \mathcal{H}$ a subspace isometrically embedded in $\mathcal{H}$ and $f \in \mathrm{L}^2(\Omega \to \mathbb{R}, \mu)$. We call $u_0 \in \mathcal{H}_0$ a solution of the equation $D[u] = f$ in the least-squares sense if $u_0$ verifies:

$$\|D[u_0] - f\|^2_{\mathrm{L}^2(\Omega \to \mathbb{R}, \mu)} = \inf_{u \in \mathcal{H}_0} \|D[u] - f\|^2_{\mathrm{L}^2(\Omega \to \mathbb{R}, \mu)} \tag{101}$$

*Remark* 9.  If $f \in D[\mathcal{H}_0]$, then $\inf_{u \in \mathcal{H}_0} \|D[u] - f\|^2_{\mathrm{L}^2(\Omega \to \mathbb{R}, \mu)} = 0$ and thus $u_0$ is a classical solution.

**Definition 9** (generalized Green's function). Let $D, \mathcal{H}_0$ and $f$ be as in Definition 8. A generalized Green's function of $D$ on $\mathcal{H}_0$ is then any kernel function $g : \Omega \times \Omega \to \mathbb{R}$ such that the operator:

$$R_{\mathcal{H}_0} : \begin{cases} \mathrm{L}^2(\Omega \to \mathbb{R}, \mu) & \to & \mathcal{H} \\ f & \mapsto & \left( x \in \Omega \mapsto \int_\Omega g(x,s) f(s) \mu(\mathrm{d}s) \right) \end{cases},$$

verifies the equation:

$$D \circ R_{\mathcal{H}_0} = \Pi^\perp_{D[\mathcal{H}_0]} \tag{102}$$

*Remark* 10. Due to the identity $\Pi^\perp_{D[\mathcal{H}_0] | D[\mathcal{H}_0]} = I_{\mathcal{H}_0}$, Definition 9 is indeed a generalization of Definition 7.

We will now state and prove a result required to prove Theorem 2.

### D.2 PROOF OF THEOREM 2

**Proposition 3.** *Let $D : \mathcal{H} \to L^2(\Omega \to \mathbb{R}, \mu)$ be a linear differential operator, and $\mathcal{H}_0 := \mathrm{Span}(u_p : 1 \leqslant p \leqslant P) \subset \mathcal{H}$ a subspace isometrically embedded in $\mathcal{H}$. Then the generalized Green's function of $D$ on $\mathcal{H}_0$ is given by: for all $x, y \in \Omega$*

$$g_{\mathcal{H}_0}(x,y) := \sum_{1 \leqslant p,q \leqslant P} u_p(x) \, G^\dagger_{p,q} D[u_q](y), \tag{103}$$

*with: for all $1 \leqslant p, q \leqslant P$,*

$$G_{p,q} := \langle D[u_p], D[u_q] \rangle_{L^2(\Omega \to \mathbb{R}, \mu)} . \tag{104}$$

*Proof.* By definition:

$$D[\mathcal{H}_0] = \{D[u] \; : \; u \in \mathcal{H}_0\} = \mathrm{Span}\left( D[u_p] \; : \; 1 \leqslant p \leqslant P \right),$$

Thus Lemma 1 applies and yields that: for all $x, y \in \Omega$

$$k(x,y) := \sum_{p,q \in \mathbb{N}} D[u_p](x) \, G^\dagger_{p,q} D[u_q](y), \tag{105}$$

with: for all $1 \leqslant p, q \leqslant P$,

$$G_{ij} := \langle D[u_p], D[u_q] \rangle_{\mathrm{L}^2(\Omega \to \mathbb{R}, \mu)}, \tag{106}$$

is the kernel of the projection $\Pi^\perp_{D[\mathcal{H}_0]} : \mathrm{L}^2(\Omega \to \mathbb{R}, \mu) \to \mathrm{L}^2(\Omega \to \mathbb{R}, \mu)$ onto the RKHS $D[\mathcal{H}_0]$. This means that: $\forall f \in \mathrm{L}^2(\Omega \to \mathbb{R}, \mu)$

$$\Pi^\perp_{D[\mathcal{H}_0]}(f)(x) = \int_\Omega k(x,y) f(y) \, \mu(\mathrm{d}y) = \int_\Omega \sum_{1 \leqslant p,q \leqslant P} D[u_p](x) \, G^\dagger_{p,q} D[u_q](y) f(y) \, \mu(\mathrm{d}y) \tag{107}$$

$$= \sum_{1 \leqslant p,q \leqslant P} D[u_p](x) \, G^\dagger_{p,q} \int_\Omega D[u_q](y) f(y) \, \mu(\mathrm{d}y) \tag{108}$$

$$= D \left[ \sum_{1 \leqslant p,q \leqslant P} u_p(\cdot) \, G^\dagger_{p,q} \int_\Omega D[u_q](y) f(y) \, \mu(\mathrm{d}y) \right](x), \tag{109}$$

where Equation (109) comes from the linearity of $D$. Refering to Definition 9, this exactly means that the kernel defined by: for all $x, y \in \Omega$

$$g_{\mathcal{H}_0}(x,y) := \sum_{p,q} u_p(x) \, G^\dagger_{p,q} D[u_q](y)$$

is the generalized Green's function of the operator $D$ on $\mathcal{H}_0$. $\qquad \square$

We are now in a position to present the proof of:

**Theorem 2.** *Let $D : \mathcal{H} \to L^2(\Omega \to \mathbb{R}, \mu)$ be a linear differential operator and $u : \mathbb{R}^P \to \mathcal{H}$ a parametric model. Then for all $\boldsymbol{\theta} \in \mathbb{R}^P$, the generalized Green's function of $D$ on $T_{\boldsymbol{\theta}}\mathcal{M} = \operatorname{Im} du_{|\boldsymbol{\theta}}$ is given by: for all $x, y \in \Omega$*

$$g_{T_{\boldsymbol{\theta}}\mathcal{M}}(x,y) := \sum_{1 \leqslant p,q \leqslant P} \partial_p u_{|\boldsymbol{\theta}}(x) \, G_{p,q}^{\dagger} \partial_q D[u_{|\boldsymbol{\theta}}](y), \tag{22}$$

*with: for all $1 \leqslant p, q \leqslant P$*

$$G_{pq} := \left\langle \partial_p D[u_{|\boldsymbol{\theta}}], \, \partial_q D[u_{|\boldsymbol{\theta}}] \right\rangle_{L^2(\Omega \to \mathbb{R}, \mu)}. \tag{23}$$

*In particular, the natural gradient of PINNs[9] can be rewritten:*

$$\boldsymbol{\theta}_{t+1} \leftarrow \boldsymbol{\theta}_t - \eta \, du_{|\boldsymbol{\theta}_t}^{\dagger} \left( x \in \Omega \mapsto \int_{\Omega} g_{T_{\boldsymbol{\theta}_t}\mathcal{M}}(x,y) \nabla \mathcal{L}_{|\boldsymbol{\theta}_t}(y) \mu(dy) \right), \tag{24}$$

*Proof.* This a simple application of Proposition 3 to the space $\mathcal{H}_0 = \operatorname{Span} \left( \partial_p u_{|\boldsymbol{\theta}} : 1 \leqslant p \leqslant P \right) = \operatorname{Im} du_{|\boldsymbol{\theta}} = T_{\boldsymbol{\theta}}\mathcal{M}$. To conclude with the proof of Equation (24), let us note that Equation (102) can be rewritten as:

$$R_{\mathcal{H}_0} = D_{|\mathcal{H}_0}^{\dagger} \circ \Pi_{D[\mathcal{H}_0]}^{\perp}. \tag{110}$$

Specifically, we have:

$$\mathrm{d}\big(D \circ u\big)_{|\boldsymbol{\theta}}^{\dagger} \circ \Pi_{D[\mathcal{H}_0]}^{\perp} = du_{|\boldsymbol{\theta}}^{\dagger} \circ D^{\dagger} \circ \Pi_{D[\mathcal{H}_0]}^{\perp} = du_{|\boldsymbol{\theta}}^{\dagger} \circ R_{\mathcal{H}_0} \tag{111}$$

Since, by definition, $R_{\mathcal{H}_0}$ is the operator associated with the Green's function $g_{\mathcal{H}_0}$, this directly implies that the natural gradient of PINNs:

$$\boldsymbol{\theta}_{t+1} \leftarrow \boldsymbol{\theta}_t - \eta \, \mathrm{d}\big((D, B) \circ u\big)_{|\boldsymbol{\theta}_t}^{\dagger} \left( \Pi_{T_{\boldsymbol{\theta}_t}\Gamma}^{\perp} \left( \nabla \mathcal{L}_{\boldsymbol{\theta}_t} \right) \right),$$

can be expressed as:

$$\boldsymbol{\theta}_{t+1} \leftarrow \boldsymbol{\theta}_t - \eta \, du_{|\boldsymbol{\theta}_t}^{\dagger} \left( x \in \Omega \mapsto \int_{\Omega} g_{T_{\boldsymbol{\theta}_t}\mathcal{M}}(x,y) \nabla \mathcal{L}_{|\boldsymbol{\theta}_t}(y) \mu(\mathrm{d}y) \right),$$

which concludes. $\square$

### D.3 A practical example : Derivation of generalized Green's function for Laplacian operator, based on PINN's natural gradient formulation on a Fourier's basis

We will illustrate Theorem 2 on a parametric model given by partial Fourier's series (*cf.* Section B.1) for the Laplace operator on $[0,1]^d$:

$$\Delta : \begin{cases} \mathrm{H}^2([0,1]^d \to \mathbb{C}) & \to & \mathrm{L}^2([0,1]^d \to \mathbb{C}) \\ v & \mapsto & \sum_{l=1}^{d} \partial_{ll} v \end{cases}$$

For this purpose, let us then fix $N \in \mathbb{N}$ and consider the associated partial Fourier's Serie $S_N$ as defined in Equation (30):

$$S_N : \begin{cases} \mathbb{R}^{[\![-N,N]\!]^d} & \to & \mathrm{L}^2([0,1]^d \to \mathbb{C}) \\ (\alpha_{k_1,\ldots,k_d}) & \mapsto & \left( x \in [0,1]^d \mapsto \sum_{k_1=-N}^{N} \cdots \sum_{k_d=-N}^{N} \alpha_{k_1,\ldots,k_d} e^{2i\pi\left(\sum_{l=1}^{d} k_l x_l\right)} \right) \end{cases}.$$

We will then derive according to Theorem 2 the generalized Green's function of $\Delta$ on the tangent space of $S_N$ defined in Equation (31), namely:

$$\mathcal{M} = T_{\boldsymbol{\theta}}\mathcal{M} = \operatorname{Span} \left( x \in [0,1]^d \mapsto e^{2i\pi\left(\sum_{l=1}^{d} k_l x_l\right)} \, : \, k \in [\![-N,N]\!]^d \right)$$

---

[9]*cf.* last line of Table 17 in Section C.4.

To this end, let define: for all $k \in [\![-N, N]\!]^d$

$$e_k : \begin{cases} [0,1]^d & \to & \mathbb{C} \\ x & \mapsto & e^{2i\pi\left(\sum_{l=1}^d k_l x_l\right)} \end{cases} \tag{112}$$

and compute: for all $k \in [\![-N, N]\!]^d$, for all $1 \leqslant m \leqslant d$

$$\frac{\partial^2}{\partial x_m^2} e_k = -(2\pi)^2 k_m^2 e_k,$$

then: for all $k \in [\![-N, N]\!]^d$

$$\Delta e_k = -(2\pi)^2 \left(\sum_{m=1}^d k_m^2\right) e_k,$$

and thus: for all $k_1, k_2 \in [\![-N, N]\!]^d$

$$G_{k_1, k_2} := \langle \Delta e_{k_1}, \Delta e_{k_2} \rangle = (2\pi)^4 \left(\sum_{m=1}^d k_{1m}^2\right)^2 \delta_{k_1, k_2},$$

where $\delta_{k_1, k_2}$ is the Kronecker symbol such that $\delta_{k_1, k_2} = 1$ if and only if $k_1 = k_2$. This implies that:

$$G^\dagger_{k_1, k_2} = (2\pi)^{-4} \left(\sum_{m=1}^d k_{1m}^2\right)^{-2} (1 - \delta_{k_1, 0}) \delta_{k_1, k_2},$$

yielding:

$$g_{T_{\boldsymbol\theta}\mathcal{M}}(x,y) = \sum_{k_1, k_2 \in [\![-N,N]\!]^d} e_{k_1}(x) G^\dagger_{k_1, k_2} \overline{\Delta e_{k_2}}(y)$$

$$= \sum_{k_1, k_2 \in [\![-N,N]\!]^d} e_{k_1}(x)(2\pi)^{-4} \left(\sum_{m=1}^d k_{1m}^2\right)^{-2} (1 - \delta_{k_1,0}) \delta_{k_1,k_2} \left(-(2\pi)^2 \left(\sum_{m=1}^d k_{2m}^2\right) \overline{e_{k_2}}(y)\right)$$

$$= \frac{-1}{(2\pi)^2} \sum_{k \in [\![-N,N]\!]^d \setminus \{0\}} \frac{e_k(x) \overline{e_k}(y)}{\sum_{m=1}^d k_m^2} = \frac{-1}{(2\pi)^2} \sum_{k \in [\![-N,N]\!]^d \setminus \{0\}} \frac{\exp\left(2i\pi\left(\sum_{l=1}^d k_l(x_l - y_l)\right)\right)}{\sum_{l=1}^d k_l^2}$$

Finally, in order to add some consistency to our illustration, let us show for $d = 1$ that in the limit $N \to \infty$, we indeed find the classical Green's function of the Laplacian operator. In this case let us first remark, that we have:

$$g_{T_{\boldsymbol\theta}\mathcal{M}}(x,y) = \frac{-1}{(2\pi)^2} \sum_{k \in [\![-N,N]\!] \setminus \{0\}} \frac{\exp\left(2i\pi k(x - y)\right)}{k^2} = \frac{-1}{2\pi^2} \sum_{k=1}^N \frac{\cos(2\pi k(x-y))}{k^2}.$$

Since $|\cos(2\pi k(x - y))| \leqslant 1$, the serie is absolutely convergent and thus in the limit $N \to \infty$, we have:

$$g_\infty(x,y) := \lim_{N \to \infty} g_{T_{\boldsymbol\theta}\mathcal{M}}(x,y) = \frac{-1}{2\pi^2} C_2(2\pi(x-y)), \text{ with } C_2(z) = \sum_{k=1}^{+\infty} \frac{\cos(kz)}{k^2}$$

From Abramowitz and Stegun (1968, page 1005), we know that for all $z \in (0, 2\pi)$:

$$C_2(z) = \frac{\pi^2}{6} - \frac{\pi z}{2} + \frac{z^2}{4},$$

Thus, thanks to the parity of $C_2$ we have for all $(x - y) \in (0, 1)$:

$$g_\infty(x,y) = \frac{-1}{2\pi^2} \left(\frac{\pi^2}{6} - \frac{\pi 2\pi |x-y|}{2} + \frac{(2\pi(x-y))^2}{4}\right) = -\frac{1}{12} + \frac{|x-y|}{2} - \frac{(x-y)^2}{2}.$$

which is indeed a Green function for the Laplacian in 1d.

## D.4 ILLUSTRATION OF ESTIMATED GREEN'S FUNCTION FOR PINNS

### D.4.1 GREEN'S FUNCTION PLOTS OF LAPLACE EQUATION IN 2 D

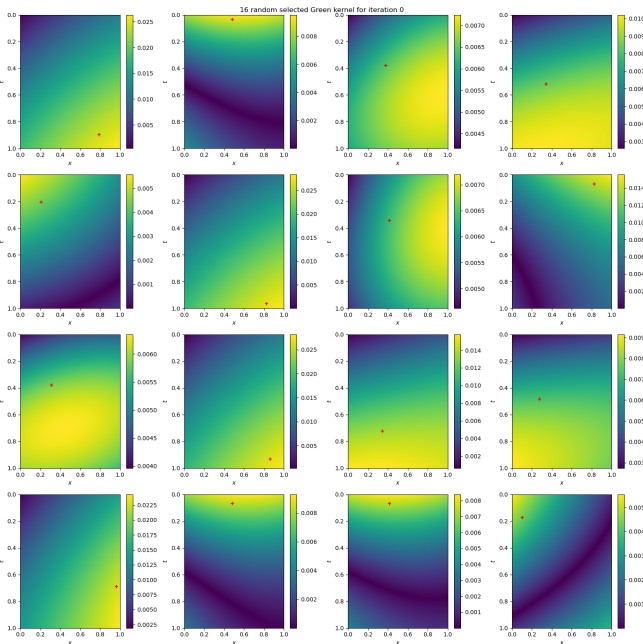

Figure 19: Green's function of the operator on the tangent space at initialization for Laplace equation in 2 D. Reading: the red cross on each subfigure representing a point $x_i$, the plot represents the function $g_{T_{\boldsymbol{\theta}_0}\mathcal{M}}(\cdot, x_i)$

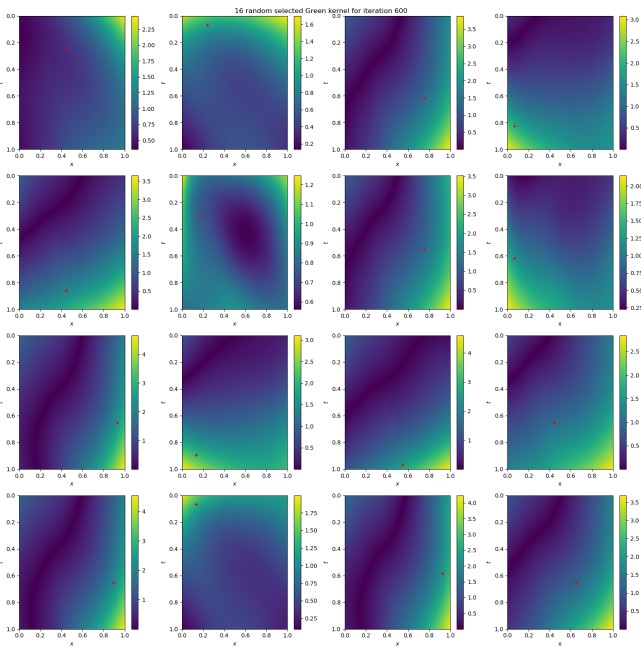

Figure 20: Green's function of the operator on the tangent space at the end of optimization for Laplace equation in 2 D. Reading: the red cross on each subfigure representing a point $x_i$, the plot represents the function $g_{T_{\boldsymbol{\theta}_{\mathrm{end}}}\mathcal{M}}(\cdot, x_i)$

### D.4.2 GREEN'S FUNCTION PLOTS OF HEAT EQUATION IN 1+1 D

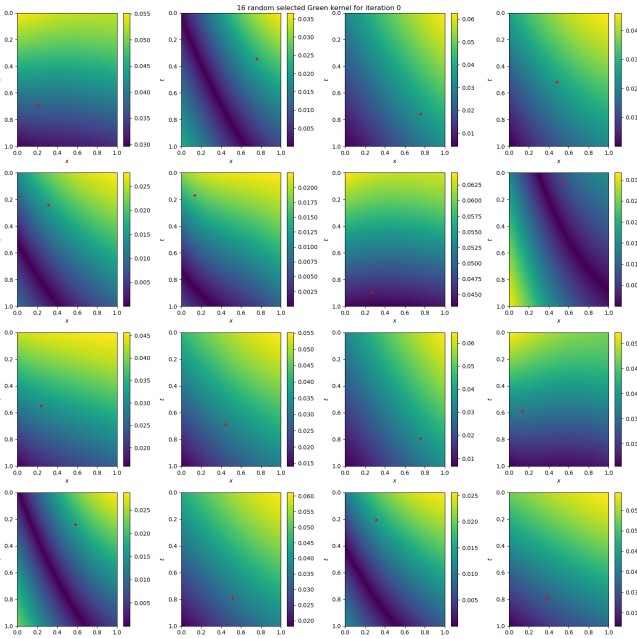

Figure 21: Green's function of the operator on the tangent space at initialization for Heat equation in 1+1 D. Reading: the red cross on each subfigure representing a point $x_i$, the plot represents the function $g_{T_{\boldsymbol{\theta}_0}\mathcal{M}}(\cdot, x_i)$

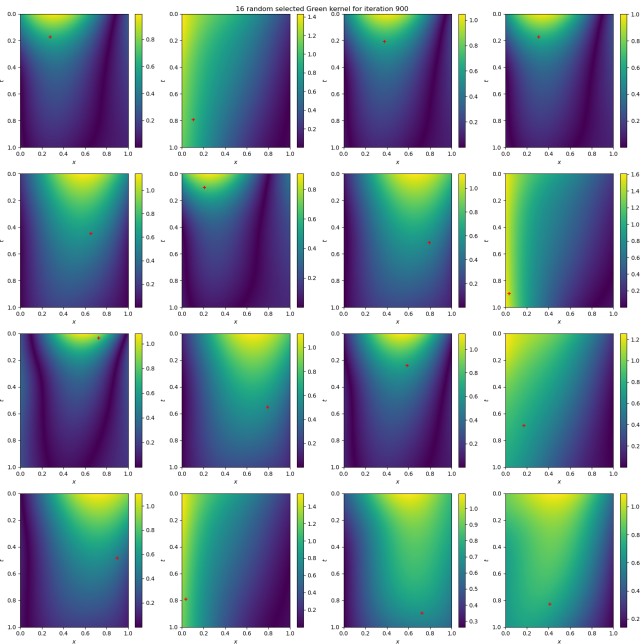

Figure 22: Green's function of the operator on the tangent space at the end of optimization for Heat equation in 1+1 D. Reading: the red cross on each subfigure representing a point $x_i$, the plot represents the function $g_{T_{\boldsymbol{\theta}_{\mathrm{end}}}\mathcal{M}}(\cdot, x_i)$

## E MATHEMATICAL EQUIVALENCE OF ANAGRAM AND ENERGY NATURAL GRADIENT IMPLEMENTATION (MÜLLER AND ZEINHOFER, 2023) FOR LINEAR OPERATORS

Let us begin by briefly recalling the definition of Energy Natural Gradient, as introduced in Müller and Zeinhofer (2023). Given an Energy (Equation 2 in Müller and Zeinhofer (2023)):

$$E(u) = \int_\Omega (\mathcal{D}[u] - f)^2 \mathrm{d}x + \tau \int_{\partial\Omega} (B[u] - g)^2 \mathrm{d}s, \tag{113}$$

associated to operators $D : \mathcal{H} \to \mathrm{L}^2(\Omega \to \mathbb{R}, \mu)$, $B : \mathcal{H} \to \mathrm{L}^2(\partial\Omega \to \mathbb{R}, \sigma)$, and a parametric model $u : \mathbb{R}^P \to \mathcal{H}$, we have the following definition (Definition 1 in Müller and Zeinhofer (2023)):

**Definition 10** (Energy Natural Gradient). Consider the problem $\min_{\boldsymbol{\theta} \in \mathbb{R}^P} L(\boldsymbol{\theta})$, where

$$L(\boldsymbol{\theta}) := E(u_{|\boldsymbol{\theta}}). \tag{114}$$

Denote the Euclidean gradient by $\nabla L(\boldsymbol{\theta})$. Then we call

$$\nabla^E L(\boldsymbol{\theta}) := G_E^\dagger(\boldsymbol{\theta}) \nabla L(\boldsymbol{\theta}), \tag{115}$$

the *energy natural gradient (E-NG)*, where $G_E$ is a Gram matrix defined by: for all $1 \leqslant p, q \leqslant P$

$$G_E(\boldsymbol{\theta})_{pq} := D^2 E(u_{\boldsymbol{\theta}})(\partial_{\boldsymbol{\theta}_p} u_{\boldsymbol{\theta}}, \partial_{\boldsymbol{\theta}_q} u_{\boldsymbol{\theta}}), \tag{116}$$

with $D^2 E$ being the second derivative of $E$ in the Fréchet sense.

As noted in Equation (9) of Müller and Zeinhofer (2023), in the case where $D$ and $B$ are linear, Equation (116) reduces to: for all $1 \leqslant p, q \leqslant P$

$$G_E(\boldsymbol{\theta})_{pq} = \int_\Omega D[\partial_{\boldsymbol{\theta}_p} u_{\boldsymbol{\theta}}](x) D[\partial_{\boldsymbol{\theta}_q} u_{\boldsymbol{\theta}}](x) \mathrm{d}x + \tau \int_{\partial\Omega} B[\partial_{\boldsymbol{\theta}_p} u_{\boldsymbol{\theta}}](s) B[\partial_{\boldsymbol{\theta}_q} u_{\boldsymbol{\theta}}](s) \mathrm{d}s, \tag{117}$$

Note that in this case, by setting $\tau = 1$ and taking $\mu$ and $\sigma$ uniform, this corresponds exactly to the Gram matrix: for all $1 \leqslant p, q \leqslant P$

$$G_{\boldsymbol{\theta} pq} := \left\langle \partial_p \big((D, B) \circ u\big)_{|\boldsymbol{\theta}}, \, \partial_q \big((D, B) \circ u\big)_{|\boldsymbol{\theta}} \right\rangle_{\mathrm{L}^2(\Omega, \partial\Omega)}, \tag{118}$$

where the compound model $(D, B) \circ u$ and the space $\mathrm{L}^2(\Omega, \partial\Omega)$ are those introduced in Equation (21). Now the key element lies in the sentence quoted from page 6, section 4.1 of Müller and Zeinhofer (2023) (which is confirmed in practice by the code):

> "The integrals in (15) are computed using the same collocation points as in the definition of the PINN loss function $L$ in (14)."

This means that the same set of collocation points is used to evaluate the integrals defining $L(\boldsymbol{\theta})$ in Equation (114) and the integrals defining $G_E(\boldsymbol{\theta})$ in Equation (117).

Let us fix some collocations points $(x_i^D)_{i=1}^{S_D}$ in $\Omega$, and $(x_i^B)_{i=1}^{S_B}$ in $\partial\Omega$. Then Equation (114) discretization exactly corresponds up to factor $\frac{1}{2}$, to the scalar loss of PINNs defined in Equation (7):

$$\ell(\boldsymbol{\theta}) := \frac{1}{2S_D} \sum_{i=1}^{S_D} \big(D[u_{|\boldsymbol{\theta}}](x_i^D) - f(x_i^D)\big)^2 + \frac{1}{2S_B} \sum_{i=1}^{S_B} \big(B[u_{|\boldsymbol{\theta}}](x_i^B) - g(x_i^B)\big)^2, .$$

The Euclidean gradient $\nabla L(\boldsymbol{\theta})$ is then approximated by the Euclidean gradient $\nabla\ell(\boldsymbol{\theta})$ given by: for all $1 \leqslant p \leqslant P$

$$\begin{aligned}
(\nabla\ell(\boldsymbol{\theta}))_p = &\frac{1}{S_D} \sum_{i=1}^{S_D} \partial_p D[u_{|\boldsymbol{\theta}}](x_i^D) \big(D[u_{|\boldsymbol{\theta}}](x_i^D) - f(x_i^D)\big) \\
&+ \frac{1}{S_B} \sum_{i=1}^{S_B} \partial_p B[u_{|\boldsymbol{\theta}}](x_i^B) \big(B[u_{|\boldsymbol{\theta}}](x_i^B) - g(x_i^B)\big)
\end{aligned} \tag{119}$$

Let us define:

- for all $1 \leqslant p \leqslant P$, for all $1 \leqslant i \leqslant S_D$, and for all $1 \leqslant j \leqslant S_B$:

$$\widehat{\phi}^D_{\boldsymbol{\theta}\,p,i} := \partial_p D[u_{|\boldsymbol{\theta}}](x^D_i); \qquad\qquad \widehat{\phi}^B_{\boldsymbol{\theta}\,p,j} := \partial_p B[u_{|\boldsymbol{\theta}}](x^B_j), \qquad (120)$$

- for all $1 \leqslant i \leqslant S_D$ and for all $1 \leqslant j \leqslant S_B$:

$$\widehat{\nabla\mathcal{L}}^D_{\boldsymbol{\theta}\,i} := D[u_{|\boldsymbol{\theta}}](x^D_i) - f(x^D_i); \qquad \widehat{\nabla\mathcal{L}}^B_{\boldsymbol{\theta}\,j} := B[u_{|\boldsymbol{\theta}}](x^B_j) - g(x^B_j), \qquad (121)$$

and finally:

$$\widehat{\phi}_{\boldsymbol{\theta}} := \begin{pmatrix} \widehat{\phi}^D_{\boldsymbol{\theta}}, & \widehat{\phi}^B_{\boldsymbol{\theta}} \end{pmatrix}; \qquad \widehat{\nabla\mathcal{L}}_{\boldsymbol{\theta}} := \begin{pmatrix} \widehat{\nabla\mathcal{L}}^D_{\boldsymbol{\theta}} \\ \widehat{\nabla\mathcal{L}}^B_{\boldsymbol{\theta}} \end{pmatrix}; \qquad \widehat{\Lambda} := \begin{pmatrix} \frac{1}{S_D}\boldsymbol{I}_{S_D} & 0 \\ 0 & \frac{1}{S_B}\boldsymbol{I}_{S_B} \end{pmatrix}. \qquad (122)$$

Thus Equation (119) can be rewritten as:

$$\nabla\ell(\boldsymbol{\theta}) = \widehat{\phi}_{\boldsymbol{\theta}}\widehat{\Lambda}\widehat{\nabla\mathcal{L}}_{\boldsymbol{\theta}}. \qquad (123)$$

But since the same set of collocation points is used to evaluate the integrals defining $G_E(\boldsymbol{\theta})$, we also have that Equation (116) is discretized by:

$$G_E(\boldsymbol{\theta}) \simeq \widetilde{G_E(\boldsymbol{\theta})} := \widehat{\phi}_{\boldsymbol{\theta}}\widehat{\Lambda}\widehat{\phi}^t_{\boldsymbol{\theta}}. \qquad (124)$$

This implies that Equation (115) is approximated by:

$$\nabla^E L(\boldsymbol{\theta}) \simeq \widetilde{\nabla^E L(\boldsymbol{\theta})} := \widetilde{G_E(\boldsymbol{\theta})}^{\dagger}\nabla\ell(\boldsymbol{\theta}) = \widehat{\phi}^{\dagger}_{\boldsymbol{\theta}}\widehat{\nabla\mathcal{L}}_{\boldsymbol{\theta}}, \qquad (125)$$

which corresponds indeed to the update direction in line 5 in ANaGRAM algorithm 2.

To conclude, it should be noted that when $D$ or $B$ are not linear, then the equivalence between Equation (117) and Equation (118) not longer holds, with the result that E-NGD and ANaGRAM are no longer equivalent either.

