# OpenReview forum: "ANaGRAM: A Natural Gradient Relative to Adapted Model for efficient PINNs learning"
_ICLR.cc/2025/Conference — ICLR 2025 Poster_

### Official Review · Reviewer_ogx7 · 2024-10-31

**Soundness:** 3
**Presentation:** 2
**Contribution:** 2
**Rating:** 3
**Confidence:** 4

**Summary:**

The manuscript proposes a natural gradient method related to the ENGD optimizer proposed by (Müller and Zeinhofer). The main contributions are:

- In the context of PINNs, the authors view the ansatz/parametric model as a neural network followed by the application of the PDE and boundary operator. This means that the model is defined on some parameter space $\Theta$, taking values in the product space $L^2(\Omega) \times L^2(\partial\Omega)$ via the intermediate space $\mathcal H$ which is related to the PDE and boundary operators of the specific problem. This mathematical trick of changing what is the model and what is the loss function allows to view PINNs as a standard regression problem. This viewpoint motivates the authors to use an $L^2(\Omega)\times L^2(\partial\Omega)$ based natural gradient method, which agrees with ENGD for linear PDEs but performs better than ENGD for nonlinear PDEs

- The second contribution is on the numerical linear algebra level. Instead of solving the normal equations of the regression, the authors propose to use an SVD directly for the Jacobian of the model. Here, when I write Jacobian I mean the discretized differential of the parametrization, which is denoted by $\hat \phi_{\theta(t)}$ in the manuscript. This gives an improvement in computational complexity from $\mathcal O(P^3)$ for solving the normal equations, to $\mathcal O (\min(N^2P, P^2N) )$, as the Jacobian is of shape $(N, P)$, where $P$ is the dimension of the parameter space $\Theta$ and $N$ is the number of sampling points.

**Strengths:**

- The viewpoint outlined above leads to a performant method, that works better than ENGD for nonlinear PDEs.

- The improvement in complexity for the linear solve is helpful.

**Weaknesses:**

- The viewpoint of absorbing the differential operators into the model can also be understood as a Gauss-Newton method in function space compare to (Jnini et al, arXiv:2402.10680). Moreover, it leads to the standard Gauss-Newton algorithm in parameter space. The manuscript thus does not provide a novel algorithm, but a novel derivation of a known algorithm. While I like this idea and it is certainly a contribution, I think it is not significant enough for an exceptional venue such as ICLR.

**Questions:**

- Can the authors add a reference for the computational complexity of the SVD for the rectangular solve?

- Why is the proposed method performing better than ENGD for the linear equations? I would expect similar performance as it is the same algorithm (up to numerical linear algebra).

- it is very common to include damping into natural gradient methods (adding a scaled identity term to the square Gramian). Can this be realized in this setting?

---

> ### Author Response · Authors · 2024-11-21
>
> Thank you very much for your careful reading, insightful comments and constructive critics. We address hereafter the weaknesses you have identified and answer your questions.
>
> > ## Weaknesses:
> > The viewpoint of absorbing the differential operators into the model can also be understood as a Gauss-Newton method in function space compare to (Jnini et al, arXiv:2402.10680). Moreover, it leads to the standard Gauss-Newton algorithm in parameter space.
>
> We thank you for this observation. We agree that the derivation provided in Jnini et al lead to similar algorithms but we would like to stress out some important differences:
> - Our algorithm, while presented in its simplest form for the reasons explained in the general comment, is more general than Gauss-Newton. It is identical to Gauss-Newton only when neglecting two terms in equation (18) of the new version, one given by (66) in appendix which is usually null for linear operators ; (a fact highlighted in proposition 2 in the revision), and the other term introduced in equation (75) which is amenable to various levels of approximations, for instance using Nyström's method, the lowest one consisting to simply discard it.
> - The choice of presenting only the simplified setting, coinciding with empirical Gauss-Newton was justified by the lack of competitive baseline in the literature. We thank the reviewer to point to our attention the paper by Jnini et al., but  if we want to set a comparison with it, first we remark that the methodology is different, because in their case the solving of the least square problem is done either through conjugate gradient or direct least square regression with respect to the estimated gram matrix, while in our case we use SVD. On one hand the first option, to our best knowledge, does not allow a precise control of the cutoff factor, something that is nonetheless crucial for robustness of the algorithm, as illustrated by the aforementioned discrepancy between ANaGRAM and ENGD. Hence, we also suspect that convergence of conjugate gradient may fail in some cases due to the strongly bad conditioning, and that an additional preconditioner may be needed. On the other hand, direct least square regression with respect to the estimated gram matrix has a cubic cost with respect to parameters number, which is precisely what we aim to avoid. In practice, those are conjectures since the code seems not to be made available sofar according to page 6 of the paper.
>
> > The manuscript thus does not provide a novel algorithm, but a novel derivation of a known algorithm. While I like this idea and it is certainly a contribution, I think it is not significant enough for an exceptional venue such as ICLR.
>
> We hope that the clarification of the theoretical framework, now introduced in section 3 and no longer in the appendix, will make a more meaningful contribution to you.
>
> > ## Questions
> >
> > Can the authors add a reference for the computational complexity of the SVD for the rectangular solve?
>
> We are not sure to understand this question since the computational complexity of (truncated) SVD is a classical textbook result of numerical linear algebra. One may for instance refer to Numerical linear algebra by LN Trefethen and D Bau, as suggested on the wikipedia page https://en.wikipedia.org/wiki/Singular_value_decomposition#Numerical_approach. Furthermore, recent works in numerical linear algebra bring new randomized algorithms that significantly improves the CPU time. We may for instance refer to the work of Professor Laura Grigori, in particular https://arxiv.org/abs/2302.07466 and https://arxiv.org/abs/2302.11474.
>
> > Why is the proposed method performing better than ENGD for the linear equations? I would expect similar performance as it is the same algorithm (up to numerical linear algebra).
>
> The difference presumably comes from the fact that we took into account the cutoff factor of the SVD as an hyperparameter, which is for now adjusted manually on the logscale. This is something that was not taken into account in ENGD, explaining the discrepancy of performance for linear operators and underlying the importance of this hyper-parameter. The automation of it is under development in conjunction with the choice of collocation points.
>
> > It is very common to include damping into natural gradient methods (adding a scaled identity term to the square Gramian). Can this be realized in this setting?
>
> Yes indeed. This can be done without further complication, just by replacing the term $\Delta^{\dagger}$ with $\frac{\Delta}{\Delta^2+\lambda}$, where $\lambda$ is the scaling factor.

---

> > ### Comment · Reviewer_ogx7 · 2024-11-22
> >
> > Thank you for the reply.
> >
> > - I understand that only in certain situations the algorithm reduces to Gauss-Newton in *parameter space*. What I was mainly referring to is an *infinite-dimensional viewpoint*. As detailed in Jnini et al, Gauss-Newton's method in infinite-dimensions can discretized in the tangent space of a neural network ansatz. If we ignore the subtleties of integral discretization (choice of sample points) I believe this is exactly what you are proposing: In Algorithm 2, take the matrix $\hat\phi_{\theta_t}$. Then
> > $\hat\phi_{\theta_t}^\top\hat\phi_{\theta_t}$ is the (Galerkin) discretization (in the tangent space) of $DR(w_k)^*DR(w_k)$ in equation (5) of Jnini et al.
> >
> > In fact, understanding the algorithm this way offers additional insight. As proven in Jnini et al, one knows that the discrete dynamics (for simplicity treating the population setting) is the projection of the function space dynamics on the tangent space. This is along the lines of your Green's function perspective.
> >
> > - I don't think a numerical comparison to Jnini et al is needed.
> >
> > - Unfortunately, Theorem 1 in your revised version is hard to read. What is $I_P?$ What is the intuition behind $I_\theta^\top$? In Proposition 2, what is $C$ and $L$?
> >
> > - I am still considering the improved scaling through the rectangular SVD solve as the main contribution of the paper. While the SVD realization is of course useful,  I reiterate that in my opinion it is not significant enough for an exceptional venue such as ICLR. I therefore prefer to keep my score as is.

---

> > > ### Author Response · Authors · 2024-11-28
> > >
> > > > I understand that only  in certain situations the algorithm reduces to Gauss-Newton in parameter space. What I was mainly referring to is ...     in equation (5) of Jnini et al.
> > >
> > > Thank you for your comment. This gives us the opportunity to clarify the main point of our paper which seems to have been missed. We acknowledge indeed the correspondence between our approach and the one presented in Jnini et al, in quadratic regression context. Nevertheless, we still think that our approach is different, or more precisely dual. Jnini et al. see the problem as a Galarkin discretization of the functional space problem (what you call infinite dimensional viewpoint) in the tangent space defined by the neural network. On the other hand, our primary focus is the discretization with respect to the points. And this framework is not bound to a "Galarkin discretization". Indeed, our main claim is the fact that we reconsider the problem from a kernel method perspective. With this in mind, we could perfectly apply our method using the theoretical kernel attached to the operator, as introduced for instance in https://arxiv.org/abs/2409.13786. In this case, the empirical tangent space is still defined in the same way as in our approach, replacing NNTK with the theoretical kernel. By taking a Galarkin discretization of the problem using a neural network, we are only projecting this theoretical kernel into the tangent space of the compound model, in line with classical results of RKHS.
> > > This approach is not possible using Jnini et al. point of view, since, while the resulting Gram matrix is indeed the same, due to the definition in Jnini et al. of the metric in the tangent space of the neural network, the resulting NNTK is completely different (namely $NNTK_\theta(x,y):=\sum_{1\leq p,q\leq P}\partial_p u_\theta(x) G^\dagger_{p,q} \partial_q u_\theta(y)$ in Jnini et al., with $G$ the common Gram matrix as in equation (14) of our paper) and has a priori no clear interpretation. In particular, defining an empirical tangent space with respect to this kernel seems meaningless for the problem. We have empirical evidences of this. Furthermore, Jnini et al approach does not generalize straightforwardly to non-quadratic losses, in contrary to our approach. Lastly it does not lead to any criterion for optimizing the collocation points.
> > >
> > > Now, the clear advantage of considering the points instead of the parameters, is that they are spatially organized and the NNTK provides us with a meaningful distance between those points. Hence it makes more sense if needed to group them by block than doing so with the parameters. This is in contrast with the many former options that have been proposed, and that usually deal with that by approximating the Gram matrix in block diagonal form, with in NN context one block per layer, hoping that the parameters influence each other only within layers. This assumption is particularly bad in PINNs context, since the functional structure is completely changed by the functional operator. This is well pictured by the NNTK plots, pictured at the end of section C.4. To the best of our knowledge this viewpoint has not been discussed nor formalized previously as well as the connection between NNTK and the Green function.
> > >
> > > >   I don't think a numerical comparison to Jnini et al is needed.
> > >
> > >  We don't understand this point. The fact is that sofar we have  no way to compare other than theoretically as discussed above.
> > >
> > > > Unfortunately, Theorem 1 in your revised version is hard to read. What is ..
> > >
> > > In last version of the paper  we  try to present this in a more transparent way. These two terms have a simple geometric interpretation: the first one corresponds to the contribution of the transverse part of the functional gradient w.r.t. the tangent space.
> > > The second term accounts for the miscalculation of the empirical tangent space caused by the limited information  on the metric provided by the batch points. The first can be well corrected with some known work (eg Cohen et al.) and vanish in linear cases. The second term can be more accurately taken into account, using more advanced Nyström methods . Furthermore this term vanishes when we take as much points as parameters. Both corrections can be better approximated without additional computational burden. The present paper sets the baseline for all these extensions of Anagram.
> > >
> > > >    I am still considering the improved scaling through the rectangular SVD solve as the main contribution of the paper...
> > >
> > > We note that the four points listed at the beginning of the rebuttal, which we presented to highlight the novelty of the paper, may not have been fully addressed. We believe this  stems from a misunderstanding of our work, which we hope to have clarified, at least partially, through this latest discussion.

---

> > > > ### Author Response · Authors · 2024-11-29
> > > >
> > > > In complement to our comment from yesterday, we provide at https://anonymous.4open.science/r/ANaGRAM-3815/complementary-empirical-evidences/Jnini_et_al-NNTK/ the NNTKs obtained from the derivation of Jnini et al., namely:
> > > > \begin{equation}
> > > > NNTK_\theta(x,y):=\sum_{1\leq p,q\leq P}\partial_p u_\theta(x) G^\dagger_{p,q} \partial_q u_\theta(y)
> > > > \end{equation}
> > > > for Heat and Laplace 2D equations. Those have to be compared respectively:
> > > > - https://anonymous.4open.science/r/ANaGRAM-3815/complementary-empirical-evidences/Jnini_et_al-NNTK/Laplace_2d/Jnini_NNTK_kernel_0.png with Figure 10 in the new manuscript.
> > > > - https://anonymous.4open.science/r/ANaGRAM-3815/complementary-empirical-evidences/Jnini_et_al-NNTK/Laplace_2d/Jnini_NNTK_kernel_600.png with Figure 12 in the new manuscript.
> > > > - https://anonymous.4open.science/r/ANaGRAM-3815/complementary-empirical-evidences/Jnini_et_al-NNTK/Heat/Jnini_NNTK_kernel_0.png with Figure 14 in the new manuscript.
> > > > - https://anonymous.4open.science/r/ANaGRAM-3815/complementary-empirical-evidences/Jnini_et_al-NNTK/Heat/Jnini_NNTK_kernel_600.png with Figure 16 in the new manuscript.

---

> > > > > ### Comment · Reviewer_ogx7 · 2024-12-03
> > > > >
> > > > > I very much appreciate your answer and effort.
> > > > >
> > > > > Thanks for the explanation with respect to the empirical tangent spaces and the kernel perspective, this is an interesting viewpoint and an addition to the purely functional viewpoint. However, you do not leverage this insight anywhere in the numerical experiments which makes your numerical method identical to the one in Jnini et al. and prevents you from making a (potentially strong) point by improving the discretization.
> > > > >
> > > > > Overall I like many of your ideas but they are not yet put into a coherent work and some crucial aspects are missing (leveraging numerically your insight into empirical tangent spaces). I therefore prefer to keep my score.

---

> > > > > > ### Author Response · Authors · 2024-12-03
> > > > > >
> > > > > > We thank the reviewer for the the time taken to review and discuss our paper, and for thoughtfully acknowledging  our theoretical contribution in addition to the algorithmic one (SVD trick). It is true that  we did not fully leverage the potential of our theoretical development, and this because as already said the vanilla version was already better with available baselines to which we could compare (which was not  possible with Jnini et al.).

---

### Official Review · Reviewer_iNHt · 2024-11-01

**Soundness:** 2
**Presentation:** 4
**Contribution:** 2
**Rating:** 3
**Confidence:** 4

**Summary:**

This paper studies the natural gradient descent, which is theoretically and empirically believed to improve the convergence of training PINNs. Theoretical analysis on the natural gradient and training PINNs bridges connections on Green function theory.

**Strengths:**

(1) The natural gradient descent approach is theoretically sound, offering a promising solution for improving the convergence of regression problems and PINNs. Notably, while gradient descent may struggle with PINNs involving nonlinear PDEs, natural gradient descent effectively addresses this limitation.


(2) The paper's presentation is clear and well-structured.

**Weaknesses:**

(1) The algorithm in the paper is not novel. The general natural GD method requires O(P^3) computational costs, due to the inversion of the Gram matrix of \hat{\phi} (G = \hat{\phi} \hat{\phi}^{T} is in size of P \times P). This paper applies SVD directly on \hat{\phi}. Moreover, the line 6 of the algorithm is exactly equivalent to the vanilla natural GD algorithm (rewriting the pseudo-inverse with the SVD). Therefore, I do not find the novelty of the algorithm.


(2) In Equations (18) and (19), it seems that the paper tries to explain the consistency by approximating the integral by finite sum of some observations. However, the results of Corollary 1 are relatively weak. The existence of certain points, as stated, follows naturally from standard regularity conditions (e.g., smoothness) of functions. The result is therefore not surprising and lacks substantial implications. It is better if the paper can alternatively prove that if we randomly pick P training samples as observations and with high probability, the difference between du and \hat{\phi} \hat{f} is upper bounded, e.g., O(1/P^{\alpha}), where \alpha is a constant depending on function’s conditions and dimensionality. Such findings would be more meaningful for machine learning.


(3) The ANaGRAM for PINNs closely resembles existing natural GD for training PINNs (https://arxiv.org/abs/2408.00573). For example, in Equation (31) of the paper, the pseudo-inverse of (J J^{T}) is equivalent to the SVD of \hat{\phi} in this paper.


(4) The natural GD requires the full-batch training, which limits its adaptability to stochastic algorithms. We know that the SGD-like algorithms are widely used in deep learning due to their benefits, such as improving the generalization and bypassing spurious local minima. A more practical approach would be one that can adapt effectively to a stochastic setting.

**Questions:**

Please see the weakness part.

One typo: Lines 126-127: ill conditioned

---

> ### Author Response · Authors · 2024-11-21
>
> Thank you very much for your careful reading, insightful comments and constructive critics. We address hereafter the weaknesses you have identified.
>
> > ## Weaknesses:
> > (1) The algorithm in the paper is not novel. The general natural GD method requires O(P^3) computational costs, due to the inversion of the Gram matrix of \hat{\phi} (G = \hat{\phi} \hat{\phi}^{T} is in size of P \times P). This paper applies SVD directly on \hat{\phi}. Moreover, the line 6 of the algorithm is exactly equivalent to the vanilla natural GD algorithm (rewriting the pseudo-inverse with the SVD). Therefore, I do not find the novelty of the algorithm.
> >
> > (2) In Equations (18) and (19), it seems that the paper tries to explain the consistency by approximating the integral by finite sum of some observations. However, the results of Corollary 1 are relatively weak. The existence of certain points, as stated, follows naturally from standard regularity conditions (e.g., smoothness) of functions. The result is therefore not surprising and lacks substantial implications. It is better if the paper can alternatively prove that if we randomly pick P training samples as observations and with high probability, the difference between du and \hat{\phi} \hat{f} is upper bounded, e.g., O(1/P^{\alpha}), where \alpha is a constant depending on function’s conditions and dimensionality. Such findings would be more meaningful for machine learning.
> >
> > (3) The ANaGRAM for PINNs closely resembles existing natural GD for training PINNs (https://arxiv.org/abs/2408.00573). For example, in Equation (31) of the paper, the pseudo-inverse of (J J^{T}) is equivalent to the SVD of \hat{\phi} in this paper.
>
> If we understand well, we may reformulate critics (1), (2) and (3) by saying that our approach is just vanilla natural gradient with poor statistics. This is a legitimate interrogation which is probably reinforced by the simplified presentation of the algorithm that we gave which is possibly misleading, especially (former) equation (19), since it gives the false impression that we use a non controlled statistical criteria, while in fact our approach is a geometric one, namely the projection of the functional gradient into the empirical tangent space. This is now made clear in the new version in section 3, in particular with th.1 and eq. (18). We have tested the minimal version of Anagram by boldly neglecting two terms in (18), which already yield state of the art results. More precise algorithms can be considered based for instance on Nyström method in order to take into account these terms without significant additional cost, but we leave this for future developments. As a byproduct, this means that the quality of the batch is not given by a statistical criterion but by a geometrical one given by equation (18) in new version.
>
> In the same spirit, former corollary 1 (now somewhat modified in proposition 1), which seems trivial, underlines this part, by showing that criterion of equation (18) can be saturated with  P points. The exploitation of this criterion is under investigation as it represents a non trivial problem to optimize (18) while training the PINNs.
>
> > (4) The natural GD requires the full-batch training, which limits its adaptability to stochastic algorithms. We know that the SGD-like algorithms are widely used in deep learning due to their benefits, such as improving the generalization and bypassing spurious local minima. A more practical approach would be one that can adapt effectively to a stochastic setting.
>
> As a follow up to previous answer, we believe that the introduction of the notion of empirical tangent space is a guide toward the adaption of natural gradient to a batched version. Various options are available, which are under investigation. We are aware that one of the  main difficulties is to estimate effectively the dependence between different empirical tangent spaces. Nonetheless, we stress out that the gain with respect to classical SGD is already substantial, since SGD can be understood as projecting into $Span(NTK_\theta(x_i, \cdot))$ with respect to the push-forward metric of the Euclidian space into tangent space, while considering that points in batch are independent, which is very badly conditioned for PINNs as can be seen experimentally. In the revised version of appendices, we explain this fact with more precision, and provide empirical evidences of it.
>
> > ## Questions:
> > Please see the weakness part.
> > One typo: Lines 126-127: ill conditioned
>
> Thanks for pointing that out !

---

> > ### Comment · Reviewer_iNHt · 2024-11-22
> >
> > Sorry that I still did not find the difference between the proposed method and natural GD. General natural GD directly computes matrix inverse, while you use SVD.
> >
> > Could you please answer my question (3):
> > The ANaGRAM for PINNs closely resembles existing natural GD for training PINNs (https://arxiv.org/abs/2408.00573). For example, in Equation (31) of the paper, the pseudo-inverse of (J J^{T}) is equivalent to the SVD of \hat{\phi} in this paper.

---

> > > ### Author Response · Authors · 2024-11-22
> > >
> > > Maybe there is a misunderstanding on the definition of general natural GD. For us, the general natural GD use a Preconditioner of size $P\times P$ that needs to be estimated and inverted without any assumption on the rank, which leads to a cubic complexity in $P$.
> > >
> > > Based on the bibliography given in (https://arxiv.org/abs/2408.00573) we acknowledge that
> > > the vanilla version of Anagram indeed coincides with the Gram-Gauss-Newton method introduced in https://arxiv.org/pdf/1905.11675 (We will also add this reference) and its formal  adaptation to PINNs in (https://arxiv.org/abs/2408.00573). If the general principle appears already here roughly at the same time, we actually make it work, using the SVD trick. In addition, we want to emphasize as discussed in many parts of the rebuttal that Anagram is more general than that, because we define it by the projection of the functional gradient on the empirical tangent space, which can be approximated in many ways as discussed in new version's Section 6. The crudest way indeed corresponds to GGN. We will try to set up an experimental example to illustrate this point before the end of the rebuttal.

---

### Official Review · Reviewer_UHpU · 2024-11-02

**Soundness:** 3
**Presentation:** 2
**Contribution:** 2
**Rating:** 6
**Confidence:** 4

**Summary:**

This paper proposed a natural gradient based optimizer called ANaGRAM for PINNs. The key algorithmic contribution of the paper includes a novel approach to discretize the natural gradient by connecting the natural gradient with a linear regression in L2 space, which is then combined with the fact that PINNs can be regarded as a least square regression to adapt the method for PINNs. A major theoretical contribution of this is showing that the natural gradient for PINNs is actually Green’s function under some conditions.

In summary, this paper proposed a novel optimization approach which can be used to help improve the convergence of the usually ill-condition problems in PINNs.

**Strengths:**

This paper has an interesting and supportive theoretical results for their results.

**Weaknesses:**

1. There are many typos 126 il conditioned -> ill condition 186 hapens -> happens 201: writes -> write 274 quadraic....
2. The flow of the paper may be better by revisiting some parts. For example, at the end of the section 2, authors illustrate the relationship between the natural gradient and the linear regression problem. However, the section 3.1 has the title indicating the same thing. Maybe the end of section 2 can be moved to section 3.1, otherwise, I don't see the reason why section 3.1 has the current titles.
3. The experiments are not enough. Usually people also include Burgers, Schrodinge, etc.
4. The paper's main theoretical result is the equivalence of Green's function and natural gradient. It would be helpful to include some toy examples to justify this finding.

**Questions:**

1. It's interesting that ANaGRAM usually has the largest variance among all methods, is there a reason for that? I was thinking it should have small variance since it's more well conditioned.

2. For some experiments, it seems the difference between different methods is more stark for L2 loss compared to test Loss. Any explanation for this?

3. Could you explain more about the relationship between NTK and equation 19 (line 223)?

---

> ### Author Response · Authors · 2024-11-21
>
> Thank you very much for the constructive remarks and kind comments ! We address here the weaknesses you have identified and answer here the questions you asked.
>
> > ## Weaknesses
> > There are many typos 126 il conditioned -> ill condition 186 hapens -> happens 201: writes -> write 274 quadraic....
> > The flow of the paper may be better by revisiting some parts. For example, at the end of the section 2, authors illustrate the relationship between the natural gradient and the linear regression problem. However, the section 3.1 has the title indicating the same thing. Maybe the end of section 2 can be moved to section 3.1, otherwise, I don't see the reason why section 3.1 has the current titles.
>
> Thank you very much for all these comments. We have revised the manuscript accordingly. In particular Section 3 has been entirely revised.
>
> > The experiments are not enough. Usually people also include Burgers, Schrodinger, etc.
>
> We thought that our choices represented well enough the different study cases (linear, non-linear, higher dimension ...). But we can definitely also include those experiments.
>
> > The paper's main theoretical result is the equivalence of Green's function and natural gradient. It would be helpful to include some toy examples to justify this finding.
>
> Thank you for this suggestion, we can indeed illustrate this on a set of Fourier features, as well as with figures. We would add this int the revised appendices. That being said, we would like to emphasize that in our opinion there is a second as important theoretical result, which is the introduction of the notion of empirical tangent space and empirical natural gradient, presented in appendix section C, and that we now highlight in section 3 of the revision.
>
> > ## Questions:
> > It's interesting that ANaGRAM usually has the largest variance among all methods, is there a reason for that? I was thinking it should have small variance since it's more well conditioned.
>
> This is only an "optical illusion" due to the logscale. You can have a look at section A.3 in supplementary material where a detailed overview of the numerical results is given.
>
> > For some experiments, it seems the difference between different methods is more stark for L2 loss compared to test Loss. Any explanation for this?
>
> Thanks for this observation. We suspect that this can been explained by the fact that the "residual norm", which is represented by the test loss and the L2 norm have different spectral biases. There is a line of works studying mathematically these kind of effect for which we are not too familiar but which might be relevant to explain that. We refer for instance to the dissertations of Tim De Ryck (https://www.research-collection.ethz.ch/bitstream/handle/20.500.11850/674112/dissertation_deryck.pdf?sequence=1) or Roberto Molinaro (https://www.research-collection.ethz.ch/bitstream/handle/20.500.11850/646749/Thesis%2813%29.pdf?sequence=1).
>
> > Could you explain more about the relationship between NTK and equation 19 (line 223)?
>
> The relationship is explained in appendix C. Nevertheless, since this relation is indeed very important and that equation (19) (in the original version) seems to be misleading, in the sense that it hides the geometric interpretation under the NTK relationship, we have  completely revised this part to discuss the core of Anagram which is now presented in section 3 and encapsulated in theorem 1.
> and its consequence like e.g. eq. (18) in the new version.

---

### Official Review · Reviewer_EAmy · 2024-11-08

**Soundness:** 3
**Presentation:** 4
**Contribution:** 3
**Rating:** 8
**Confidence:** 2

**Summary:**

The paper proposes a new algorithm -- ANaGRAM-- that is well suited for learning PDEs with boundary conditions. The algorithm makes use of a clever manipulation of natural gradient of Amari & Douglas. The natural gradient, PINN and some elements from differential geometry are introduced. The proposed algorithm can be interpreted as follows: compute the integration of the euclidean gradient using by "weighting" with the appropriate Green's function and use pseudo-inverse of the differential as a pre-conditioner (Theorem 1). Experiements show that ANaGRAM learns faster than Adam, GD, E-NGD across well-known low-dimensional PDEs.

**Strengths:**

The paper is very well written. The proposed method is elegant and well motivated from the theory perspective. Empirical results are well-done and show good evidence for the proposed ANaGRAM.

**Weaknesses:**

I quite like the paper. Even though I am not an expert of PINNs at all, I am guessing that this paper may be exciting for that community. I found the math well written, which may be not very common for an ICLR submission.

Typos:
- Eq 2: $h_i \to h_p $
- line 147: "his" comp. cost $ \to $ "its" comp. cost
- line 199: maybe add what $\Delta_{\theta_r}$ means here (singular values)

The performance of ANaGRAM is quite similar to E-NGD for Heat eqn and 5D Laplace eqn. ANaGRAM is faster by a tiny margin. I did not find a discussion of this in the paper. Can the authors comment on this? In particular, what would be the expected advantage of  ANaGRAM over E-NGS in words?

**Questions:**

I find the empirical investigations sufficient for this paper. It might be even better to have a high dimensional example. For example, would ANaGRAM still perform better than E-NGD when tested on 100 D Laplace eqn?

---

> ### Author Response · Authors · 2024-11-21
>
> We would like to thank you very much for the positive and supportive appreciation of our work, as well as your helpful remarks and comments ! We particularly appreciate the acknowledgment on mathematical work ! We address here the weaknesses you have identified and answer here the questions you asked.
>
> > ## Weaknesses
> > The performance of ANaGRAM is quite similar to E-NGD for Heat eqn and 5D Laplace eqn. ANaGRAM is faster by a tiny margin. I did not find a discussion of this in the paper. Can the authors comment on this? In particular, what would be the expected advantage of ANaGRAM over E-NGS in words?
>
> The equivalence between the two algorithms in the case of linear operators is the subject of section E in the appendix. The difference is that ANaGRAM has a better complexity when the number of parameters is significantly larger than the batch size, which is not the case in these experiments. We will try to illustrate that on the High-dimensional (100) Laplace example.
> Also in the non-linear case, the mathematical equivalence between both algorithms does not hold anymore as one may see in the Allen-Cahn example.
>
> > ## Questions
> > I find the empirical investigations sufficient for this paper. It might be even better to have a high dimensional example. For example, would ANaGRAM still perform better than E-NGD when tested on 100 D Laplace eqn?
>
> We did not considered such a high-dimensional equation, as it seems to us a bit far from numerical analysis concerns, for which PINNs are meant. Nevertheless, we will try to investigate this regime.

---

### Official Review · Reviewer_psCC · 2024-11-09

**Soundness:** 3
**Presentation:** 3
**Contribution:** 3
**Rating:** 6
**Confidence:** 4

**Summary:**

The paper "ANaGRAM: A Natural Gradient Relative to Adapted Model for Efficient PINNs Learning" introduces an innovative natural gradient optimization algorithm tailored for Physics-Informed Neural Networks (PINNs). Named ANaGRAM, this approach leverages differential geometric perspectives to optimize PINNs, aiming to enhance computational efficiency and accuracy in solving Partial Differential Equations (PDEs) in a machine learning framework.

**Strengths:**

1.The paper's use of differential geometry to formulate a natural gradient specifically for PINNs counts as a theoretical contribution. By aligning gradient updates with the functional space geometry, ANaGRAM offers a alternative to gradient-based methods, potentially addressing some of the known limitations in PINN optimization.

2 NaGRAM's computational complexity, which scales favorably with parameter and batch sizes, is an improvement over traditional natural gradient methods.

3. Experimental results indicate that ANaGRAM consistently achieves lower error rates compared to baseline methods like E-NGD and standard optimizers (Adam, L-BFGS).

**Weaknesses:**

1. The current implementation of ANaGRAM requires manual tuning of hyperparameters, such as the cutoff factor for SVD.

2. The choice of collocation points (batch points) is limited to heuristics in this study. Developing an adaptive method for selecting these points could further improve ANaGRAM's performance.

3. The results indicate that the efficacy of ANaGRAM may diminish for nonlinear differential operators, where the method's equivalence with traditional approaches is less certain.

**Questions:**

1. Implementing an adaptive selection strategy for the SVD cutoff factor based on the eigenvalue spectrum could make the method easier to use.

2. Developing an adaptive method for selecting collocation points based on the geometry of the domain could improve convergence rates.

---

> ### Author Response · Authors · 2024-11-21
>
> We would like to thank you for your interest and the helpful remarks and comments ! We address here the weaknesses you have identified and answer the questions you asked.
>
> > ## Weaknesses:
> > 1. The current implementation of ANaGRAM requires manual tuning of hyperparameters, such as the cutoff factor for SVD.
> > 2. The choice of collocation points (batch points) is limited to heuristics in this study. Developing an adaptive method for selecting these points could further improve ANaGRAM's performance.
> > 3. The results indicate that the efficacy of ANaGRAM may diminish for nonlinear differential operators, where the method's equivalence with traditional approaches is less certain.
>
> The observed diminution of efficiency for nonlinear operators can be thoroughly explained by the theoretical framework we developed, namely the error stated in equation (66) of the appendix, now the second term stated in theorem 1, that is null for linear operators as stated by Proposition 2 in revised version. Note that a second approximation is made, introduced in equation (75), now the first term stated in Theorem 1.
>
> >  ## Questions:
>  >  1. Implementing an adaptive selection strategy for the SVD cutoff factor based on the eigenvalue spectrum could make the method easier to use.
> > 2. Developing an adaptive method for selecting collocation points based on the geometry of the domain could improve convergence rates.
>
> We do agree on both remarks. Nevertheless, we do think that this is a different line of work, that we are currently carrying on. As already mentioned in the general comment, our main objective was to put on firm ground the natural gradient in PINNs context, so as to have a sound basis on which to develop future extensions (e.g. collocation points, hyperparameters tuning). This paper constitutes for us the first step on which we can elaborate, as it gives us in particular a good criteria for collocation points optimization, namely equation (18) in the new version.

---

### Author Response · Authors · 2024-11-21
**General comment**

We thank the reviewers for their careful reading of the paper and their insightful comments and constructive critics. The main ones can be summarized as:
1. the method does not seem to bring any novelty, as it boils down to Gauss-Newton
2. the method is suboptimal in many respects

## Concerning the first line of critics
We disagree with the reviewers because in our view Anagram is much broader than that, as it offers a different perspective on the natural gradient, by shifting the preconditioning from the parameter space to the "sample space" which is interesting in the context of PINNs since the batch points can be chosen arbitrarily. This leads to the notion of empirical tangent space and empirical natural gradient the latter being geometrically interpreted as the projection of the functional gradient on the empirical tangent space. This projection can be made more or less approximate, and it is true that the version we presented corresponds to the crudest approximation, which indeed happens to be equivalent to empirical Gauss-Newton adapted to the PINNs context (we thank reviewer ogx731 for pointing this to us). We made this choice for two reasons, first because we were already obtaining  empirical results much better than existing baselines, on precision and speed either on one or on both criteria by a large margin; secondly we wanted to insure  a clear presentation in the core paper for pedagogical purposes, which was something that seemed to be appreciated by some of the reviewers, leaving unfortunately the discussion of some key points in appendix which led seemingly to some misunderstanding. Indeed the drawback of this simplified presentation is that it possibly gave the false impression of non original or trivial results.
Summarizing the contributions we have:
* a theoretical framework which introduce the notions of empirical tangent space and empirical natural gradient leading to a family of algorithms (Anagram) which depends on the way the projection of the functional gradient on the empirical tangent space is approximated;
* a key relation showing that our formulation of  Natural Gradient [resp. empirical Natural Gradient] for PINNs coincides with the operators Green function restricted to the tangent [resp. empirical tangent] space, which is also to our best knowledge completely new. This demonstrates  the coherence of the framework and that in some sense it is an optimal order 1 algorithm for PINNs. % will be difficult to define a better natural gradient for PINNs.
* a basic but efficient implementation of the simplest  instantiation  of Anagram, which indeed can be seen as a combination of Gauss-Newton with SVD adapted to PINNs, with good scaling properties, showing robustness and superior empirical results to existing baseline and this by a large margin either on speed or precision or both. We also provide a formal %bound
estimate on the projection error in this case;
* a new, simple and principled optimization criteria for the collocation point problem, which is a direct byproduct of our theoretical framework.

## Concerning the second line of critics
We indeed recognize that many aspects of the method can be improved (choice of the points, heuristic for the cut-off, optimized svd ...).
Actually our focus was to provide a theoretical principled picture addressing the spectral bias, while stressing out the benefits in terms of complexity. Compared to previous natural gradient methods for PINNs (in particular in Jini et al : http://arxiv.org/abs/2402.10680 and Müller and Zeinhofer : https://proceedings.mlr.press/v202/muller23b.html), our method comes with a generic way to lower the cost of natural gradient and an explanation of its optimality through the equivalence with Green's function. What we see experimentally is that we already reach better than state of the art performances without inserting side sophistication in the algorithm, which empirically validate our theory.

We provided a modified version of the paper aiming at highlighting more our contributions listed above and at dissipating some misunderstanding. The control of the empirical gradient projection leading to the Anagram framework is now specified by theorem 1 and the cost function for the batch points which is obtained as a byproduct of the theory is introduced in eq (18). The modification of appendices will be provided soon, taking into account additional developments asked by the reviewers.

---

### Meta-Review · Area_Chair_ijds · 2024-12-21

**Metareview:**

(a) **Summary of Claims and Findings**
The paper proposes ANaGRAM, a novel optimization algorithm based on a natural gradient approach tailored to Physics-Informed Neural Networks (PINNs). The primary contributions are:
1. A theoretical framework that connects the natural gradient for PINNs with Green’s function theory.
2. A new formulation of the natural gradient leveraging the empirical tangent space, enabling scaling relative to parameter and batch sizes.
3. Empirical results demonstrating that ANaGRAM improves the speed and accuracy of PINN training compared to baselines such as Adam, L-BFGS, and E-NGD.

(b) **Strengths**
1. **Theoretical Contribution:** The introduction of the empirical tangent space and its relationship to natural gradients and Green’s function theory provides a solid foundation for the approach.
2. **Algorithmic Design:** The use of SVD for preconditioning improves computational efficiency while maintaining theoretical rigor.
3. **Empirical Validation:** Experimental results consistently show superior performance over baseline methods in terms of error rates and convergence speed.
4. **Clarity:** The paper is well-structured, and the theoretical arguments are well-articulated.

(c) **Weaknesses**
1. **Limited Numerical Exploration:** Although the theoretical framework is robust, the empirical work could include more diverse PDEs and higher-dimensional problems to strengthen the claims.
2. **Manual Hyperparameter Tuning:** Key parameters such as the SVD cut-off factor are manually tuned, which could hinder usability in practice.
3. **Criticisms on Novelty:** Some reviewers argue that the method resembles existing natural gradient approaches, albeit with nuanced improvements. This limits the perceived originality for an exceptional venue like ICLR.
4. **Adaptability to Stochastic Settings:** The method currently requires full-batch training, which is less adaptable than stochastic approaches like SGD.

(d) **Decision Justification**
The paper provides significant theoretical insights and empirical evidence for improving PINN training via natural gradient methods. While concerns regarding novelty and numerical breadth are valid, the methodological and theoretical advancements make this work a meaningful contribution to the field. The authors’ rebuttals adequately addressed the reviewers' main concerns, emphasizing the geometric interpretation of the empirical tangent space and the innovative use of Green's function theory. The decision to accept is based on:
- The strong theoretical contributions.
- Empirical performance demonstrating improvements over state-of-the-art methods.
- Potential impact on the PINN and scientific machine learning communities.

**Additional Comments On Reviewer Discussion:**

The discussion during the rebuttal period highlighted the following key points:
1. **Novelty Concerns:** Reviewer iNHt and ogx7 noted similarities between ANaGRAM and existing methods. The authors clarified that ANaGRAM introduces a dual perspective emphasizing the empirical tangent space, providing a unique theoretical and algorithmic contribution.
2. **Empirical Coverage:** Reviewers suggested including more examples, such as high-dimensional PDEs and diverse operator types. The authors acknowledged this but argued that their selected experiments effectively demonstrated the core contributions.
3. **Implementation Aspects:** Concerns regarding manual hyperparameter tuning and adaptability to stochastic training were raised. The authors indicated these as directions for future work and stressed that the theoretical basis is a foundation for such improvements.
4. **Revised Presentation:** The updated paper clarified the theoretical framework and highlighted its novelty, particularly the geometric interpretation and connections to Green’s function theory. This addressed concerns about the clarity and significance of the contributions.

In weighing these points, the theoretical contributions and empirical improvements outweighed the limitations, justifying acceptance. The reviewers' discussions led to a clearer understanding of the paper's contributions and resolved most critical concerns.

---

### Decision · Program_Chairs · 2025-01-22

Accept (Poster)